# Esoteric Language Models: A Family of Any-Order Diffusion LLMs

Subham Sekhar Sahoo [* 1]  Zhihan Yang [* 2]  Yash Akhauri [† 1]  Johnna Liu [† 1]  Deepansha Singh [† 1]
Zhoujun Cheng [3]  Zhengzhong Liu [3]  Eric Xing [3]  John Thickstun [2]  Arash Vahdat [4]

## Abstract

Diffusion Language Models offer a compelling alternative to autoregressive (AR) models by enabling parallel and controllable generation. Within this family, Masked Diffusion Models (MDMs) currently perform best but still underperform AR models in perplexity and lack key inference-time efficiency features, most notably KV caching. We introduce Esoteric Language Models (Eso-LMs), a new family of models that fuses AR and MDM paradigms, smoothly interpolating between their perplexities while overcoming their respective limitations. Unlike prior work, which uses transformers with bidirectional attention as MDM denoisers, we exploit the connection between MDMs and Any-Order autoregressive models and adopt causal attention. This design lets us (1) **compute the exact likelihood of MDMs for the first time** and, crucially, (2) **allows exact KV caching for MDMs** while preserving parallel generation over the full sequence length for the first time, significantly improving inference efficiency. Combined with an optimized sampling schedule, Eso-LMs establish a new state of the art on the speed-quality Pareto frontier for unconditional generation. We provide the code, model checkpoints, and the video tutorial on the project page:

https://s-sahoo.com/Eso-LMs

## 1. Introduction

Language modeling is undergoing a paradigm shift: Autoregressive (AR) language models, long considered the gold standard, are now being rivaled by diffusion language

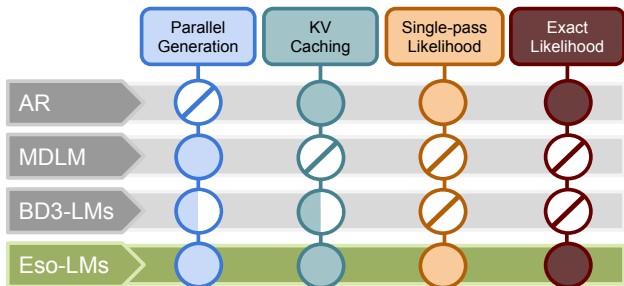

*Figure 1.* Key features supported by our proposed Eso-LMs versus those by relevant baselines: an autoregressive (AR) model, MDLM (Sahoo et al., 2024a), and BD3-LMs (Block Diffusion; Arriola et al., 2025). By combining parallel generation over the full sequence length, exact and complete KV caching, and hybrid modeling, Eso-LMs achieve the state of the art on the speed-quality Pareto frontier for unconditional generation (Fig. 4).

models for standard language generation (Song et al., 2025). Recent works (Sahoo et al., 2024a; Shi et al., 2025; Ou et al., 2025; Arriola et al., 2025) show that Masked Diffusion Models (MDMs) are closing the gap with AR models on small-scale language benchmarks, and even outperform them on tasks involving discrete structures, such as molecular generation (Lee et al., 2025), speech synthesis (Ku et al., 2025) and graph generation (Liu et al., 2023). When scaled to larger sizes (e.g., 8B parameters), MDMs match models like LLaMA on challenging benchmarks such as math, science, and code (Nie et al., 2025).

These results make MDMs a compelling alternative to AR models. However, they suffer from two key limitations: (1) **Inference speed**: Despite supporting parallel generation, MDMs are significantly slower than AR models in practice, largely due to the lack of KV caching, a crucial optimization for real-time applications like chat systems. (2) **Generation quality**: MDMs still show a noticeable likelihood gap on complex language modeling tasks (Sahoo et al., 2024a).

Recently proposed BD3-LMs (Arriola et al., 2025) address the speed issue by introducing a semi-autoregressive generation strategy. These models perform diffusion over fixed-length blocks of text sequentially. Because previously denoised blocks can be cached, BD3-LMs partially support KV caching and are faster than standard MDMs. However, we identify three key shortcomings in BD3-LMs: (1) **De-**

---

[*]The joint first authors contributed equally. Their order is alphabetic and may be rotated to reflect this equal contribution. [†]Joint second authors [1]Cornell Tech [2]Cornell University [3]MBZUAI [4]NVIDIA. Correspondence to: Subham Sekhar Sahoo <ssahoo@cs.cornell.edu>, Zhihan Yang <zhihany@cs.cornell.edu>.

*Proceedings of the 43rd International Conference on Machine Learning*, Seoul, South Korea. PMLR 306, 2026. Copyright 2026 by the author(s).

**graded samples at low sampling steps**: When the number of denoising steps is reduced for faster inference, BD3-LMs exhibit severe degradation in sample quality and diversity—worse than both AR (at high Number of Function Evaluations (NFEs), i.e., number of sampling steps) and other diffusion models (at low NFEs) (Sec. A.2 and Sec. 5.2). (2) **Incomplete KV caching**: While KV caching is possible across blocks, intra-block diffusion still lacks KV support, limiting overall speed gains.

To address these challenges, we propose a language model that fuses AR and masked diffusion paradigms at multiple levels. Our model is trained with a hybrid loss—a combination of AR and MDM objectives—which allows it to interpolate smoothly between the two paradigms in terms of perplexity. This requires two key innovations: (1) A revised attention mechanism in the denoising transformer to support both AR and MDM styles of generation. (2) A new training and sampling procedure that enables KV caching within the diffusion phase, a feature previously unavailable in MDMs. Due to its unconventional design exploring the boundary of two paradigms, we name our method **Eso**teric **L**anguage **M**odels (Eso-LMs), inspired by esoteric programming languages that probe the limits of programming language design. Our main contributions are:

1. We introduce Eso-LMs, a new hybrid AR–MDM language modeling framework that outperforms the previous hybrid approach BD3-LMs and **enables fine-grained interpolation between AR and MDM perplexities**, narrowing the gap to AR models (Sec. 5.1).

2. By enabling exact KV caching during diffusion while preserving parallel generation, **Eso-LMs achieves a new state of the art on the speed-quality Pareto frontier** for unconditional generation: BD3-LMs degrade at low sampling steps, whereas Eso-LMs remains competitive with MDMs in the low-NFE regime and with AR models in the high-NFE regime (Sec. 5.2).

3. On long contexts, Eso-LMs provides $14 - 65\times$ faster inference than standard MDMs and $3 - 4\times$ faster inference than block diffusion baseline (BD3-LMs) (Sec. 5.3).

4. Leveraging properties of the denoising transformer architecture of Eso-LMs, **we derive the first (asymptotically) exact likelihood formula for MDMs** (Sec. 3.3).

## 2. Background

**Notation** We represent scalar discrete random variables that can take $K$ values as 'one-hot' column vectors and define $\mathcal{V} \in \{\mathbf{x} \in \{0,1\}^K : \sum_{i=1}^K \mathbf{x}_i = 1\}$ as the set of all such vectors. In the context of language modeling, $K$ is the vocabulary size and $\mathcal{V}$ is the vocabulary. Let $\mathbf{m} \in \mathcal{V}$

be a special mask vector such that its $K$-th entry is one, i.e., $\mathbf{m}_K = 1$. Define $\mathrm{Cat}(\cdot; \boldsymbol{\pi})$ as the categorical distribution over $K$ classes with probabilities given by $\boldsymbol{\pi} \in \Delta^K$, where $\Delta^K$ denotes the $K$-simplex. Let $\langle \mathbf{a}, \mathbf{b} \rangle$ denote the dot product between vectors $\mathbf{a}$ and $\mathbf{b}$. We use parentheses $()$ to denote ordered sets (tuples) and curly brackets $\{\}$ to denote unordered sets. $|A|$ is the cardinality of the set $A$.

MDMs feature two salient orderings: sequence order and denoising order. We relate them via a permutation $\sigma$. Let $\mathcal{P}_L$ denote the set of all permutations of $[L] = \{1, \ldots, L\}$. A permutation $\sigma \in \mathcal{P}_L$ is both an ordered set (tuple) and a bijective function: $\sigma(\ell)$ gives the sequence position denoised at step $\ell$, and while $\sigma^{-1}(i)$ gives the denoising step of sequence position $i$. For example, $\sigma = (2, 4, 1, 3)$ is a denoising order of $(1, 2, 3, 4)$; $\sigma^{-1}(4) = 2$ means the $4^{\text{th}}$ token in sequence is the $2^{\text{nd}}$ one to denoise.

Let $\mathbf{x} \in \mathcal{V}^L$ denote a sequence of length $L$ with no mask tokens, and let $\mathbf{x}^\ell$ denote the $\ell^{\text{th}}$ entry in $\mathbf{x}$. Note that $\mathbf{x}^\ell$ is one-hot under our notation. We use the term 'token index' to refer to the position of a token in the original ordering, e.g., the token index for $\mathbf{x}^\ell$ is $\ell$. Let $(\mathbf{z}_t)_{t \in [0,1]} \in \mathcal{V}^L$ denote a sequence of length $L$ that may contain mask tokens. Let $\mathcal{M}(\mathbf{z}_t) = \{\ell \mid \mathbf{z}_t^\ell = \mathbf{m}\}$ denote mask token indices in $\mathbf{z}_t$ and $\mathcal{C}(\mathbf{z}_t) = \{\ell \mid \mathbf{z}_t^\ell \neq \mathbf{m}\}$ denote clean token indices in $\mathbf{z}_t$.

Let $\oplus : \mathcal{V}^m \times \mathcal{V}^n \to \mathcal{V}^{m+n}$ denote a concatenation operator on two sequences $\mathbf{x} = (\mathbf{x}^1, \mathbf{x}^2, \ldots, \mathbf{x}^m)$ and $\mathbf{z} = (\mathbf{z}^1, \mathbf{z}^2, \ldots, \mathbf{z}^n)$ of length $m$ and $n$. When $\mathbf{x} \oplus \mathbf{z}$ is fed into the transformer, $\mathbf{x}$ and $\mathbf{z}$ carry the same positional embeddings as they would if they were fed into a transformer independently. Let $\odot : \mathcal{V}^m \times \mathcal{V}^n \to \mathcal{V}^m$ denote a substitution operator; for any $\mathbf{z} \in \mathcal{V}^m$ and $\mathbf{x} \in \mathcal{V}^n$ with $m > n$, the output $\mathbf{y} = \mathbf{z} \odot \mathbf{x}$ is given by: $\mathbf{y}^{1:n} = \mathbf{x}$ and $\mathbf{y}^{n+1:m} = \mathbf{z}^{n+1:m}$.

### 2.1. Autoregressive Models

Given a sequence $\mathbf{x} \in \mathcal{V}^L \sim q_{\text{data}}$, AR models define the following factorization of the joint distribution: $\log p_\theta(\mathbf{x}) = \sum_{\ell=1}^L \log p_\theta(\mathbf{x}^\ell \mid \mathbf{x}^{<\ell})$, where the model $p_\theta$ is usually parameterized by a causal transformer (Vaswani et al., 2017) model. Sampling is sequential, requiring $L$ steps (NFEs) to generate a length-$L$ sequence. Causal attention also enables KV caching (see Suppl. A.1), crucial for efficient inference.

### 2.2. Masked Diffusion Models

Masked Diffusion Models (Austin et al., 2021; Lou et al., 2024; Sahoo et al., 2024b; Shi et al., 2025; Ou et al., 2025) learn to invert a forward masking process $q$ that maps clean data $\mathbf{x} \in \mathcal{V}^L \sim q_{\text{data}}$ to latent sequences $\mathbf{z}_t \in \mathcal{V}^L$ for $t \in [0, 1]$, where each $\mathbf{z}_t$ is a progressively noisier (more masked) version of $\mathbf{x}$. The forward process factorizes across positions, $q_t(\mathbf{z}_t|\mathbf{x}) = \prod_\ell q_t(\mathbf{z}_t^\ell|\mathbf{x}^\ell)$. The marginal of each token at $t$ is

$$q_t(\mathbf{z}_t^\ell|\mathbf{x}^\ell) = \mathrm{Cat}(\mathbf{z}_t^\ell; \alpha_t \mathbf{x}^\ell + (1 - \alpha_t)\mathbf{m}), \quad (1)$$

where $\alpha_t \in [0, 1]$ is a strictly decreasing function in $t$ with $\alpha_0 \approx 1$ and $\alpha_1 \approx 0$; a standard choice is the linear schedule $\alpha_t = 1 - t$. The reverse posterior $q_{s|t}(\mathbf{z}_s^\ell | \mathbf{z}_t^\ell, \mathbf{x}^\ell)$ for $s < t$ is

$$q_{s|t}(\mathbf{z}_s^\ell | \mathbf{z}_t^\ell, \mathbf{x}^\ell) = \begin{cases} \text{Cat}(\mathbf{z}_s^\ell; \mathbf{z}_t^\ell) & \mathbf{z}_t^\ell \neq \mathbf{m}, \\ \text{Cat}\left(\mathbf{z}_s^\ell; \frac{(1-\alpha_s)\mathbf{m} + (\alpha_s - \alpha_t)\mathbf{x}^\ell}{1 - \alpha_t}\right) & \mathbf{z}_t^\ell = \mathbf{m}. \end{cases} \quad (2)$$

**Training**   Let $\mathbf{x}_\theta : \mathcal{V}^L \rightarrow (\Delta^K)^L$ denote a denoising model, typically implemented as a transformer with bidirectional attention. We parameterize the reverse unmasking process over the sequence $\mathbf{z}_s$ as

$$p_{s|t}^\theta(\mathbf{z}_s | \mathbf{z}_t) = \prod_\ell p_{s|t}^\theta(\mathbf{z}_s^\ell | \mathbf{z}_t) = \prod_\ell q_{s|t}(\mathbf{z}_s^\ell | \mathbf{z}_t^\ell, \mathbf{x}^\ell = \mathbf{x}_\theta^\ell(\mathbf{z}_t)). \quad (3)$$

The resulting Negative Evidence Lower Bound (NELBO) is

$$\mathcal{L}_{\text{MDM}}(\mathbf{x}) = \underbrace{\mathbb{E}_{q_t, t \sim [0,1]}\left[\frac{\alpha_t'}{1 - \alpha_t} \sum_{\ell \in \mathcal{M}(\mathbf{z}_t)} \log\langle \mathbf{x}_\theta^\ell(\mathbf{z}_t), \mathbf{x}^\ell\rangle\right]}_{\mathcal{L}_{\text{MDM}}}$$
$$\equiv \underbrace{-\mathbb{E}_{\sigma \sim \mathcal{P}_L}\left[\sum_{\ell=1}^L \log p_\theta(\mathbf{x}^{\sigma(l)} | \mathbf{x}^{\sigma(<l)})\right]}_{\mathcal{L}_{\text{AO}}}, \quad (4)$$

where the middle expression is a weighted masked language modeling loss over the masked positions $\mathcal{M}(\mathbf{z}_t)$ (Sahoo et al., 2024a; Shi et al., 2025; Ou et al., 2025). Ou et al. (2025) further shows that this is equivalent to the autoregressive loss (a sum capturing all $L$ latents on a diffusion trajectory) averaged over all possible permutations of the input (4, line 2); we dub this Any-Order NEBLO as $\mathcal{L}_{\text{AO}}$. In $\mathcal{L}_{\text{AO}}$ (4), we have $p_\theta(\mathbf{x}^{\sigma(l)} | \mathbf{x}^{\sigma(<l)}) = \left\langle \mathbf{x}_\theta^{\sigma(l)}(\mathbf{x}^{\sigma(<l)}), \mathbf{x}^{\sigma(l)}\right\rangle$, in which $\mathbf{x}_\theta$ is applied to $\mathbf{x}^{\sigma(<l)} \in \mathcal{V}^L$, i.e., the sequence in which all entries other than the first $\ell - 1$ elements (under the permutation $\sigma$) are masked out.

**(Ancestral) Sampling**   To generate a sequence of length $L$, the reverse process starts from a fully masked sequence $\mathbf{z}_{t=1}$ with $(\mathbf{z}_{t=1}^\ell = \mathbf{m})_{\ell \in [L]}$. Time is discretized into $T \leq L$ steps with step size $\Delta = 1/T$, and at each step we update from $t$ to $s = t - \Delta$. At every denoising step, a mask token transitions to a clean token with probability $(\alpha_s - \alpha_t)/(1 - \alpha_t)$, as implied by (2). This can be viewed as two sub-steps. Let $n_t$ be the number of mask tokens denoised at time $t$. Then

$$n_t \sim \text{Binomial}\left(n = |\mathcal{M}(\mathbf{z}_t)|, \ p = \frac{\alpha_s - \alpha_t}{1 - \alpha_t}\right), \quad (5)$$

where $|\mathcal{M}(\mathbf{z}_t)|$ is the number of mask tokens at time $t$. Next, $n_t$ positions are sampled uniformly from $\mathcal{M}_t$ and independently denoised according to the probabilities given by $\mathbf{x}_\theta(\mathbf{z}_t)$. The ancestral sampler enforces that clean tokens are never remasked. Because multiple tokens are updated in parallel, the total number of steps (NFEs) can be smaller than $L$, enabling faster generation. However, each denoising forward pass is more expensive than in AR models, since bidirectional attention in the denoising transformer prevents KV caching; see Suppl. A.1 for further discussion.

## 2.3. Block Discrete Diffusion Models

Block Denoising Diffusion Discrete Language Models (BD3-LMs; Arriola et al., 2025) autoregressively model blocks of tokens and perform masked diffusion modeling (Sec. 2.2) within each block. By changing the size of blocks, BD3-LMs interpolate AR models and MDMs. BD3-LMs group tokens in $\mathbf{x}$ into $B$ blocks of $L'$ consecutive tokens with $B = L/L'$, where $B$ is an integer. The likelihood over $\mathbf{x}$ factorizes autoregressively over blocks as $-\log p_\theta(\mathbf{x}) = -\sum_{b=1}^B \log p_\theta(\mathbf{x}^b | \mathbf{x}^{<b}) \leq \sum_{b=1}^B \mathcal{L}_{\text{MDM}}(\mathbf{x}^b; \mathbf{x}^{<b})$, where $p_\theta(\mathbf{x}^b | \mathbf{x}^{<b})$ is a conditional MDM and $\mathcal{L}_{\text{MDM}}(\mathbf{x}^b; \mathbf{x}^{<b})$ is the NELBO for MDLM as defined in (4), applied within a block. During generation, we use $T' = T/L'$ to denote the number of diffusion sampling steps per block.

# 3. Esoteric Language Models

In this section, we propose a new paradigm for language modeling: **Eso**teric **L**anguage **M**odels (Eso-LMs), which form a symbiotic combination of AR models and MDMs.

AR models currently achieve state-of-the-art language modeling performance but generate tokens sequentially, making inference slow. In contrast, MDMs generate multiple tokens in parallel and are well-suited to controllable generation (Schiff et al., 2025; Nisonoff et al., 2024), but they typically have higher (worse) perplexity than AR models (Sahoo et al., 2024a; 2025). This raises a natural question: can we design an algorithm that combines their strengths? We propose a hybrid generative process (Fig. 2) in which an MDM first generates a partially masked sequence in parallel, and an AR model then fills in the remaining tokens left-to-right. This design leads to two key questions. (i) Can we compute the likelihood of such a generative process? We show that Eso-LMs admits a principled variational bound on the true likelihood. (ii) How can we adapt the attention mechanism so that a single transformer (Vaswani et al., 2017) supports both generation styles? We address this in Sec. 4.

## 3.1. Fusing Autoregressive & Masked Diffusion Models

Let $p_\theta$ denote our generative process parameterized by $\theta$. Eso-LMs decomposes $p_\theta$ into two components: an MDM component $p_\theta^{\text{MDM}}$, which generates a partially masked sequence $\mathbf{z}_0 \in \mathcal{V}^L$ in parallel, $\mathbf{z}_0 \sim p_\theta^{\text{MDM}}(\mathbf{z}_0)$, and an AR component $p_\theta^{\text{AR}}$, which unmasks the remaining mask tokens sequentially, $\mathbf{x} \sim p_\theta^{\text{AR}}(. | \mathbf{z}_0)$. The marginal data distribution for this hybrid process is given as:

$$p_\theta(\mathbf{x}) = \sum_{\mathbf{z}_0 \in \mathcal{V}^L} p_\theta^{\text{AR}}(\mathbf{x} | \mathbf{z}_0) p_\theta^{\text{MDM}}(\mathbf{z}_0). \quad (6)$$

### 3.1.1. TRAINING

Computing the exact likelihood $\log p_\theta(\mathbf{x})$ is intractable, but we can obtain a variational bound (Kingma & Welling, 2014) on the true likelihood using a posterior $q(\mathbf{z}_0 | \mathbf{x})$.

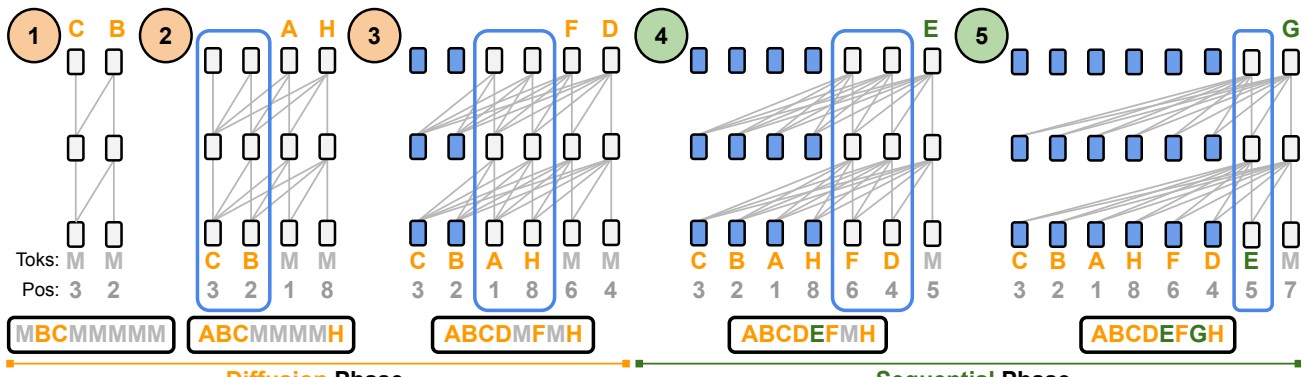

*Figure 2.* Efficient generation of an example sequence with our proposed Eso-LMs. During **Diffusion** Phase, Eso-LMs denoise one or more, potentially non-neighboring mask tokens (M) per step. During **Sequential** Phase, Eso-LMs denoise the remaining mask tokens one at a time from left to right. Eso-LMs allow for **KV caching in both phases** using just **a single unified KV cache**: **blue** bounding boxes enclose transformer cells that are building their KV cache; a cell becomes **blue** once its KV cache is built. The sequences below the transformers show tokens in natural order.

Since $p_\theta^{\mathrm{MDM}}$ models masked sequences, we choose $q$ to be a simple masking distribution. Specifically, we set $q(\mathbf{z}_0|\mathbf{x}) = q_0(\mathbf{z}_0|\mathbf{x})$ as defined in (1), which independently masks each token $(\mathbf{x}^\ell)_{\ell \in [L]}$ with probability $(1-\alpha_0)_{\alpha_0 \in [0,1]}$. Intuitively, $\alpha_0$ is the expected fraction of tokens in $\mathbf{x}$ generated by the MDM. In Suppl. B.1, we demonstrate that this yields the following variational bound:

$$
-\log p_\theta(\mathbf{x}) \le \mathbb{E}_{\mathbf{z}_0 \sim q_0}\Bigg[ \underbrace{- \sum_{\ell \in \mathcal{M}(\mathbf{z}_0)} \log \langle \mathbf{x}_\theta^\ell(\mathbf{z}_0 \odot \mathbf{x}^{<\ell}), \mathbf{x}^\ell \rangle}_{\text{AR loss}} \Bigg]
$$

$$
+ \underbrace{\mathbb{E}_{q_t, t \in [0,1]}\Bigg[ \frac{\alpha_t'}{1-\alpha_t} \sum_{\ell \in \mathcal{M}(\mathbf{z}_t)} \log \langle \mathbf{x}_\theta^\ell(\mathbf{z}_t), \mathbf{x}^\ell \rangle \Bigg]}_{\text{MDM loss}}. \quad (7)
$$

Here, $\mathbf{x}_\theta : \mathcal{V}^L \to (\Delta^K)^L$ is the shared denoising model used by both $p_\theta^{\mathrm{AR}}$ and $p_\theta^{\mathrm{MDM}}$ in (6). We implement $\mathbf{x}_\theta$ as a transformer; its attention mechanism is described in Sec. 4. Following common practice, we use the linear noise schedule $\alpha_t = \alpha_0(1-t)$. The AR loss in (7) features a cross-entropy loss between $\mathbf{x}_\theta^\ell(\mathbf{z}_0 \odot \mathbf{x}^{<\ell})$ and $\mathbf{x}_\ell$, where the substitution operator $\odot$ replaces the first $\ell-1$ tokens in $\mathbf{z}_0$ with $\mathbf{x}^{<\ell}$. This ensures that each mask token denoised by the AR model has clean tokens to its left.

> **Corollary:** When $\alpha_0 = 1$ (full diffusion mode), $\mathbf{z}_0$ has no mask tokens and $\mathcal{L}_{\mathrm{NELBO}}$ reduces to the MDM loss, so Eso-LMs ($\alpha_0 = 1$) is an MDM. When $\alpha = 0$ (full AR mode), $\mathbf{z}_0$ is fully masked and $\mathcal{L}_{\mathrm{NELBO}}$ reduces to the AR loss. Hence, **Eso-LMs interpolates between AR and MDM as $\alpha_0$ varies**.

### 3.2. Sampling

Eso-LMs sample in two distinct phases: an MDM phase with parallel generation and an AR phase with sequential

generation. We describe this combined procedure via a *unified denoising schedule* that specifies the subset of tokens denoised at each sampling step.

**Denoising Schedule** As described in Sec. 2.2, the generation order in the MDM phase is random: at each denoising step, the number of mask tokens to unmask is specified by (5), and these positions are chosen uniformly at random among the masked tokens in the sequence. Hence, under this standard ancestral sampler, we can recursively pre-compute the order in which tokens will be denoised. We refer to this as the *diffusion denoising schedule*, denoted by $\mathcal{S}^{\mathrm{MDM}} = (S_1, \ldots, S_{1/T})$, where $S_t$ is the (ordered) subset of mask-token indices denoised at diffusion step $t$, and $T$ is the total number of denoising steps. After the MDM phase has generated an expected $\alpha_0$ fraction of tokens, the sequential AR phase unmasks the remaining tokens in a left-to-right fashion. We define the *AR denoising schedule* as $\mathcal{S}^{\mathrm{AR}} = ((i) \mid i \in \mathcal{M}(\mathbf{z}_0))$, where the mask indices in $\mathcal{M}(\mathbf{z}_0)$ appear in strictly ascending order.

Finally, we define the *unified denoising schedule* as $\mathcal{S} = \mathcal{S}^{\mathrm{MDM}} \cup \mathcal{S}^{\mathrm{AR}}$, the concatenation of the two schedules, which partitions $[L]$. When $\alpha_0 = 1$, all tokens are generated by diffusion ($\mathcal{S} = \mathcal{S}^{\mathrm{MDM}}$ and $\mathcal{S}^{\mathrm{AR}} = \varnothing$); when $\alpha_0 = 0$, all tokens are generated sequentially ($\mathcal{S} = \mathcal{S}^{\mathrm{AR}}$ and $\mathcal{S}^{\mathrm{MDM}} = \varnothing$). NFEs $= |\mathcal{S}|$ for this sampler. See Alg. 2 for the full algorithm for pre-computing $\mathcal{S}$ and Suppl. B.5 for an illustrative example.

**KV Caching** One goal of our design is to eliminate inference-time redundancy in MDMs. Sampling begins from a fully masked sequence $\mathbf{z}_{t=1} = \mathbf{m}^{1:L}$. Standard ancestral sampling as implemented in MDLM (Sec. 2.2) updates only a subset of mask tokens at each step but still performs a forward pass over the full sequence, wasting FLOPs. To improve sampling efficiency, (i) at sampling step $k$ we restrict the forward pass to only the clean tokens and the current

mask tokens to be updated, i.e., $\cup_{i \le k} S_i$, instead of the entire context. This substantially reduces computation, especially for long sequences. (ii) To unlock KV caching, previously predicted tokens must not depend on future tokens that will be denoised, which requires causal attention over the input $\cup_{i \le k} S_i$. Fig. 2 visualizes (i) and (ii). In Sec. 4.1, we describe a training method supporting this style of generation.

### 3.3. Tractable and Exact Likelihood Estimation

**Single-Pass NELBO Estimation**   Reinforcement learning (RL) is a key technique for improving LLM reasoning (Shao et al., 2024). A major bottleneck for applying RL to MDMs is that the policy-gradient objective (e.g., GRPO in Shao et al. (2024)) requires evaluating the likelihood / NELBO of a data sample $\mathbf{x}$, yet this is intractable for standard MDMs. **In contrast, for Eso-LMs, the NELBO becomes tractable under the AO formulation** (4, $\mathcal{L}_{AO}$). This makes Eso-LMs particularly suitable for RL-based finetuning.

For standard MDMs, computing $\mathcal{L}_{MDM}$ (4) for a given $\mathbf{x}$ via Monte Carlo (MC) estimation requires approximately $L$ samples of $t$, where each sample entails a forward pass of the denoising model over the full sequence length. However, this quantity can be computed equivalently using $\mathcal{L}_{AO}$ (4) with a single MC sample of $\sigma$ because each $\sigma$ captures an entire diffusion trajectory of $L$ latents, as discussed in Sec. 2.2. While this still requires $L$ forward passes for standard MDMs, it requires only a single forward pass for Eso-LMs (see Corollary of Sec. 4.1.2 for details). Notably, our estimator has been adopted by Wang et al. (2025b), where it is used as the likelihood estimator for GRPO (Nie et al., 2025), outperforming Black et al. (2024) and Zhao et al. (2025) at the 0.1B and the 8B scale respectively.

**Exact Likelihood Estimation**   Leveraging this tractability, we prove an importance-weighted (IW) bound to estimate the exact likelihood for Eso-LMs in full diffusion mode ($\alpha_0 = 1$). **This is the first (asymptotically) exact likelihood formula for MDMs.**

**Theorem 3.1.** *Let $\mathcal{L}_{AO}^K$ denote the IW bound:*

$$-\mathbb{E}_{\sigma_{1:K} \sim \mathcal{P}_L} \left[ \log \frac{1}{K} + \log \sum_{k=1}^{K} \exp \left( \sum_{l=1}^{L} \log p_\theta(\mathbf{x}^{\sigma_k(l)} \mid \mathbf{x}^{\sigma_k(<l)}) \right) \right] \quad (8)$$

*where $\mathbf{x}^{\sigma_k(<l)}$ is the sequence $\mathbf{x}$ in which all entries other than the first $\ell - 1$ elements (under the permutation $\sigma_k$) are masked out. Then, the following chained inequality holds for all $K \ge 1$, generalizing (4, $\mathcal{L}_{AO}$) by Ou et al. (2025):*

$$-\log p_\theta(\mathbf{x}) \le \mathcal{L}_{AO}^K \le \mathcal{L}_{MDM}. \quad (9)$$

*Crucially, $\mathcal{L}_{AO}^K$ monotonically decreases as $K$ increases, and converges to $-\log p_\theta(\mathbf{x})$ as $K \to \infty$.*

See Suppl. B.2 for the proof. In principle, (8) also applies to standard MDMs, but since $\mathcal{L}_{AO}$ is intractable for them,

this bound is likewise intractable. Kong et al. (2023) also present an exact likelihood formula for MDMs, which we discuss critically in Suppl. E.1. One can estimate the exact likelihood of Eso-LMs for $\alpha_0 < 1$ using an analogous formula (22) that generalizes (8); we prove it in Suppl. B.3.

# 4. Attention Mechanisms for the Shared Denoising Transformer

We now present a unified attention scheme that enables both sequential (AR) and parallel (MDM) generation using a shared transformer architecture. Our main technical contribution is a flexible attention mechanism that reconciles the architectural mismatch between AR models, which require causal attention and shift-by-one prediction, and MDMs, which rely on bidirectional attention. To this end, we introduce an attention bias matrix $A \in \{-\infty, 0\}^{L' \times L'}$, where $L'$ is the input length, that modulates the standard attention as:

$$\text{ATTENTION}(Q, K, V, A) = \texttt{softmax}\left( (QK^\top)/\sqrt{d} + A \right) V$$

where $Q, K, V \in \mathbb{R}^{L' \times d}$ denote the query, key, and value matrices; $A$ controls information flow: $A_{i,j} = 0$ "permits" and $A_{i,j} = -\infty$ "blocks" attention from token $i$ to $j$.

### 4.1. Training

Our training objective in (7) has two terms: an AR loss and a diffusion loss. Given a batch of clean sequences, we train a fraction $\kappa$ with the diffusion objective and the remaining $1 - \kappa$ with the AR objective (Fig. 3). We set $\kappa = 0.5$ based on the ablation in Table 4; for $\alpha_0 = 1$ we use $\kappa = 1$. Below we describe the attention biases used for each loss. Code for the full transformer forward pass is shown in Fig. 12 and requires only minor changes compared to AR and MDLM.

#### 4.1.1. DIFFUSION PHASE

The diffusion sampling scheme in Sec. 3.2 motivates our training setup. It has three key properties: (i) clean tokens are generated in random order, (ii) the forward pass should be restricted to clean tokens and the current mask tokens to be denoised, and (iii) the previously predicted tokens must not depend on the future tokens that will be denoised. We adopt a simple solution: given $\mathbf{z}_t \sim q_t(\cdot \mid \mathbf{x})$, we shuffle $\mathbf{z}_t$ (with their corresponding positional embeddings) such that clean tokens precede masked tokens, and we replace bidirectional attention with standard left-to-right causal attention (Fig. 3). See Suppl. B.6 for a detailed explanation.

#### 4.1.2. SEQUENTIAL PHASE

Given $\mathbf{z}_0 \sim q_0(\cdot \mid \mathbf{x})$, the AR term in (7) applies a cross-entropy loss to the logits at each mask token $(\mathbf{z}_0^i)_{i \in \mathcal{M}(\mathbf{z}_0)}$, which requires a clean left context. This is non-trivial because many mask tokens in $\mathbf{z}_0$ do not have a fully clean left context. We address this by feeding the concatenated

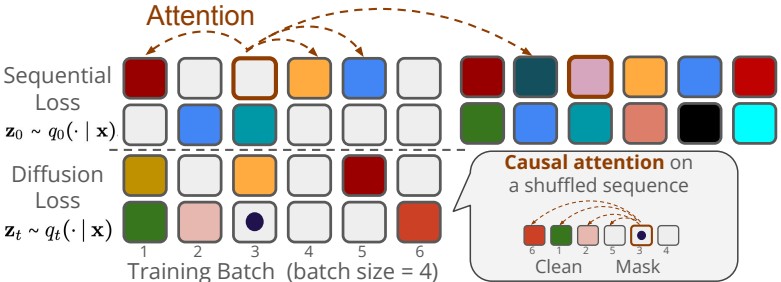

*Figure 3.* To train a transformer to support both sequential and diffusion generation with KV caching, we use half of the training batch (2 sequences in this example) for diffusion training and the other half for sequential training. Tokens for sequential training are **masked** with $p = 1 - \alpha_0$, while tokens for diffusion training are **masked** with $p = 1 - \alpha_t$ with $t \sim \mathcal{U}[0, 1]$. (□) For sequential training, a mask token attends to clean tokens and clean versions of mask tokens on its left. (◉) For diffusion training, a mask token attends to all clean tokens and prior mask tokens after shuffling.

sequence $\mathbf{z}_0 \oplus \mathbf{x}$ into the transformer and designing a specialized attention mask so that $(\mathbf{z}_0^i)_{i \in \mathcal{M}(\mathbf{z}_0)}$ can attend to $\mathbf{x}^{<i}$ (Fig. 3). The transformer outputs over $\mathbf{x}$ are ignored. Since only half of each batch is used for sequential training, the doubled sequence length has limited impact on training speed (Fig. 16). In sampling, this concatenation is not needed, as mask tokens are filled out from left to right.

**Specialized Attention Mask**  During sequential sampling, we reuse the KV values of the clean tokens in $\mathbf{z}_0$, which were generated in a random order during the diffusion phase. Training must therefore enforce causal attention for different random orders of clean tokens $\{\mathbf{x}^i \mid i \in \mathcal{C}(\mathbf{z}_0)\}$. Given $\mathbf{z}_0 \sim q_0(\cdot \mid \mathbf{x})$, we sample a permutation $\sigma \sim \mathcal{P}_L$ such that (i) clean tokens precede mask tokens, and (ii) mask tokens remain in natural order. We then enforce the desired information flow by applying a structured sparse $2L \times 2L$ attention bias $A$ (which depends on $\sigma$) on $\mathbf{z}_0 \oplus \mathbf{x}$. We provide the full mathematical definition of $A$ in (26)-(32).

**Simplified and Efficient Implementation**  When rows and columns of each of $A$'s four $L \times L$ blocks are sorted by $\sigma$, $A$ displays classic patterns (Fig. 8) that are simple to implement via FlexAttention (Dong et al., 2024) (Fig. 9).

> **Corollary:** This specialized attention bias **allows Eso-LMs to estimate the NELBO of a data-point in a single forward pass** via the Any-Order formulation in (4); see Suppl. B.8. **This unlocks tractable exact likelihood for Eso-LMs** via (8).

### 4.2. Sampling

Given a denoising schedule $\mathcal{S}$ as defined in Sec. 3.2, sampling proceeds as follows. At step 1, we run a forward pass on the initial set of mask tokens $\mathcal{S}_1$; since all positions are masked, we do not cache the KV values. At step 2, we run a forward pass on the now-clean tokens in $\mathcal{S}_1$ together with the mask tokens in $\mathcal{S}_2$, denoising $\mathcal{S}_2$ while caching KV values for $\mathcal{S}_1$. For each step $k > 2$, we run a forward pass on the clean tokens from $\mathcal{S}_{k-1}$ and the mask tokens in $\mathcal{S}_k$

while reusing the cached KV values for tokens in $\mathcal{S}_{<k-1}$. See Fig. 2 for a visual. In principle, our sampler can follow any denoising schedule, including ones unseen during training, enabling flexible inference-time trade-offs (Sec. 5.2).

## 5. Experiments

We evaluate Eso-LMs on two standard language modeling benchmarks: the One Billion Words dataset (LM1B; Chelba et al., 2014) & OpenWebText (OWT; Gokaslan et al., 2019). We describe data processing, model architecture, training, and hardware details in Suppl. C.3. Pre-training experiments & ablations add up to ~9K H200 GPU hours, as pre-training is compute-intensive even for small models; see breakdown in Suppl. C.3. Downstream tasks are left for future work.

### 5.1. Likelihood Evaluation

> **Finding 1: Eso-LMs enable fine-grained interpolation between MDM and AR perplexities** on LM1B and OWT (Table 1) by adjusting $\alpha_0$ for training.

**Experimental Setup**  The primary baselines for Eso-LMs are an autoregressive Transformer (AR), the state-of-the-art MDM MDLM (Sahoo et al., 2024a), and BD3-LMs (Arriola et al., 2025), which also interpolate between MDM and AR and support KV caching. In discrete diffusion models, the denoising transformer is typically a DiT (Peebles & Xie, 2023), a standard Transformer augmented with Adaptive LayerNorm (Ada-LN) to condition on the diffusion timestep. For MDMs this conditioning is not required, so Sahoo et al. (2024a); Arriola et al. (2025) fix the DiT timestep to $t = 0$. Because Ada-LN increases the parameter count, we train the AR baseline both with and without Ada-LN. All models are trained with batch size 512, following prior work. Unless stated otherwise, we split the batch evenly ($\kappa = 0.5$) between the AR and diffusion losses; see Table 4 for an ablation over $\kappa$ and Algo. 1 for the full training procedure. Attention biases are configured as in Sec. 4. When training Eso-LMs as a pure MDM ($\alpha_0 = 1$), the full batch uses

*Table 1.* Test perplexities (PPL; ↓) on LM1B ($L = 128$, 1M steps) and OWT ($L = 1024$, 250K steps). For diffusion models, we report PPL computed using the NELBO (7) as in prior work. For Eso-LMs, we report the exact PPL as described in Sec. 3.3 and Sec. 5.1. **Bold** values highlight the best PPL in each method category. ¶No sentence packing. △Reported in He et al. (2022). ‡Reported in Sahoo et al. (2025). †Reported in Arriola et al. (2025). ⊡Denotes models trained from scratch (not finetuned from MDLM unlike in Arriola et al. (2025)). °250K checkpoints by Sahoo et al. (2024a; 2025); Schiff et al. (2025). See Suppl. E.3 for references.

| Method | LM1B | | OWT | |
|---|---|---|---|---|
| | PPL (↓) | | PPL (↓) | |
| | Exact | NELBO | Exact | NELBO |
| *Autoregressive (AR)* | | | | |
| Transformer | 22.83‡ | – | 17.90° | – |
| + AdaLN | **21.86** | – | **17.78** | – |
| *Diffusion* | | | | |
| D3PM Uniform | – | 137.90¶ | – | – |
| D3PM Absorb | – | 76.90¶ | – | – |
| Diffusion-LM | – | 118.62¶△ | – | – |
| DiffusionBert | – | 63.78 | – | – |
| SEDD Absorb | – | 32.71¶‡ | – | 26.81° |
| SEDD Uniform | – | 40.25¶ | – | – |
| MDLM | 26.82 | **31.78**‡ | – | **25.19**° |
| UDLM | – | 36.71‡ | – | 30.52° |
| Duo | – | 33.68‡ | – | 27.14° |
| *Interpolating diffusion & AR* | | | | |
| BD3-LMs | | | | |
| $L' = 16$ | – | 30.60† | – | 23.57⊡ |
| $L' = 8$ | – | 29.83† | – | 22.04⊡ |
| $L' = 4$ | – | **28.23**† | – | **20.96**⊡ |
| **Eso-LMs (Ours)** | | | | |
| $\alpha_0 = 1$ | 31.65 | 36.12 | 29.31 | 30.06 |
| $\alpha_0 = 0.5$ | 28.07 | 32.53 | 26.61 | 27.94 |
| $\alpha_0 = 0.25$ | 24.80 | 29.23 | 23.15 | 24.71 |
| $\alpha_0 = 0.125$ | 23.02 | 26.29 | 20.53 | 21.92 |
| $\alpha_0 = 0.0625$ | 22.39 | 24.53 | – | – |
| $\alpha_0 = 0$ | **21.86** | – | **17.78** | – |

the MDM loss, and we replace the diffusion coefficient $\alpha_t'/(1 - \alpha_t)$ with $-1$, which empirically reduced training variance and improved convergence. We train Eso-LMs with $\alpha_0 \in \{0, 0.125, 0.25, 0.5, 1.0\}$ on LM1B and OWT.

**AR-MDM Interpolation** Ada-LN improves the perplexity (PPL; ↓) of the AR model by about 1 point on LM1B and 0.1 on OWT as shown in Table 1. For all diffusion models, PPL is computed from the upper bound (7) on the negative log-likelihood, which we denote as PPL (NELBO). For Eso-LMs, we additionally compute the exact likelihood as discussed in the subsequent section.

On both LM1B and OWT, Eso-LMs smoothly interpolate between diffusion and AR perplexities, with $\alpha_0 = 0$ recovering the true AR likelihood. However, Eso-LMs have a worse PPL (NELBO) than MDLM by ~ 4 points at $\alpha_0 = 1$ on LM1B (Table 1), as the former is MDLM with sparse

causal rather than bidirectional attention. The same trend holds on OWT. We discuss how to choose $\alpha_0$ in Sec. 5.2.

**Zero-Shot Likelihood** As shown in Table 5, on 4 out of 7 unseen datasets, the ordering of AR < Eso-LMs ($\alpha_0 = 0.125$) < MDLM perplexities is preserved, consistent with the OWT validation results. Sahoo et al. (2026) scale Eso-LM ($\alpha_0 = 1$) to 1.7B, showing that they achieve competitive accuracy with MDLM on zero-shot likelihood downstream tasks.

> **Remark 1:** In diffusion models, **perplexity** measures sample quality only under an infinite sampling budget ($T = \infty$) and **does not reflect performance under realistic finite-time sampling**. As a result, it fails to capture the efficiency advantages of Eso-LMs over MDLM: although Eso-LMs ($\alpha_0 = 1$) achieve worse perplexity than MDLM, they consistently produce higher-quality samples at every fixed sampling-time budget (see Sec. 5.2).

**Single-Pass NELBO Variance** As discussed in Sec. 3.3, the single-sample MC estimator of $\mathcal{L}_{AO}$ (4) should have lower variance than that of $\mathcal{L}_{MDM}$ (4) because a single order captures all latents along one diffusion trajectory. We verify this on one OWT test sequence over 100 trials (Table 2).

*Table 2.* The MC estimator of $\mathcal{L}_{AO}$ exhibits lower variance.

| NELBO Estimator | MC Samples / Trial | Mean | S.D. |
|---|---|---|---|
| $\mathcal{L}_{MDM}$ (over $t$ and $\mathbf{z}_t$) | 10 | 3.25 | 0.56 |
| $\mathcal{L}_{AO}$ (over $\sigma$) | 1 | 3.28 | **0.03** |

**Exact Likelihood Estimation** We compute the exact likelihood for Eso-LMs using (22) with $K=5000$ and $K=1000$ for LM1B and OWT, respectively. To our knowledge, **this is the first work to report exact likelihoods for MDMs**. The gap between the true PPL and the PPL (NELBO) is much larger on LM1B than on OWT, likely due to the shorter context length. As $\alpha_0$ decreases and Eso-LMs become more autoregressive, this gap shrinks. Notably, for Eso-LMs with $\alpha_0 = 1$, the true PPL on OWT nearly matches MDLM's NELBO PPL. As discussed in Sec. 3.3, IW bounds require $K \times L$ forward passes for MDLM. While this is intractable in general, we found $K=1000$ to be tractable for MDLM on LM1B due to LM1B's short sequence length ($L=128$). Notably, we see that the estimated exact likelihood for MDLM still lags behind the AR model.

**Ablation** In Table 1, Eso-LMs in full diffusion mode ($\alpha_0 = 1$) have worse perplexity than MDLM. To study this, we ablate the changes made when converting MDLM to Eso-LMs. MDLM uses bidirectional attention over the full context, whereas Eso-LMs introduce: (1) causal attention on mask tokens with bidirectional attention among clean tokens; (2) causal attention on both clean and mask tokens. We define a family of models, Eso-LMs (A) (see Suppl. D), that

apply only change (1) to MDLM. In Table 9 and Table 10, Eso-LMs (A) at $\alpha_0 = 1$ matches MDLM perplexity, unlike Eso-LMs. Since Eso-LMs (A) does not support KV caching during diffusion, we do not pursue it further. **This suggests that causal attention over clean tokens drives the likelihood gap between MDLM and Eso-LMs at $\alpha_0 = 1$.** As shown in Table 9 and Table 10, Eso-LMs (A) also interpolates between MDLM and AR perplexities on LM1B and OWT, achieving better perplexity than Eso-LMs for every $\alpha_0$.

### 5.2. Pareto Frontier of Generation Speed vs. Quality

> **Finding 2: Eso-LMs establish a new SOTA on the Pareto frontier of sampling speed and quality** for unconditional generation (Fig. 17 and Fig. 4).
>
> **Finding 3: Eso-LMs don't produce degenerate samples** (poor quality and low diversity) **at low NFEs** unlike the previous interpolating method BD3-LMs.

**Experimental Setup** We sample unconditionally ($L = 1024$) from OWT models. We train Eso-LMs with $\alpha_0^{\text{train}} \in \{0.125, 0.25, 0.5, 1\}$ and during sampling, vary both $\alpha_0^{\text{eval}}$ and the diffusion time discretization $T$ to control NFEs, using $(\alpha_0^{\text{eval}}, T) \in \{0.0625, 0.25, 0.5, 1\} \times \{16, 128, 1024\}$ with additional fine-grained $T$ values for $\alpha_0^{\text{eval}} = 1$. Each model is trained with a single $\alpha_0^{\text{train}}$ and evaluated across all $\alpha_0^{\text{eval}}$ values. MDLM and BD3-LMs use ancestral sampling as proposed in Sahoo et al. (2024a), with $T \in \{8, 16, 32, 64, 128, 256, 512, 1024, 4096\}$ for MDLM and $T \in \{128, 256, 512, 1024, 2048, 4096\}$ for BD3-LMs. BD3-LMs are evaluated with block sizes $L' \in \{4, 8, 16\}$ and $T' = T/(1024/L')$; $T = 128$ is not applicable to BD3-LM with $L' = 4$ and $T = 16$ is not applicable to all BD3-LMs considered, since these would result in $T' < 1$.

We measure Generative perplexity (Gen. PPL) via GPT-2 Large and MAUVE (Pillutla et al., 2021) via ModernBERT-Large for sample quality and average entropy for diversity (Zheng et al., 2024), using nucleus sampling with $p = 0.9$ (Wang et al., 2025a). Gen. PPL is the standard sample-quality metric in prior work and MAUVE is known to correlate with human judgments on open-ended text.

**Speed-Quality Pareto Frontier** We record the mean sampling duration across 10 trials by each method to generate a batch of 512 samples, and evaluate MAUVE and Gen. PPL using 5120 samples. In Fig. 4 and Fig. 17, for each method, we plot its speed-quality Pareto frontier over all its configurations: Eso-LMs (over $\alpha_0^{\text{train}}$, $\alpha_0^{\text{eval}}$, and $T$), BD3-LM (over $L'$ and $T$), and MDLM (over $T$). **Sampling consecutive tokens in parallel can yield conflicting tokens and degrade quality** (Liu et al., 2024), an effect that is especially pronounced for BD3-LMs with small blocks ($L \leq 16$; Sec. 5.2),

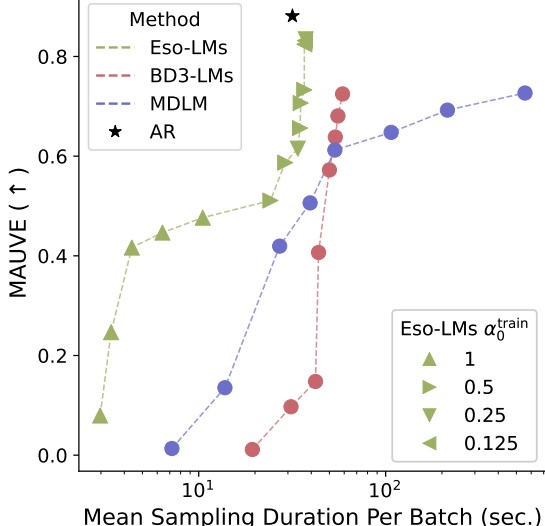

*Figure 4.* Eso-LMs establish SOTA on the Pareto frontier of sampling speed and sample quality (MAUVE; ↑).

causing a steeper quality drop as speed increases compared to other methods. **Overall, Eso-LMs set a new state of the art on the speed-quality Pareto frontier.** See Sec. E.12 for individual metric values and Sec. E.13 for text samples.

**Best $\alpha_0$ for Training** The Pareto frontier of Eso-LM trained for diffusion only ($\alpha_0^{\text{train}} = 1$) is competitive with the frontier obtained by the four Eso-LMs models trained at different $\alpha_0^{\text{train}}$ (Fig. 18 and Fig. 19). This shows Eso-LMs trained for diffusion only can flexibly adapt to a diverse set of denoising schedules.

> **Remark 2:** Under a compute budget, train Eso-LMs w/ $\alpha_0^{\text{train}} = 1$ and vary $\alpha_0^{\text{eval}}$ during sampling to trade-off speed and quality.

**Improved Block Sampler** BD3-LMs suffer a sharp quality drop at low NFEs due to parallel decoding of nearby tokens (Sec. 6). Exploiting the flexibility of our sampler, we propose a new sampler for Eso-LMs that improves upon the ancestral sampler in MDLM (Sec. 2.2), called the *block sampler*, that parallelizes decoding only for far-apart tokens (Sec. E.10). This substantially improves Eso-LMs' generation quality at low NFEs (Fig. 20 and Fig. 21).

### 5.3. Generation Latency at Long Context

> **Finding 4: At longer contexts, Eso-LMs are $3 - 4\times$ faster** than prior diffusion based methods that support partial KV caching and **$14 - 65\times$ faster** than MDMs that don't support KV caching.

**Experimental Setup** We compare sampling times of Eso-LMs against MDLM and BD3-LMs with context lengths $L \in$

$\{2048, 8192, 10240\}$, using the first-hitting sampler (Zheng et al., 2024) and batch size 1.To simulate a worst-case setting, we choose $T \gg L$ so that all methods perform roughly $L$ NFEs: $T = 10^6$ for MDLM and Eso-LMs (for $T \gg L$, NFE is $L$ for all $\alpha_0^{\text{eval}}$'s), $T' = 5000$ (sampling steps per block) for BD3-LMs. Nucleus sampling introduces a non-trivial overhead for all methods, so we disable it to isolate speed as a function of sequence length.

**Results**   As shown in Table 11, as compared to MDLM which lacks KV caching, Eso-LMs are ~14× faster for $L = 2048$, and ~65× faster for $L = 8192$. Compared to BD3-LMs, which partially support caching, Eso-LMs are ~3.2× faster than BD3-LM ($L' = 16$) and ~3.8× faster than BD3-LMs ($L' = 4$) at $L = 8192$. Additionally, we fine-tune Eso-LMs ($\alpha_0^{\text{train}} = 0.125$) and BD3-LMs ($L' = 4$), originally trained with $L = 1024$ (Sec. 5.1), for 1K steps with $L = 10240$ on OWT; as shown in Table 12, the Eso-LMs produces similar quality samples while being 5× faster ($\alpha_0^{\text{eval}} = 0.125$, $T \gg L$). These speedups arise from KV caching and the scheduler $\mathcal{S}$, which restricts the forward pass to masked tokens and previously denoised clean tokens, avoiding redundant computation. For the same NFE, Eso-LMs are slightly slower than AR models because KV reuse is only available from the penultimate step (Fig. 2).

# 6. Related Work, Discussion, and Conclusion

**MDM denoising architecture**   Prior work (Sahoo et al., 2024a; Shi et al., 2025) uses BERT-style, encoder-only transformers with bidirectional attention as MDM denoisers. In contrast, we use a decoder-only transformer with causal attention, as in AR models. However, instead of a strict left-to-right order, we use a random permutation of the input sequence (via the Any-Order AR view), which unlocks KV caching while retaining parallel generation during diffusion.

**Any-Order AR Models**   Uria et al. (2014) introduce Any-Order AR models, and Hoogeboom et al. (2021); Ou et al. (2025) connect them to MDMs while using encoder-only denoisers. **We instead advocate training MDMs with decoder-only denoisers**, which yields faster sampling (Sec. 5.2) and single-pass NELBO estimation (Sec. 3.3).

**Block diffusion**   BD3-LMs (Arriola et al., 2025) partition the context into token blocks, treat each block as an MDM, and generate blocks autoregressively, interpolating between AR and MDMs via block size. Eso-LMs instead interpolate by varying the fraction of tokens generated by diffusion, $\alpha_0$, over the full sequence. Sampling consecutive tokens in parallel can yield conflicting tokens and degraded quality (Liu et al., 2024), which is pronounced for BD3-LMs with small blocks ($L \leq 16$) (Sec. 5.2); Eso-LMs do not suffer from this.

**KV Caching**   KV caching behaves differently in AR models, BD3-LMs, and Eso-LMs. BD3-LMs cache only after fully denoising a block and do not cache intra-block diffusion steps. AR models cache keys and values for each token as soon as it is generated. In Eso-LMs, mask tokens are converted to clean tokens, so we cache their keys and values one denoising step later, when they first participate as clean tokens, causing a one-step lag in KV reuse (see Sec. 3.2).

**Concurrent work**   Hu et al. (2025); Wu et al. (2025); Ma et al. (2025) study approximate KV caching for MDLM. Hu et al. (2025) and Wu et al. (2025) focuses on block-wise sampling for MDLM with heuristics that allow KV reuse from generated blocks. However, each iteration still requires a forward pass over the full context, which includes all mask tokens. Ma et al. (2025) further supports random decoding orders. However, it frequently refreshes the KV cache for all tokens in the context. The methods above becomes highly restrictive at long context lengths. Xue et al. (2025) explores any-order generation for exact KV caching, but modify the transformer with adaptive LayerNorm to inject absolute positional embeddings for target positions, whereas Eso-LMs rely solely on attention masks and introduce no additional parameters. Pannatier et al. (2024); Xue et al. (2025) can be seen as special cases of Eso-LMs at $\alpha_0 = 1$, whereas Eso-LMs interpolate between AR and diffusion.

**Limitations**   Due to the use of doubled sequence length in sequential-phase training, Eso-LMs are about $1.37\times$ slower to train than MDLM when $\alpha_0 < 1$ (Sec. 4.1.2); however, since only half of each batch participates in sequential training, Eso-LMs train substantially faster than BD3-LMs. Also, the perplexity of Eso-LMs at $\alpha_0 = 1$ is worse than that of MDLM. We elaborate on this in Sec. 5.1: perplexity does not capture inference inefficiency and Eso-LMs achieve higher-quality samples than MDLM at every sampling-time budget (Sec. 5.2). Furthermore, KV reuse in Eso-LMs has a one-step lag, which causes Eso-LMs to be slightly slower than AR models under the same NFE (Sec. 5.3).

**Conclusion**   We introduce a new paradigm for language modeling that fuses AR models and MDMs, enabling seamless interpolation between the two in both generation speed and sample quality. Our method introduces KV caching in MDMs while preserving parallel generation, significantly accelerating inference. It outperforms block diffusion methods in both speed and accuracy, setting a new state of the art on language modeling benchmarks.

# Impact Statement

This paper presents work whose goal is to advance the field of Machine Learning. There are many potential societal consequences of our work, specifically those related to the generation of synthetic text. Our work can also be applied to the design of biological sequences, which carries both potential benefits and risks.

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

# Appendices

## A. Background

### A.1. KV Caching

Key-value (KV) caching (Pope et al., 2022) is a technique for efficient transformer inference that relies on causal attention (Vaswani et al., 2017), where the representation (i.e., keys and values) of token $\mathbf{x}^{\ell}$ depends only on that of previously generated tokens $\mathbf{x}^{<\ell}$. This causal dependency allows keys and values of past tokens to be computed once and reused across all subsequent decoding steps.

In contrast, transformers with bidirectional attention (e.g., Sahoo et al. (2024a)) allow each token to attend to both past and future positions within the sequence. As a result, the key and values of any token depend on the entire input sequence, including placeholder tokens that are not yet denoised at inference time. Consequently, when a new token is generated, the keys and values for all positions would change, invalidating previously computed keys and values and preventing their reuse. This lack of a causal dependency structure renders exact KV caching inapplicable to bidirectional transformers.

### A.2. BD3-LMs hyperparameter $T'$ and `num_tries`

In the original codebase of BD3-LMs (Arriola et al., 2025), the number of diffusion sampling steps $T'$ for each block is set to 5000. This is an extremely high $T'$ considering the fact that the number of tokens in each block $L'$ is at most 16. Having $L' \leq 16'$ and $T' = 5000$ means that off-the-shelf BD3-LMs are **not performing parallel generation** because tokens are almost always denoised one at a time.

Further, we found that BD3-LMs' codebase **cherry-picks its samples**. More specifically, to generate a single sample, the codebase keeps generating new samples (up to `num_tries` times) until one sample passes some quality-control test. By default, `num_tries` = 10 and the codebase reports sampling failure when the 10 tries are exhausted with no samples passing the test. Empirically, we found that sampling failures don't occur for $T' = 5000$.

To investigate the true performance of BD3-LMs for parallel generation, we set `num_tries` = 1, disable the quality-control test and evaluate samples from BD3-LMs across a wide range of $T$ values (Fig. 5). Here and in Fig. 5, $T$ means the sum of sampling steps across all blocks for BD3-LMs, e.g., $L' = 16$ and $T = 4096$ means that $T' = 4096/(1024/16)) = 64$ sampling steps is used per block. In contrast, BD3-LMs' codebase uses $T' = 5000$ by default, which corresponds to $T = \infty$ in Fig. 5. For MDLM, $T$ can be interpreted normally because it has no blocks.

As shown in Fig. 5, as $T$ is decreased to enable more parallel generation, **both sample quality and sample diversity of BD3-LMs becomes significantly worse than MDLM** which is discussed in Sec. 6. We also found that increasing `num_tries` can somewhat improve the sample entropy of BD3-LMs (second row of Table 3) and avoid degenerate samples, but doing so provides less or no improvements for AR and MDLM.

All five 1M-step checkpoints used in this section are publicly available Hugging Face checkpoints uploaded by BD3-LMs authors. In particular, their BD3-LM checkpoints are finetuned from MDLM.

*Table 3.* Gen. PPL (↓) and entropy (↑) (in parentheses) with nucleus sampling ($p = 0.9$) for AR, MDLM, and BD3-LM $L' = 16$ trained for 1M. We observe that the `num_tries` parameter introduced in (Arriola et al., 2025) for BD3-LMs selectively helps BD3-LMs but not the baselines. AR is not affected by $T$.

|  | BD3-LM $L' = 16$ | | MDLM | | AR | |
|---|---|---|---|---|---|---|
| num_tries | 1 | 10 | 1 | 10 | 1 | 10 |
| $T = 1024$ | 72.80 (5.35) | 77.71 (5.41) | 41.92 (5.36) | 41.79 (5.37) | 13.03 (5.26) | 13.76 (5.32) |
| $T = 256$ | 356.02 (5.11) | 440.69 (5.28) | 45.07 (5.40) | 44.57 (5.39) | 13.03 (5.26) | 13.76 (5.32) |

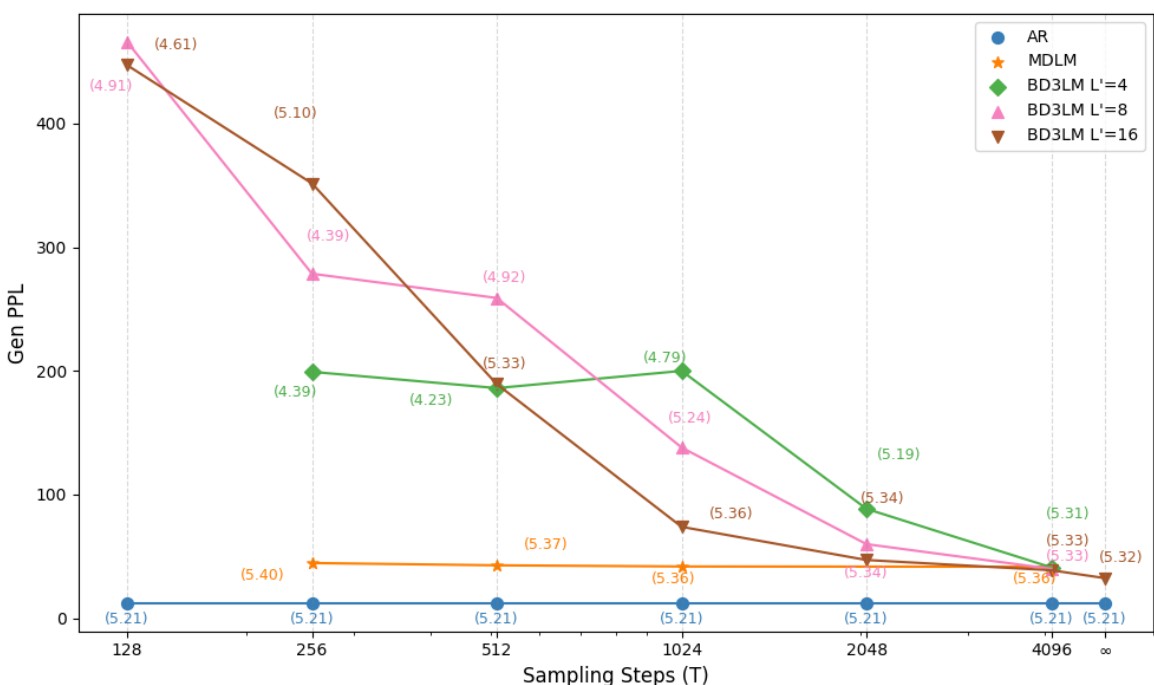

*Figure 5.* Gen. Perplexity (↓) with nucleus sampling ($p = 0.9$) against the number of sampling steps for AR, MDLM and BD3-LMs trained for 1M steps. The number of sampling steps for AR is always 1024; we extend it to other values for easier comparison. The number next to each data point records its sample entropy (↑); a value < 5 usually indicates low diversity degenerate samples.

# B. Esoteric Language Models

## B.1. NELBO

**Derivation of (7)** Let $\mathbf{x}$ denote the clean data and $\mathbf{z}_0$ be the latent that we wish to model using MDM with the conditional marginal $q(\mathbf{z}_t|\mathbf{x}) = \text{Cat}(\cdot \mid \alpha_t\mathbf{x} + (1 - \alpha_t)\mathbf{m})$ with $\alpha_t = \alpha_0(1 - t)$ and $t \in [0, 1]$. Let $\Delta = 1/T$ and $\mathbf{z}_0, \mathbf{z}_\Delta, \mathbf{z}_{2\Delta}, \dots, \mathbf{z}_1$ denote the MDM latents in discrete time.

$$\log p_\theta(\mathbf{x})$$
$$= \log \sum_{\mathbf{z}_{0:1}} p_\theta(\mathbf{x}, \mathbf{z}_0, \mathbf{z}_\Delta, \dots, \mathbf{z}_1)$$
$$= \log \sum_{\mathbf{z}_{0:1}} q(\mathbf{z}_0, \mathbf{z}_\Delta, \dots, \mathbf{z}_1|\mathbf{x}) \frac{p_\theta(\mathbf{x}, \mathbf{z}_0, \mathbf{z}_\Delta, \dots, \mathbf{z}_1)}{q(\mathbf{z}_0, \mathbf{z}_\Delta, \dots, \mathbf{z}_1|\mathbf{x})}$$
$$\geq \sum_{\mathbf{z}_{0:1}} q(\mathbf{z}_0, \mathbf{z}_\Delta, \dots, \mathbf{z}_1|\mathbf{x}) \log \frac{p_\theta(\mathbf{x}, \mathbf{z}_0, \mathbf{z}_\Delta, \dots, \mathbf{z}_1)}{q(\mathbf{z}_0, \mathbf{z}_\Delta, \dots, \mathbf{z}_1|\mathbf{x})}$$
$$= \sum_{\mathbf{z}_{0:1}} q(\mathbf{z}_0, \mathbf{z}_\Delta, \dots, \mathbf{z}_1|\mathbf{x}) \log \frac{p_\theta(\mathbf{x}|\mathbf{z}_0)p_\theta(\mathbf{z}_0|\mathbf{z}_\Delta)\dots p_\theta(\mathbf{z}_{1-\Delta}|\mathbf{z}_1)p_\theta(\mathbf{z}_1)}{q(\mathbf{z}_0|\mathbf{z}_\Delta, \mathbf{x})q(\mathbf{z}_\Delta|\mathbf{z}_{2\Delta}, \mathbf{x})\dots q(\mathbf{z}_{1-\Delta}|\mathbf{z}_1, \mathbf{x})q(\mathbf{z}_1|\mathbf{x})}$$
$$= \sum_{\mathbf{z}_{0:1}} q(\mathbf{z}_0, \mathbf{z}_\Delta, \dots, \mathbf{z}_1|\mathbf{x}) \left[ \log p_\theta(\mathbf{x}|\mathbf{z}_0) + \log \frac{p_\theta(\mathbf{z}_0|\mathbf{z}_\Delta)\dots p_\theta(\mathbf{z}_{1-\Delta}|\mathbf{z}_1)p_\theta(\mathbf{z}_1)}{q(\mathbf{z}_0|\mathbf{z}_\Delta, \mathbf{x})q(\mathbf{z}_\Delta|\mathbf{z}_{2\Delta}, \mathbf{x})\dots q(\mathbf{z}_{1-\Delta}|\mathbf{z}_1, \mathbf{x})q(\mathbf{z}_1|\mathbf{x})} \right]$$
$$= \sum_{\mathbf{z}_0} q(\mathbf{z}_0|\mathbf{x}) \log p_\theta(\mathbf{x}|\mathbf{z}_0) + \sum_{\mathbf{z}_{0:1}} q(\mathbf{z}_0, \mathbf{z}_\Delta, \dots, \mathbf{z}_1|\mathbf{x}) \log \frac{p_\theta(\mathbf{z}_0|\mathbf{z}_\Delta)\dots p_\theta(\mathbf{z}_{1-\Delta}|\mathbf{z}_1)p_\theta(\mathbf{z}_1)}{q(\mathbf{z}_0|\mathbf{z}_\Delta, \mathbf{x})q(\mathbf{z}_\Delta|\mathbf{z}_{2\Delta}, \mathbf{x})\dots q(\mathbf{z}_{1-\Delta}|\mathbf{z}_1, \mathbf{x})q(\mathbf{z}_1|\mathbf{x})}$$
$$= \sum_{\mathbf{z}_0} q(\mathbf{z}_0|\mathbf{x}) \log p_\theta(\mathbf{x}|\mathbf{z}_0) + \sum_{\mathbf{z}_0, \mathbf{z}_\Delta} q(\mathbf{z}_0, \mathbf{z}_\Delta|\mathbf{x}) \log \frac{p_\theta(\mathbf{z}_0|\mathbf{z}_\Delta)}{q(\mathbf{z}_0|\mathbf{z}_\Delta, \mathbf{x})}$$
$$+ \sum_{\mathbf{z}_\Delta, \mathbf{z}_{2\Delta}} q(\mathbf{z}_\Delta, \mathbf{z}_{2\Delta}|\mathbf{x}) \log \frac{p_\theta(\mathbf{z}_\Delta|\mathbf{z}_{2\Delta})}{q(\mathbf{z}_\Delta|\mathbf{z}_{2\Delta}, \mathbf{x})} + \dots + \sum_{\mathbf{z}_1} q(\mathbf{z}_1|\mathbf{x}) \log \frac{p_\theta(\mathbf{z}_1)}{q(\mathbf{z}_1|\mathbf{x})}$$
$$= \sum_{\mathbf{z}_0} q(\mathbf{z}_0|\mathbf{x}) \log p_\theta(\mathbf{x}|\mathbf{z}_0) + \sum_{\mathbf{z}_0, \mathbf{z}_\Delta} q(\mathbf{z}_\Delta|\mathbf{x})q(\mathbf{z}_0|\mathbf{z}_\Delta, \mathbf{x}) \log \frac{p_\theta(\mathbf{z}_0|\mathbf{z}_\Delta)}{q(\mathbf{z}_0|\mathbf{z}_\Delta, \mathbf{x})}$$
$$+ \sum_{\mathbf{z}_\Delta, \mathbf{z}_{2\Delta}} q(\mathbf{z}_{2\Delta}|\mathbf{x})q(\mathbf{z}_\Delta|\mathbf{z}_{2\Delta}, \mathbf{x}) \log \frac{p_\theta(\mathbf{z}_\Delta|\mathbf{z}_{2\Delta})}{q(\mathbf{z}_\Delta|\mathbf{z}_{2\Delta}, \mathbf{x})} + \dots + \sum_{\mathbf{z}_1} q(\mathbf{z}_1|\mathbf{x}) \log \frac{p_\theta(\mathbf{z}_1)}{q(\mathbf{z}_1|\mathbf{x})}$$
$$= \sum_{\mathbf{z}_0} q(\mathbf{z}_0|\mathbf{x}) \log p_\theta(\mathbf{x}|\mathbf{z}_0) - \sum_{\mathbf{z}_\Delta} q(\mathbf{z}_\Delta|\mathbf{x})\text{D}_{\text{KL}}(q(\mathbf{z}_0|\mathbf{z}_\Delta, \mathbf{x})\|p_\theta(\mathbf{z}_0|\mathbf{z}_\Delta))$$
$$- \sum_{\mathbf{z}_{2\Delta}} q(\mathbf{z}_{2\Delta}|\mathbf{x})\text{D}_{\text{KL}}(q(\mathbf{z}_\Delta|\mathbf{z}_{2\Delta}, \mathbf{x})\|p_\theta(\mathbf{z}_\Delta|\mathbf{z}_{2\Delta})) - \dots - \text{D}_{\text{KL}}(q(\mathbf{z}_1|\mathbf{x})\|p_\theta(\mathbf{z}_1))$$
$$= \sum_{\mathbf{z}_0} q(\mathbf{z}_0|\mathbf{x}) \log p_\theta(\mathbf{x}|\mathbf{z}_0) - \mathbb{E}_{\mathbf{z}_\Delta}\text{D}_{\text{KL}}(q(\mathbf{z}_0|\mathbf{z}_\Delta, \mathbf{x})\|p_\theta(\mathbf{z}_0|\mathbf{z}_\Delta))$$
$$- \mathbb{E}_{\mathbf{z}_{2\Delta}}\text{D}_{\text{KL}}(q(\mathbf{z}_\Delta|\mathbf{z}_{2\Delta}, \mathbf{x})\|p_\theta(\mathbf{z}_\Delta|\mathbf{z}_{2\Delta})) - \dots - \text{D}_{\text{KL}}(q(\mathbf{z}_1|\mathbf{x})\|p_\theta(\mathbf{z}_1))$$
$$= \sum_{\mathbf{z}_0} q(\mathbf{z}_0|\mathbf{x}) \log p_\theta(\mathbf{x}|\mathbf{z}_0) - \sum_{t=\Delta}^{1} \mathbb{E}_{\mathbf{z}_t}\text{D}_{\text{KL}}(q(\mathbf{z}_{t-\Delta}|\mathbf{z}_t, \mathbf{x})\|p_\theta(\mathbf{z}_{t-\Delta}|\mathbf{z}_t)) - \text{D}_{\text{KL}}(q(\mathbf{z}_1|\mathbf{x})\|p_\theta(\mathbf{z}_1))$$
$$= \sum_{\mathbf{z}_0} q(\mathbf{z}_0|\mathbf{x}) \log p_\theta(\mathbf{x}|\mathbf{z}_0) - \sum_{t=\Delta}^{1} \mathbb{E}_{\mathbf{z}_t}\text{D}_{\text{KL}}(q(\mathbf{z}_{t-\Delta}|\mathbf{z}_t, \mathbf{x})\|p_\theta(\mathbf{z}_{t-\Delta}|\mathbf{z}_t))$$
$$= \sum_{\mathbf{z}_0} q(\mathbf{z}_0|\mathbf{x}) \log p_\theta(\mathbf{x}|\mathbf{z}_0) - \mathbb{E}_{t\sim\mathcal{U}[\Delta,1]}\mathbb{E}_{\mathbf{z}_t}\text{D}_{\text{KL}}(q(\mathbf{z}_{t-\Delta}|\mathbf{z}_t, \mathbf{x})\|p_\theta(\mathbf{z}_{t-\Delta}|\mathbf{z}_t))\frac{1}{\Delta} \tag{10}$$

For Eso-LMs, the true and learned reverse posteriors are (2) and (3) respectively with $\alpha_t = \alpha_0(1 - t)$. Sahoo et al. (2024a) shows that (10) simplifies the following as $\Delta \to 0$ (continuous-time):

$$\log p_\theta(\mathbf{x}) \geq \mathbb{E}_{\mathbf{z}_0\sim q_0} \log p_\theta(\mathbf{x}|\mathbf{z}_0) - \mathbb{E}_{t\sim\mathcal{U}[0,1],\mathbf{z}_t\sim q_t} \left[ \frac{\alpha'_t}{1-\alpha_t} \sum_{\ell\in\mathcal{M}(\mathbf{z}_t)} \log\langle\mathbf{x}_\theta^\ell(\mathbf{z}_t), \mathbf{x}^\ell\rangle \right]. \tag{11}$$

## B.2. Importance Weighted Bounds for Masked Diffusion Models

**Theorem B.1.** *(Copy of Theorem 3.1) The IW bound $\mathcal{L}_{AO}^K$ holds for all $K$, monotoniically decreases as $K$ increases, and converges to $-\log p_\theta(\mathbf{x})$ as $K \to \infty$. This result generalizes (4, $\mathcal{L}_{AO}$) by Ou et al. (2025).*

$$-\log p_\theta(\mathbf{x}) \le \mathcal{L}_{AO}^K \triangleq -\mathbb{E}_{\sigma_1,\ldots,\sigma_K \sim \mathcal{P}_L} \left[ \log \frac{1}{K} + \log \sum_{k=1}^K \exp\left( \sum_{l=1}^L \log p_\theta(\mathbf{x}^{\sigma_k(l)} \mid \mathbf{x}^{\sigma_k(<l)}) \right) \right] \le \mathcal{L}_{MDM}. \tag{12}$$

*Proof.* (First inequality) Treating permutation $\sigma$ as a latent variable (Shih et al., 2022; Hoogeboom et al., 2021), one can derive the following NELBO:

$$-\log p(\mathbf{x}) \tag{13}$$

$$= -\log \mathbb{E}_{\sigma \sim \mathcal{P}_L} [p_\theta(\mathbf{x} \mid \sigma)] \tag{14}$$

$$= -\log \mathbb{E}_{\sigma \sim \mathcal{P}_L} \left[ \prod_{l=1}^L p_\theta(\mathbf{x}^{\sigma(l)} \mid \mathbf{x}^{\sigma(<l)}) \right] \tag{15}$$

$$= -\log \mathbb{E}_{\sigma_1,\ldots,\sigma_K \sim \mathcal{P}_L} \left[ \frac{1}{K} \sum_{k=1}^K \prod_{l=1}^L p_\theta(\mathbf{x}^{\sigma_k(l)} \mid \mathbf{x}^{\sigma_k(<l)}) \right] \tag{16}$$

$$\le -\mathbb{E}_{\sigma_1,\ldots,\sigma_K \sim \mathcal{P}_L} \left[ \log \frac{1}{K} \sum_{k=1}^K w_k \right] \triangleq \mathcal{L}_{AO}^K(\mathbf{x}), \tag{17}$$

where $w_k = \prod_{l=1}^L p_\theta(\mathbf{x}^{\sigma_k(l)} \mid \mathbf{x}^{\sigma_k(<l)})$. Since $w_k$ is clearly bounded (by 0 and 1), one can invoke Theorem 1 in Burda et al. (2015) to establish two key properties for $\mathcal{L}_{AO}^K$: (1) $\mathcal{L}_{AO}^k \le \mathcal{L}_{AO}^m$ for $k \ge m$ and (2) $-\log p(\mathbf{x}) = \lim_{K \to \infty} \mathcal{L}_{AO}^K$. Finally, by simple algebra, one can simplify (17) into

$$-\mathbb{E}_{\sigma_1,\ldots,\sigma_K \sim \mathcal{P}_L} \left[ \log \frac{1}{K} + \log \sum_{k=1}^K \exp\left( \sum_{l=1}^L \log p_\theta(\mathbf{x}^{\sigma_k(l)} \mid \mathbf{x}^{\sigma_k(<l)}) \right) \right]. \tag{18}$$

(Second inequality) When $K = 1$, $\mathcal{L}_{AO}^K$ reduces to $\mathcal{L}_{AO}$ in (4), which is equal to $\mathcal{L}_{MDM}$. $\square$

## B.3. Importance Weighted Bounds for Esoteric Language Models

Let $\mathbf{x}$ denote the clean data. Let $\mathbf{z}_0 \sim q_0(\mathbf{z}_0 \mid \mathbf{x})$ be the latent that we wish to model using MDM with the conditional marginal $q(\mathbf{z}_t|\mathbf{z}_0) = \text{Cat}(\cdot \mid \alpha_t \mathbf{z}_0 + (1 - \alpha_t)\mathbf{m})$ with $\alpha_t = 1 - t$ and $t \in [0, 1]$. Note that (1) the condition is now $\mathbf{z}_0$ rather than $\mathbf{x}$ and (2) $\alpha_t$ here has endpoints at $\alpha_{t=0} = 1$ and $\alpha_{t=1} = 0$.

Let $\Delta = 1/T$ and $\mathbf{z}_0, \mathbf{z}_\Delta, \mathbf{z}_{2\Delta}, \ldots, \mathbf{z}_1$ denote the MDM latents in discrete time. Let $\mathcal{C}$ denote the ordered set of indices (from small to large) of clean tokens in $\mathbf{z}_0$; whether each index is clean in $\mathbf{z}_0$ independently follows Bernoulli($\alpha_0$). Let $f$ be the bijection such that $\mathbf{z}_0 = f(\mathbf{x}, \mathcal{C})$ and vice versa.

**Lemma B.2.** *The RHS of the following inequality is an alternative NELBO for Eso-LMs as $\Delta \to 0$:*

$$\log p_\theta(\mathbf{x}) \ge \mathbb{E}_{\mathbf{z}_0 \sim q_0} \log p_\theta(\mathbf{x}|\mathbf{z}_0) - \mathbb{E}_{\mathcal{C} \sim q(\mathcal{C}), t \sim \mathcal{U}[0,1], \mathbf{z}_t \sim q_t(\cdot|\mathbf{z}_0 = f(\mathbf{x}, \mathcal{C}))} \left[ \frac{\alpha_t'}{1 - \alpha_t} \sum_{\ell \in \mathcal{M}(\mathbf{z}_t) \cap \mathcal{C}} \log \langle \mathbf{x}_\theta^\ell(\mathbf{z}_t), \mathbf{x}^\ell \rangle \right]. \tag{19}$$

*Proof.* By introducing $\mathcal{C}$ and mirroring the steps in the derivation of (11), we obtain

$$\log p_\theta(\mathbf{x})$$

$$= \log \sum_{\mathbf{z}_{0:1},\mathcal{C}} p_\theta(\mathbf{x}, \mathbf{z}_0, \mathbf{z}_\Delta, \dots, \mathbf{z}_1, \mathcal{C})$$

$$= \log \sum_{\mathbf{z}_{0:1},\mathcal{C}} q(\mathbf{z}_0, \mathbf{z}_\Delta, \dots, \mathbf{z}_1, \mathcal{C}|\mathbf{x}) \frac{p_\theta(\mathbf{x}, \mathbf{z}_0, \mathbf{z}_\Delta, \dots, \mathbf{z}_1, \mathcal{C})}{q(\mathbf{z}_0, \mathbf{z}_\Delta, \dots, \mathbf{z}_1, \mathcal{C}|\mathbf{x})}$$

Applying Jensen's inequality,

$$\geq \sum_{\mathbf{z}_{0:1},\mathcal{C}} q(\mathbf{z}_0, \mathbf{z}_\Delta, \dots, \mathbf{z}_1, \mathcal{C}|\mathbf{x}) \log \frac{p_\theta(\mathbf{x}, \mathbf{z}_0, \mathbf{z}_\Delta, \dots, \mathbf{z}_1, \mathcal{C})}{q(\mathbf{z}_0, \mathbf{z}_\Delta, \dots, \mathbf{z}_1, \mathcal{C}|\mathbf{x})}$$

Factorizing the joint distribution,

$$= \sum_{\mathbf{z}_{0:1},\mathcal{C}} q(\mathbf{z}_0, \mathbf{z}_\Delta, \dots, \mathbf{z}_1, \mathcal{C}|\mathbf{x}) \log \frac{p_\theta(\mathbf{x}|\mathbf{z}_0) p_\theta(\mathbf{z}_0|\mathbf{z}_\Delta, \mathcal{C}) \dots p_\theta(\mathbf{z}_{1-\Delta}|\mathbf{z}_1, \mathcal{C}) p_\theta(\mathbf{z}_1) p_\theta(\mathcal{C})}{q(\mathbf{z}_0|\mathbf{z}_\Delta, \mathbf{x}, \mathcal{C}) q(\mathbf{z}_\Delta|\mathbf{z}_{2\Delta}, \mathbf{x}, \mathcal{C}) \dots q(\mathbf{z}_{1-\Delta}|\mathbf{z}_1, \mathbf{x}, \mathcal{C}) q(\mathbf{z}_1|\mathbf{x}, \mathcal{C}) q(\mathcal{C})}$$

$$= \sum_{\mathbf{z}_{0:1},\mathcal{C}} q(\mathbf{z}_0, \mathbf{z}_\Delta, \dots, \mathbf{z}_1, \mathcal{C}|\mathbf{x}) \log \frac{p_\theta(\mathbf{x}|\mathbf{z}_0) p_\theta(\mathbf{z}_0|\mathbf{z}_\Delta, \mathcal{C}) \dots p_\theta(\mathbf{z}_{1-\Delta}|\mathbf{z}_1, \mathcal{C}) p_\theta(\mathbf{z}_1)}{q(\mathbf{z}_0|\mathbf{z}_\Delta, \mathbf{z}_0) q(\mathbf{z}_\Delta|\mathbf{z}_{2\Delta}, \mathbf{z}_0) \dots q(\mathbf{z}_{1-\Delta}|\mathbf{z}_1, \mathbf{z}_0) q(\mathbf{z}_1|\mathbf{z}_0)}$$

$$\vdots$$

$$= \sum_{\mathbf{z}_0} q(\mathbf{z}_0|\mathbf{x}) \log p_\theta(\mathbf{x}|\mathbf{z}_0) - \mathbb{E}_{\mathcal{C}\sim q(\mathcal{C}), t\sim\mathcal{U}[0,1], \mathbf{z}_t\sim q_t(\cdot|\mathbf{z}_0=f(\mathbf{x},\mathcal{C}))} D_{\text{KL}}(q(\mathbf{z}_{t-\Delta}|\mathbf{z}_t, \mathbf{z}_0) \| p_\theta(\mathbf{z}_{t-\Delta}|\mathbf{z}_t, \mathcal{C})) \frac{1}{\Delta} \quad (20)$$

The true posterior is (2) with $\mathbf{x}$ replaced by $\mathbf{z}_0$. The learned posterior is (3) with the following modification: similar to Carry-Over Unmasking in (Sahoo et al., 2024a), we can substitute the output of $\mathbf{x}_\theta$ to simply copy masked inputs outside $\mathcal{C}$; this allows us to ignore the loss over positions outside $\mathcal{C}$. Finally, Sahoo et al. (2024a) shows that (20) simplifies the following as $\Delta \to 0$ (continuous-time):

$$\log p_\theta(\mathbf{x}) \geq \mathbb{E}_{\mathbf{z}_0\sim q_0} \log p_\theta(\mathbf{x}|\mathbf{z}_0) - \mathbb{E}_{\mathcal{C}\sim q(\mathcal{C}), t\sim\mathcal{U}[0,1], \mathbf{z}_t\sim q_t(\cdot|\mathbf{z}_0=f(\mathbf{x},\mathcal{C}))} \left[ \frac{\alpha'_t}{1-\alpha_t} \sum_{\ell\in\mathcal{M}(\mathbf{z}_t)\cap\mathcal{C}} \log\langle \mathbf{x}_\theta^\ell(\mathbf{z}_t), \mathbf{z}_0^\ell \rangle \right].$$

$$\square$$

**Lemma B.3.** *Let $\mathcal{P}(\mathcal{C})$ denote the set of all permutations of $\mathcal{C}$ and let $\mathcal{C}'$ denote the ordered complement of $\mathcal{C}$. Then, the RHS of (19) is equivalent to the following expression:*

$$\mathbb{E}_{\sigma\sim\mathcal{P}_L^{\alpha_0}} \left[ \sum_{\ell=1}^L \log p_\theta(\mathbf{x}^{\sigma(l)} \mid \mathbf{x}^{\sigma(<l)}) \right], \quad (21)$$

*where $\mathcal{P}_L^{\alpha_0} \triangleq \{\sigma \cup \mathcal{C}' : \mathcal{C} \sim q(\mathcal{C}), \sigma \sim \mathcal{P}(\mathcal{C})\}$.*

*Proof.* Applying to $(4, \mathcal{L}_{\text{AO}})$ to (19), we obtain

$$\mathbb{E}_{\mathcal{C}\sim q(\mathcal{C})} \left[ \sum_{\ell\in\mathcal{C}'} \log p_\theta(\mathbf{x}^\ell \mid \mathbf{z}_0, \mathbf{x}^{<\ell}) \right] + \mathbb{E}_{\mathcal{C}\sim q(\mathcal{C}), \sigma\sim\mathcal{P}(\mathcal{C})} \left[ \sum_{\ell=1}^L \log p_\theta(\mathbf{x}^{\sigma(l)} \mid \mathbf{x}^{\sigma(<l)}) \right]$$

$$= \mathbb{E}_{\mathcal{C}\sim q(\mathcal{C})} \left[ \sum_{\ell\in\mathcal{C}'} \log p_\theta(\mathbf{x}^\ell \mid \mathbf{z}_0, \mathbf{x}^{<\ell}) + \mathbb{E}_{\sigma\sim\mathcal{P}(\mathcal{C})} \left[ \sum_{\ell=1}^L \log p_\theta(\mathbf{x}^{\sigma(l)} \mid \mathbf{x}^{\sigma(<l)}) \right] \right]$$

$$= \mathbb{E}_{\sigma\sim\mathcal{P}_L^{\alpha_0}} \left[ \sum_{\ell=1}^L \log p_\theta(\mathbf{x}^{\sigma(l)} \mid \mathbf{x}^{\sigma(<l)}) \right].$$

$$\square$$

**Theorem B.4.** *The IW bound for Eso-LMs with $\alpha_0 < 1$ holds for all $K$, monotonically decreases as $K$ increases, and converges to $-\log p_\theta(\mathbf{x})$ as $K \to \infty$.*

$$-\log p_\theta(\mathbf{x}) \leq -\mathbb{E}_{\sigma_1,\dots,\sigma_K\sim\mathcal{P}_L^{\alpha_0}} \left[ \log\frac{1}{K} + \log\sum_{k=1}^K \exp\left( \sum_{l=1}^L \log p_\theta(\mathbf{x}^{\sigma_k(l)} \mid \mathbf{x}^{\sigma_k(<l)}) \right) \right]. \quad (22)$$

*Proof.* Proof closely parallels Theorem 3.1. $\square$

## B.4. Training Algorithm

Algo. 1 outlines the complete training procedure.

---

**Algorithm 1** Eso-LMs Training

---

**Input:** dataset $D$, batch size $\texttt{bs}$, forward noise process $q_t(\cdot|\mathbf{x})$, model $\mathbf{x}_\theta$, learning rate $\eta$
**while** not converged **do**
    $\mathbf{x}_1, \mathbf{x}_2, \ldots, \mathbf{x}_{\texttt{bs}} \sim D$
    **for** $i \leftarrow 1$ to $\texttt{bs}/2$ **do**                                      *(If $\alpha_0 = 1$, loop through $1$ to $\texttt{bs}$)*
        $\mathbf{z}_0 \sim q_0(\cdot|\mathbf{x}_i)$
        $\sigma \sim \mathcal{P}_L$ with constraints               *(Used to construct attention bias $A$ in $\mathbf{x}_\theta$; Sec. 4)*
        $\mathcal{L}_i \leftarrow -\sum_{\ell \in \mathcal{M}(\mathbf{z}_0)} \log\langle \mathbf{x}_\theta^\ell(\mathbf{z}_0, \mathbf{x}_i^{<\ell}), \mathbf{x}_i^\ell \rangle$             *(Sequential loss estimator in (7))*
    **end for**
    **for** $i \leftarrow \texttt{bs}/2 + 1$ to $\texttt{bs}$ **do**                                      *(If $\alpha_0 = 1$, skip this loop)*
        Sample $t \sim \mathcal{U}[0, 1]$
        $\mathbf{z}_t \sim q_t(\cdot|\mathbf{x}_i)$
        $\sigma \sim \mathcal{P}_L$ with constraints               *(Used to construct attention bias $A$ in $\mathbf{x}_\theta$; Sec. 4)*
        $\mathcal{L}_i \leftarrow \frac{\alpha_t'}{1-\alpha_t} \sum_{\ell \in \mathcal{M}(\mathbf{z}_t)} \log\langle \mathbf{x}_\theta^\ell(\mathbf{z}_t), \mathbf{x}_i^\ell \rangle$             *(MDM loss estimator in (7))*
    **end for**
    $\theta \leftarrow \theta - \eta \nabla_\theta \sum_{i=1}^{\texttt{bs}} \mathcal{L}_i$
  **end while**

---

## B.5. Denoising Schedule and Sampling Algorithm

**Pre-computing the unified denoising schedule**   Eso-LMs perform two phases of sampling: the diffusion phase and the sequential phase. Within the diffusion phase, tokens are denoised in random order and potentially in parallel. Within the sequential phase, remaining mask tokens are denoised sequentially from left to right and one at a time.

First, to determine (i) the total number of tokens to denoise during the diffusion phase and (ii) the number of tokens to denoise per diffusion step, we run a modified version of the first-hitting algorithm proposed in (Zheng et al., 2024). Suppose the sequence to generate has length $L$, the number of discretization steps is $T$, and the noise schedule is $\alpha$ (with $\alpha_0 \geq 0$). Let $dt = 1/T$. We iterate from $t = 1$ to $1 - dt$ (inclusive) for $T$ steps. For each step, we compute the number of tokens to denoise at time $t$ as

$$n_t = \text{Binom}\left(n = n_t^{\text{mask}}, p = \frac{\alpha_s - \alpha_t}{1 - \alpha_t}\right), \tag{23}$$

where $s = t - dt$ and $n_t^{\text{mask}} = L - \sum_{t'>t} n_{t'}$. When $T$ is large, some $n_t$'s could be zero. All the $n_t$'s produced by this algorithm are collected in an ordered list, except for the $n_t$'s that are zeros. We denote the sum of all $n_t$'s as $n^{\text{MDM}}$ and define $n^{\text{AR}} = L - n^{\text{MDM}}$. We select $n^{\text{MDM}}$ token indices from $[L]$ to denoise by diffusion and use the complementing subset of token indices to denoise sequentially.

---

**Algorithm 2** Pre-computing the Unified Denoising Schedule for Eso-LMs

---

**Input:** sequence length $L$, expected fraction of tokens by diffusion $\alpha_0$, diffusion steps $T$
$\mathcal{S}^{\text{MDM}} \leftarrow (), \mathcal{S}^{\text{AR}} \leftarrow (), \Delta \leftarrow 1/T$
$\mathcal{M} = \{1, \ldots, L\}$          *(Set of all mask tokens)*
// Diffusion Denoising Schedule
**for** $t \in [1, 1 - \Delta, \ldots, \Delta]$ **do**
    $\alpha_t \leftarrow \alpha_0(1 - t)$
    $n_t \sim \text{Binomial}\left(n = |\mathcal{M}|, p = \frac{\alpha_0 \Delta}{1 - \alpha_t}\right)$        *(See (5))*
    $S_t \leftarrow \text{SampleWithoutReplace}(\mathcal{M}, n_t)$
    $\mathcal{S}^{\text{MDM}} \leftarrow \mathcal{S}^{\text{MDM}} \cup (S_t)$
    $\mathcal{M} \leftarrow \mathcal{M} - S_t$
**end for**
// Autoregressive Denoising Schedule
**for** $i \in \mathcal{M}$ **do**
    $\mathcal{S}^{\text{AR}} \leftarrow \mathcal{S}^{\text{AR}} \cup ((i))$
**end for** **return** $\mathcal{S}^{\text{MDM}} \cup \mathcal{S}^{\text{AR}}$

---

**Sampling**    Given a unified denoising schedule, we sample from the model using Alg. 3.

---

**Algorithm 3** Eso-LMs Sampling

---

**Input:** sequence length $L$, unified sampling schedule $\mathcal{S}$
$\mathbf{z} = [\text{MASK\_INDEX}, \ldots, \text{MASK\_INDEX}]$
$C = \{\}$          *(Indices of clean tokens)*
**for** $i \leftarrow 1$ to $|\mathcal{S}|$ **do**      *(Sequential happens automatically when $|C| \geq n^{\text{MDM}}$)*
    `logits` $\leftarrow \mathbf{x}_\theta(\mathbf{z}[C \cup S_i])$        *(See Remark)*
    `logits` $\leftarrow$ select `logits` corresponding to $S_i$
    $\mathbf{z}[\mathcal{S}_i] \leftarrow$ `categorical_sample(logits, dim=-1)`      *(`logits` has shape $(|S_i|, |\mathcal{V}|)$)*
    $C \leftarrow C \cup S_i$
**end for**
**Return:** $\mathbf{z}$

*Remark.* $\mathbf{z}[C \cup S_i]$ denotes the subset of tokens in $\mathbf{z}$ fed into the denoising model $\mathbf{x}_\theta$. The position embeddings for a token $\mathbf{z}^\ell \in \mathbf{z}[C \cup S_i]$ are ensured to be the same as in the original sequence $\mathbf{z}$. Refer to Sec. D.3 and Sec. 4.2 for computing the sampling attention bias $A$ for Eso-LMs and Eso-LMs (A) respectively. For Eso-LMs, due to causal attention, $\mathbf{x}_\theta$ can cache KV-values of a clean token upon first processing.

---

**Concrete example**    Suppose $L = 8$ and the token indices are $[1, 2, \ldots, 8]$. Suppose we obtained $n^{\text{MDM}} = 5$ from the algorithm above. Then, the diffusion indices we may select are $(1, 3, 4, 6, 7)$ and the complementing sequential indices are $(2, 5, 8)$. We further randomly permute the diffusion indices to be, e.g., $(3, 1, 6, 4, 7)$, for random-order denoising.

Given the list of non-zero $n_t$'s and the permuted ordered set of diffusion indices, we create the sampling schedule for diffusion by partitioning the diffusion indices per the $n_t$'s. Suppose the list of non-zero $n_t$'s is $(2, 1, 2)$. Using it to partition the permuted set of diffusion indices $(3, 1, 6, 4, 7)$, we obtain the following sampling schedule for the diffusion phase: $\mathcal{S}^{\text{MDM}} = ((3, 1), (6), (4, 7))$. The denoising schedule for the sequential phase is simply $\mathcal{S}^{\text{AR}} = ((2), (5), (8))$. The unified sampling schedule $\mathcal{S}$ is the concatenation of $\mathcal{S}^{\text{MDM}}$ and $\mathcal{S}^{\text{AR}}$. In this example, $\mathcal{S} = (S_1, S_2, S_3, S_4, S_5, S_6)$ where $S_1 = (3, 1), S_2 = (6), S_3 = (4, 7), S_4 = (2), S_5 = (5)$ and $S_6 = (8)$. This corresponds to 6 NFEs. Finally, $\mathcal{S}$ is passed to Algo. 3, which handles the rest of the sampling procedure. Connecting back to the denoising ordering $\sigma$ discussed in Sec. D.3 and Sec. 4.2, we have $\sigma = (3, 1, 6, 4, 7, 2, 5, 8)$ in this example.

## B.6. Attention Mechanism for Diffusion Phase Training

For a short and intuitive description, refer to Sec. 4.1.1.

In the diffusion phase, the denoising transformer receives $\mathbf{z}_t \sim q_t(.|\mathbf{x})$ as input, which contains mask tokens to denoise, and

**x** as target. We leverage the connection of MDMs with AO-ARMs (Ou et al., 2025), which establishes that mask tokens $\{\mathbf{z}_t^i | i \in \mathcal{M}(\mathbf{z}_t)\}$ can be denoised in any random order, and clean tokens $\{\mathbf{z}_t^i | i \in \mathcal{C}(\mathbf{z}_t)\}$ also could have been generated in any random order. Hence, we first sample a random ordering $\sigma \sim \mathcal{P}_L$ with the only constraint that clean tokens in $\mathbf{z}_t$ precede mask tokens in $\mathbf{z}_t$ per $\sigma$. We then constrain a clean token $(\mathbf{z}_t^i)_{i \in \mathcal{C}(\mathbf{z}_t)}$ to only attend to itself and prior clean tokens per $\sigma$; a mask token $(\mathbf{z}_t^i)_{i \in \mathcal{M}(\mathbf{z}_t)}$ attends to clean tokens, itself, and prior mask tokens per $\sigma$. Hence we define the $L \times L$ attention bias by

$$A_{i,j} = \begin{cases} 0 & \text{if } \sigma^{-1}(i) \geq \sigma^{-1}(j) \ \forall (i,j) \in [L] \times [L] \tag{24} \\ -\infty & \text{otherwise.} \tag{25} \end{cases}$$

See Fig. 6 for an example.

**Simplified Implementation**    $A$ becomes a causal attention bias if we sort the rows and columns of $A$ by $\sigma$ (Fig. 6), which is simple to implement. We also sort the positional embeddings of $\mathbf{z}_t$ by $\sigma$ so tokens keep their original positional embeddings. When calculating loss, we sort the target **x** by $\sigma$.

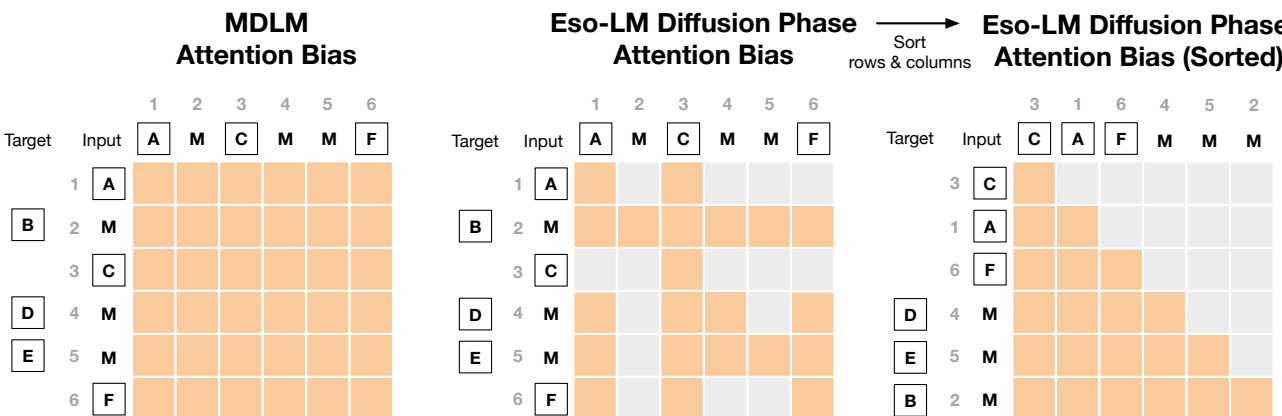

*Figure 6.* Comparison of attention biases for MDLM and Eso-LMs diffusion-phase training, before and after sorting the rows and columns by $\sigma$. **Orange** represents 0 (attention) and gray represents $-\infty$ (no attention). The clean sequence is $\mathbf{x} = (A, B, C, D, E, F)$ and hence $L = 6$. After random masking, we obtain $\mathbf{z}_t = (A, M, C, M, M, F)$. The integers denote position indices: $\mathcal{M}(\mathbf{z}_t) = \{2, 4, 5\}$ and $\mathcal{C}(\mathbf{z}_t) = \{1, 3, 6\}$. The ordering is $\sigma = (3, 1, 6, 4, 5, 2) \sim \mathcal{P}_6$ with clean tokens before mask tokens.

```
from torch.nn.attention.flex_attention import create_block_mask

def _causal_mask(b, h, q_idx, kv_idx):
  causal = q_idx >= kv_idx
  return causal

def _get_causal_mask(seq_len):
  return create_block_mask(
    _causal_mask,
    B=None, H=None, Q_LEN=seq_len, KV_LEN=seq_len)
```

*Figure 7.* We implement the attention bias from Fig. 6 (Right) as a FlexAttention-compatible sparse masking function shown above that can handle arbitrary sequence lengths. This enables Just-In-Time compilation that's significantly faster and more memory efficient than `scaled_dot_product_attention` in PyTorch.

## B.7. Attention Mechanism for Sequential Phase Training

The denoising transformer receives the concatenated sequence $\mathbf{z}_0 \oplus \mathbf{x} \in \mathcal{V}^{2L}$ as input, where $\mathbf{z}_0 \sim q_0(.|\mathbf{x})$ contains the mask tokens to denoise, and **x** as target. Given $\mathbf{z}_0$, we sample a permutation $\sigma \sim \mathcal{P}_L$ such that (i) clean tokens precede mask tokens, and (ii) mask tokens remain in natural order. To enforce the correct information flow, we define the $2L \times 2L$ attention

bias by

$$
\begin{aligned}
A_{i,j} &= 0 & &\text{if } i = j \ \forall (i,j) \in \mathcal{M}(\mathbf{z}_0) \times \mathcal{M}(\mathbf{z}_0) & &(26) \\
A_{i,j+L} &= 0 & &\forall (i,j) \in \mathcal{M}(\mathbf{z}_0) \times \mathcal{C}(\mathbf{z}_0) & &(27) \\
A_{i,j+L} &= 0 & &\text{if } i > j \ \forall (i,j) \in \mathcal{M}(\mathbf{z}_0) \times \mathcal{M}(\mathbf{z}_0) & &(28) \\
A_{i+L,j+L} &= 0 & &\text{if } \sigma^{-1}(i) \geq \sigma^{-1}(j) \ \forall (i,j) \in \mathcal{C}(\mathbf{z}_0) \times \mathcal{C}(\mathbf{z}_0) & &(29) \\
A_{i+L,j+L} &= 0 & &\forall (i,j) \in \mathcal{M}(\mathbf{z}_0) \times \mathcal{C}(\mathbf{z}_0) & &(30) \\
A_{i+L,j+L} &= 0 & &\text{if } i \geq j \ \forall \ (i,j) \in \mathcal{M}(\mathbf{z}_0) \times \mathcal{M}(\mathbf{z}_0) & &(31) \\
A_{i,j} &= -\infty & &\text{otherwise.} & &(32)
\end{aligned}
$$

This construction ensures: each mask token $(\mathbf{z}_0^i)_{i \in \mathcal{M}(\mathbf{z}_0)}$ attends to (i) itself (26), (ii) clean tokens in $\mathbf{z}_0$ (equivalently $(\mathbf{x}^i)_{i \in \mathcal{C}(\mathbf{z}_0)}$) (27), and (iii) clean versions of mask tokens on its left (28). A clean token $(\mathbf{z}_0^i)_{i \in \mathcal{C}(\mathbf{z}_0)}$ can attend to anything because no other token attends to them. The tokens in $\mathbf{x}$ that are unmasked in $\mathbf{z}_0$, $\{\mathbf{x}^i \mid i \in \mathcal{C}(\mathbf{z}_0)\}$, have causal attention per $\sigma$ (29); while the ones corresponding to mask tokens in $\mathbf{z}_0$, $(\mathbf{x}^i)_{i \in \mathcal{M}(\mathbf{z}_0)}$, attend to $\{\mathbf{x}^j \mid j \in \mathcal{C}(\mathbf{z}_0)\}$ (30) and $\{\mathbf{x}^j \mid j \in \mathcal{M}(\mathbf{z}_0), i \geq j\}$ (31).

See Fig. 8 for an illustrative example.

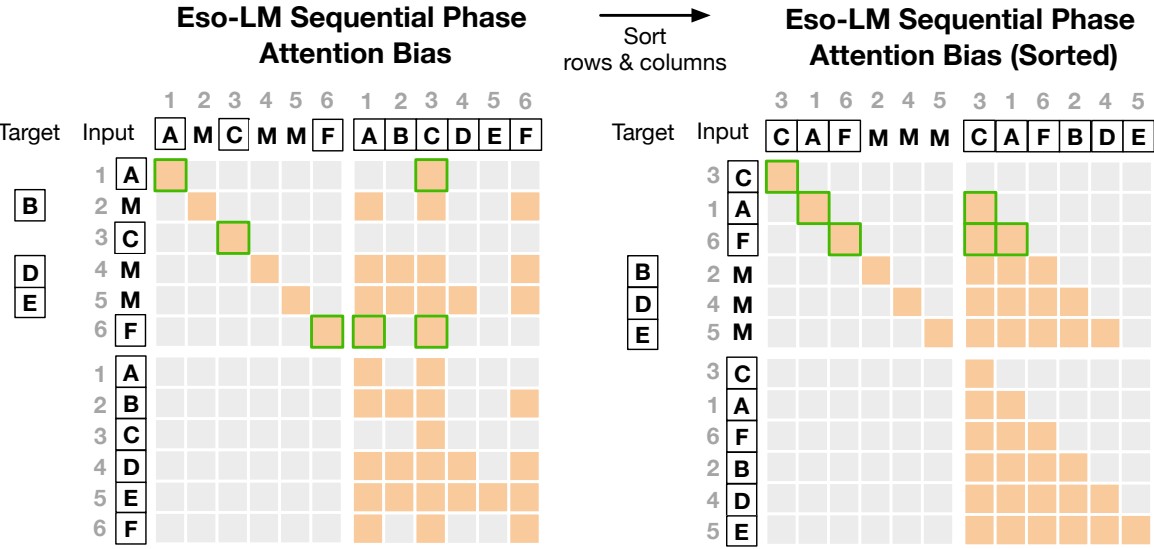

*Figure 8.* Comparison of attention biases for Eso-LMs sequential-phase training, before and after sorting the rows and columns of each of the four $L \times L$ blocks by $\sigma$. **Orange** represents 0 (attention) and **gray** represents $-\infty$ (no attention). The clean sequence is $\mathbf{x} = (A, B, C, D, E, F)$ and hence $L = 6$. After random masking, we obtain $\mathbf{z}_0 = (A, M, C, M, M, F)$. The integers denote the position indices with $\mathcal{M}(\mathbf{z}_0) = \{2, 4, 5\}$ and $\mathcal{C}(\mathbf{z}_0) = \{1, 3, 6\}$. The random ordering among $\mathcal{C}(\mathbf{z}_0)$ is $(3, 1, 6)$. **Green** highlights the extra connections added from clean tokens in $\mathbf{z}_0$ so that the attention bias display classic patterns after sorting – they don't contribute to the transformer output because no other token attends to clean tokens in $\mathbf{z}_0$.

```python
from torch.nn.attention.flex_attention import create_block_mask
from functools import partial

def _seq_mask(b, h, q_idx, kv_idx, n=None):
  # Indicate whether token belongs to zt or x
  x_flag_q = (q_idx >= n)
  x_flag_kv = (kv_idx >= n)

  # Adjust indices
  q_idx2 = torch.where(x_flag_q == 1, q_idx - n, q_idx)
  kv_idx2 = torch.where(x_flag_kv == 1, kv_idx - n, kv_idx)

  # 1. Diagonal Mask (Upper Left)
  diagonal = (q_idx2 == kv_idx2) & (x_flag_q == x_flag_kv)

  # 2. Offset Causal Mask (Upper Right)
  offset_causal = (q_idx2 > kv_idx2) & (x_flag_kv == 1) & (x_flag_q == 0)

  # 3. Causal Mask (Lower Right)
  causal = (q_idx2 >= kv_idx2) & (x_flag_kv == 1) & (x_flag_q == 1)

  # Combine the 3 masks together
  return diagonal | offset_causal | causal

def _get_seq_mask(seq_len):
  # Here, seq_len means the length of zt only
  return create_block_mask(
    partial(_seq_mask, n=seq_len),
    B=None, H=None, Q_LEN=seq_len*2, KV_LEN=seq_len*2)
```

*Figure 9.* We implement the attention bias from Fig. 8 (Right) as a FlexAttention-compatible sparse masking function shown above that can handle arbitrary sequence lengths. This enables Just-In-Time compilation that's significantly faster and more memory efficient than `scaled_dot_product_attention` in PyTorch.

## B.8. Efficient Any-order Likelihood Evaluation

See Fig. 10 for an illustrative example.

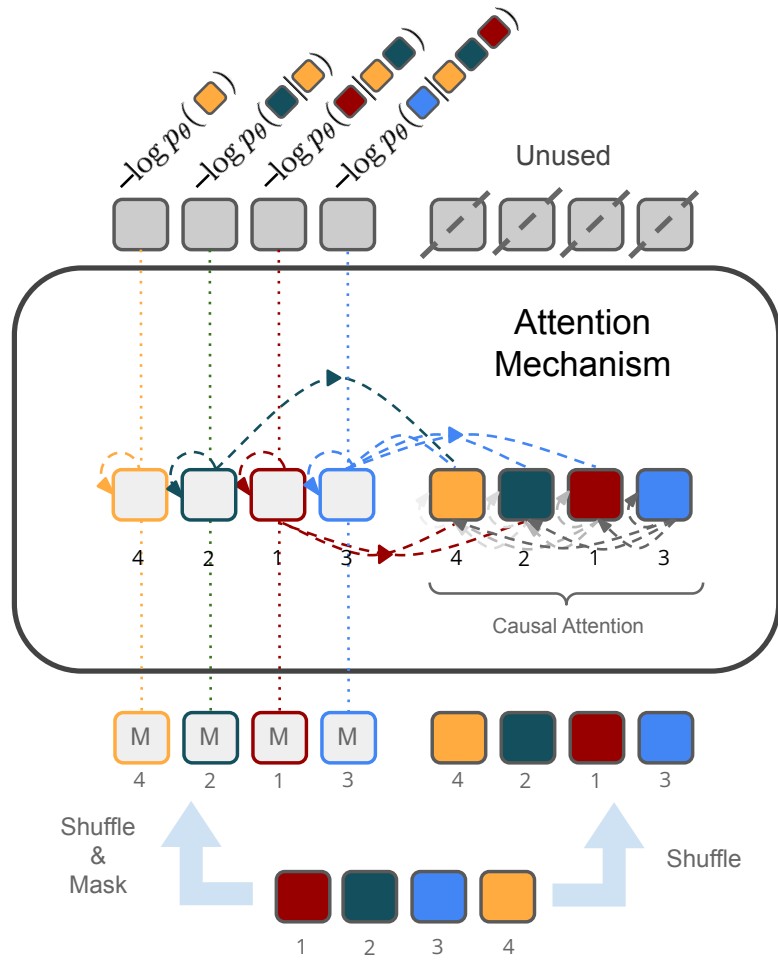

*Figure 10.* Efficient any-order likelihood evaluation in a single forward pass using the attention bias in Fig. 8 (Right). Given a clean sequence, we shuffle it according to a chosen ordering and also create a fully masked version. We concatenate these two sequences and feed them into the transformer. Each mask position attends to itself and previous clean positions under the chosen ordering. Similarly, each clean position attends to itself and previous clean positions but we ignore outputs over the clean positions; this is to simulate a clean context for each mask position, as described in Sec. 4.1.2.

## B.9. Attention Mechanism for Sampling

During sampling step $k$, given a partially masked sequence $\mathbf{z}_k$, the denoising model is required to denoise the mask tokens $\{\mathbf{z}_k^i | i \in S_k\}$ for $S_k \in \mathcal{S} = \{S_1, \ldots, S_K\}$ where $K = |\mathcal{S}|$. We perform a forward pass on the subset of tokens $\{\mathbf{z}_k^i | i \in \mathcal{C}(\mathbf{z}_k) \cup S_k\}$. It is crucial to note that while performing a forward pass on a subset of tokens, the positional embeddings of these tokens in the actual sequence are preserved. Below we discuss the attention bias used in the forward pass.

Let $D_k = \mathcal{C}(\mathbf{z}_k)$ be the set of position indices of tokens decoded prior to step $k$. Importantly, we do not need to make any distinction between tokens decoded in the diffusion phase or those decoded in the sequential phase. This flexibility allows our sampler to use any denoising schedule $\mathcal{S}$.

Let $\sigma$ be the denoising ordering derived from $\mathcal{S}$. We define the $L \times L$ attention bias at step $k$ by

$$A_{i,j} = \begin{cases} 0 & \text{if } \sigma^{-1}(i) \geq \sigma^{-1}(j) \ \forall (i,j) \in (D_k \cup S_k) \times (D_k \cup S_k) \quad (33) \\ -\infty & \text{otherwise,} \quad (34) \end{cases}$$

which is simply causal attention applied to clean tokens generated prior to step $k$ and mask tokens to be decoded in step $k$, both sorted by $\sigma$. Causal attention allows for KV caching, as shown in Fig. 11.

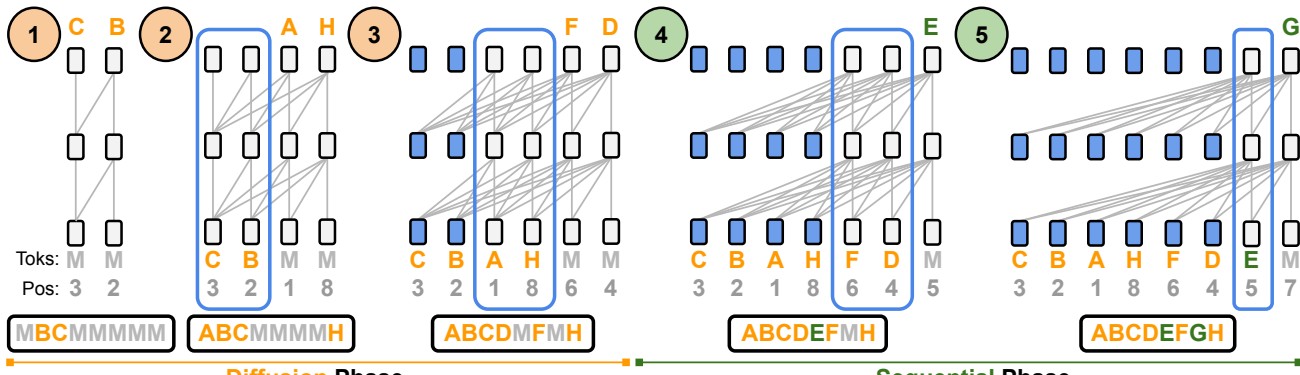

*Figure 11.* (Copy of Fig. 2) Efficient generation of an example sequence with Eso-LMs. During **Diffusion** Phase, Eso-LMs denoise one or more, potentially non-neighboring mask tokens (M) per step. During **Sequential** Phase, Eso-LMs denoise the remaining mask tokens one at a time from left to right. Eso-LMs allows for **KV caching in both phases** using just **a single unified KV cache**: **blue** bounding boxes enclose transformer cells that are building their KV cache; a cell becomes **blue** once its KV cache is built. The sequences below the transformers depict tokens in their natural order.

## B.10. Transformer Implementation

```python
import torch.nn as nn

class Transformer(nn.Module):
  # ...
  def _get_attention_mask(self, diffusion_mode, seq_len):
    if diffusion_mode:
      return _get_causal_mask(seq_len)
    else:
      return _get_seq_mask(seq_len)

  def _sample_ordering(self, zt, shuffle_masks):
    masked = zt == self.mask_index
    offsets = torch.rand(zt.shape)
    if not shuffle_masks:
      # Induce left-to-right order within masked tokens
      offsets[masked] = torch.linspace(0, 1, torch.sum(masked))
    ordering = (masked + offsets).argsort(descending=False)
    return ordering

  def _sort(self, zt, ordering):
    return torch.gather(zt, dim=1, index=ordering)

  def forward(self, zt, x=None):
    '''
    x [batch size, L]: clean sequence (only for sequential training)
    zt [batch size, L]: randomly masked sequence
    '''
    seq_len = zt.shape[1]
    # Construct rotary embeddings for a given sequence
    rotary = self.rotary_emb(zt)  # [batch size, L, d]

    ### -- Start Extra Code --
    diffusion_mode = x is None
    attn_mask = self._get_attention_mask(diffusion_mode, seq_len)

    if diffusion_mode:  # Diffusion Mode Shuffling
      # [batch size, L]
      ordering = self._sample_ordering(zt, shuffle_masks=True)
      x = self._sort(zt, ordering)
    else:  # Sequential Mode Shuffling
      # [batch size, L]
      ordering = self._sample_ordering(zt, shuffle_masks=False)
      x = torch.cat([
        self._sort(x, ordering), self._sort(zt, ordering)], dim=1)
      rotary = self._sort(rotary, ordering)
      rotary = torch.cat([rotary, rotary], dim=1)
    ### -- End Extra Code --

    # Standard transformer forward pass
    for i in range(len(self.blocks)):
      x = self.transformer_blocks[i](
        x, rotary=rotary, attn_mask=attn_mask)
    logits = self.output_layer(x)

    # Logits will be compared against shuffled targets
    return logits, ordering
```

*Figure 12.* Eso-LMs introduce minimal changes to the Transformer architecture. See Fig. 7 for `_get_causal_mask` and Fig. 9 for `_get_seq_mask`. Code for diffusion and sequential mode shuffling follows the description in Sec. 4.1.1 and Sec. 4.1.2 respectively.

# C. Experimental Details

## C.1. Low discrepancy sampler

To reduce variance during training we use a low-discrepancy sampler, similar to that in Kingma et al. (2021). Specifically, when processing a minibatch of $N$ samples, instead of independently sampling $N$ from a uniform distribution, we partition the unit interval and sample the time step for each sequence $i \in \{1, \ldots, N\}$ from a different portion of the interval $t_i \sim U\left[\frac{i-1}{N}, \frac{i}{N}\right]$. This ensures that our sampled timesteps are evenly spaced across the interval $[0, 1]$, reducing ELBO variance.

## C.2. Likelihood evaluation

We use a single monte-carlo estimate for $t$ for each example to evaluate the likelihood. We use a low discrepancy sampler (Kingma et al., 2021) to reduce the variance of the estimate.

## C.3. Language modeling

We detokenize the One Billion Words dataset following (Lou et al., 2024; Sahoo et al., 2024a), whose code can be found here[1]. We tokenize the One Billion Words dataset with the `bert-base-uncased` tokenizer, following Austin et al. (2021); He et al. (2022). We concatenate and wrap sequences (also known as sequence packing) to a length of 128 (Raffel et al., 2020). When wrapping, we add the `[CLS]` token in-between concatenated sequences. The final preprocessed sequences also have the `[CLS]` token as their first and last token. Unlike Sahoo et al. (2024a); Lou et al. (2024); He et al. (2022), we apply sequence packing to LM1B, making our setup more challenging and resulting in higher perplexities given the same model (Table 1).

We tokenize OpenWebText with the `GPT2` (Radford et al., 2019) tokenizer. We concatenate and wrap them to a length of 1,024. When wrapping, we add the `eos` token in-between concatenated sequences. Unlike for One Billion Words, the final preprocessed sequences for OpenWebText do not have special tokens as their first and last token. Since OpenWebText does not have a test split, we leave the last 100k docs as test.

Eso-LMs shares the same parameterization as our autoregressive baseline, SEDD, MDLM, UDLM, and Duo: a modified diffusion transformer architecture (Peebles & Xie, 2023) from Lou et al. (2024); Sahoo et al. (2024a). We use 12 layers, a hidden dimension of 768, 12 attention heads. Eso-LMs do not use timestep embedding used in uniform diffusion models (SEDD Uniform, UDLM, Duo). Word embeddings are not tied between the input and output. We train BD3-LMs using the original code provided by their authors.

We use the standard linear noise schedule $\alpha_t = 1 - t$ for MDLM and a scaled-down linear noise schedule $\alpha_t = \alpha_0(1 - t)$ for Eso-LMs. We use the AdamW optimizer with a batch size of 512, constant learning rate warmup from 0 to a learning rate of 3e-4 for 2,500 steps. We use a constant learning rate for 1M steps on One Billion Words and for 250K steps for OpenWebText. We use a dropout rate of 0.1. We train models on nodes of 8 H200 GPUs. On OpenWebText for 250K steps, training takes ~27 hours when $\alpha_0 = 1$ and ~37 hours when $\alpha_0 < 1$ due to the additional AR loss. Throughput is benchmarked on single H200 GPUs and latency is benchmarked on single A6000 GPUs.

---

[1]https://github.com/louaaron/Score-Entropy-Discrete-Diffusion/blob/main/data.py

# D. Eso-LMs (A) as an Ablation

## D.1. Attention Mechanism for Diffusion Phase Training

The denoising transformer receives $\mathbf{z}_t \sim q_t(.|\mathbf{x})$ as input, which contains the mask tokens to denoise, and $\mathbf{x}$ as target. A random ordering $\sigma \sim \mathcal{P}_L$ is sampled with the only constraint that clean tokens in $\mathbf{z}_t$ precede mask tokens in $\mathbf{z}_t$ in $\sigma$. We define the $L \times L$ attention bias by

$$A_{i,j} = \begin{cases} 0 & \forall (i,j) \in \mathcal{C}(\mathbf{z}_t) \times \mathcal{C}(\mathbf{z}_t) & (35) \\ 0 & \text{if } \sigma^{-1}(i) \geq \sigma^{-1}(j) \ \forall (i,j) \in \mathcal{M}(\mathbf{z}_t) \times [L] & (36) \\ -\infty & \text{otherwise.} & (37) \end{cases}$$

Clean tokens $\{\mathbf{z}_t^i | i \in \mathcal{C}(\mathbf{z}_t)\}$ have bidirectional attention among them (35), while a mask token $(\mathbf{z}_t^i)_{i \in \mathcal{M}(\mathbf{z}_t)}$ attends to clean tokens, itself and prior mask tokens per $\sigma$ (36). We can ignore the ordering among clean tokens in $\sigma$ due to the use of bidirectional attention. See Fig. 13 for an example.

**Simplified Implementation** $A$ becomes a Prefix-LM (Raffel et al., 2020) attention bias if we sort the rows and columns of $A$ by $\sigma$ (Fig. 6), which is simple to implement.

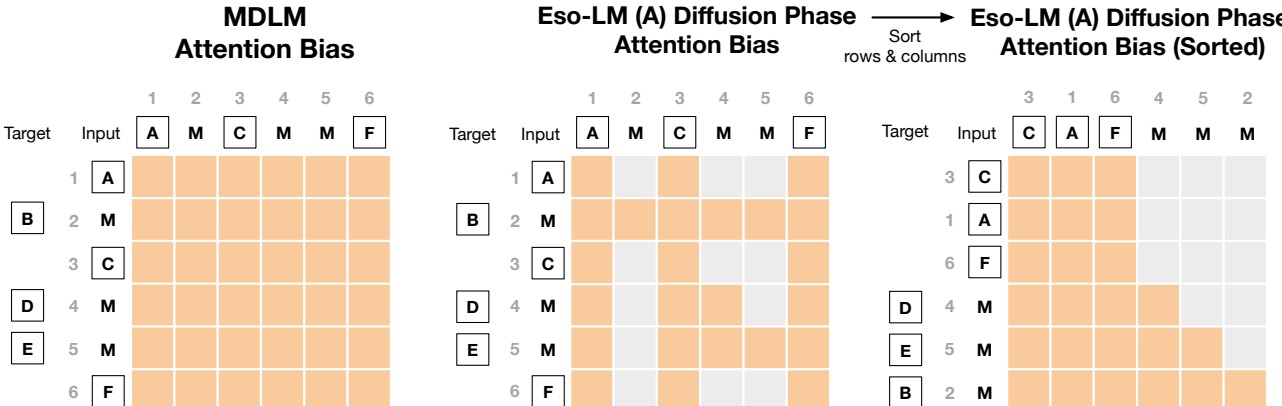

*Figure 13.* Comparing attention biases for MDLM and Eso-LMs (A) diffusion-phase training, before and after sorting the rows and columns by $\sigma$. **Orange** represents 0 (attention) and **gray** represents $-\infty$ (no attention). The clean sequence is $\mathbf{x} = (A, B, C, D, E, F)$ and hence $L = 6$. After random masking, we obtain $\mathbf{z}_t = (A, M, C, M, M, F)$. The integers denote position indices: $\mathcal{M}(\mathbf{z}_t) = \{2, 4, 5\}$ and $\mathcal{C}(\mathbf{z}_t) = \{1, 3, 6\}$. $\sigma = (3, 1, 6, 4, 5, 2) \sim \mathcal{P}_6$ with clean tokens before mask tokens.

## D.2. Attention Mechanism for Sequential Phase Training

The denoising transformer receives the concatenated sequence $\mathbf{z}_0 \oplus \mathbf{x} \in \mathcal{V}^{2L}$ as input, where $\mathbf{z}_0 \sim q_0(.|\mathbf{x})$ contains the mask tokens to denoise, and $\mathbf{x}$ as target. We define the $2L \times 2L$ attention bias by

$$\begin{aligned} A_{i,j} &= 0 & \text{if } i = j \ \forall (i,j) \in \mathcal{M}(\mathbf{z}_0) \times \mathcal{M}(\mathbf{z}_0) & (38) \\ A_{i,j+L} &= 0 & \forall (i,j) \in \mathcal{M}(\mathbf{z}_0) \times \mathcal{C}(\mathbf{z}_0) & (39) \\ A_{i,j+L} &= 0 & \text{if } i > j \ \forall (i,j) \in \mathcal{M}(\mathbf{z}_0) \times \mathcal{M}(\mathbf{z}_0) & (40) \\ A_{i+L,j+L} &= 0 & \forall (i,j) \in \mathcal{C}(\mathbf{z}_0) \times \mathcal{C}(\mathbf{z}_0) & (41) \\ A_{i+L,j+L} &= 0 & \forall (i,j) \in \mathcal{M}(\mathbf{z}_0) \times \mathcal{C}(\mathbf{z}_0) & (42) \\ A_{i+L,j+L} &= 0 & \text{if } i \geq j \ \forall (i,j) \in \mathcal{M}(\mathbf{z}_0) \times \mathcal{M}(\mathbf{z}_0) & (43) \\ A_{i,j} &= -\infty & \text{otherwise.} & (44) \end{aligned}$$

See Fig. 14 for an example. This construction ensures that a mask token $(\mathbf{z}_0^i)_{i \in \mathcal{M}(\mathbf{z}_0)}$ attends to (i) itself (38), (ii) the clean tokens $\{\mathbf{x}^j | j \in \mathcal{C}(\mathbf{z}_0)\}$ (39) and (iii) the clean versions of mask tokens on its left $\{\mathbf{x}^j | j \in \mathcal{M}(\mathbf{z}_0), i > j\}$ (40). A clean token $(\mathbf{z}_0^i)_{i \in \mathcal{C}(\mathbf{z}_0)}$ can attend to anything because no other token attends to them (44). The attention mechanism for tokens in the clean context $\mathbf{x}_0$ is described as follows. Tokens $\{\mathbf{x}^i | i \in \mathcal{C}(\mathbf{z}_0)\}$ have bidirectional attention (41). A clean token corresponding to a mask token, $(\mathbf{x}^i)_{i \in \mathcal{M}(\mathbf{z}_0)}$, attends to $\{\mathbf{x}^j | j \in \mathcal{C}(\mathbf{z}_0)\}$ (42) and $\{\mathbf{x}^j | j \in \mathcal{M}(\mathbf{z}_0), i \geq j\}$ (43).

**Simplified Implementation**  Let $\sigma$ be an ordering such that: (i) clean tokens in $\mathbf{z}_0$ precede mask tokens in $\mathbf{z}_0$ in $\sigma$ and (ii) mask tokens in $\mathbf{z}_0$ are in natural order in $\sigma$. The ordering among clean tokens $\{\mathbf{x}^i | i \in \mathcal{C}(\mathbf{z}_0)\}$ can be ignored with bidirectional attention. When the rows and columns of each of the four $L$-by-$L$ blocks are sorted by $\sigma$, $A$ shows classic attention patterns (Fig. 14) that are simple to implement.

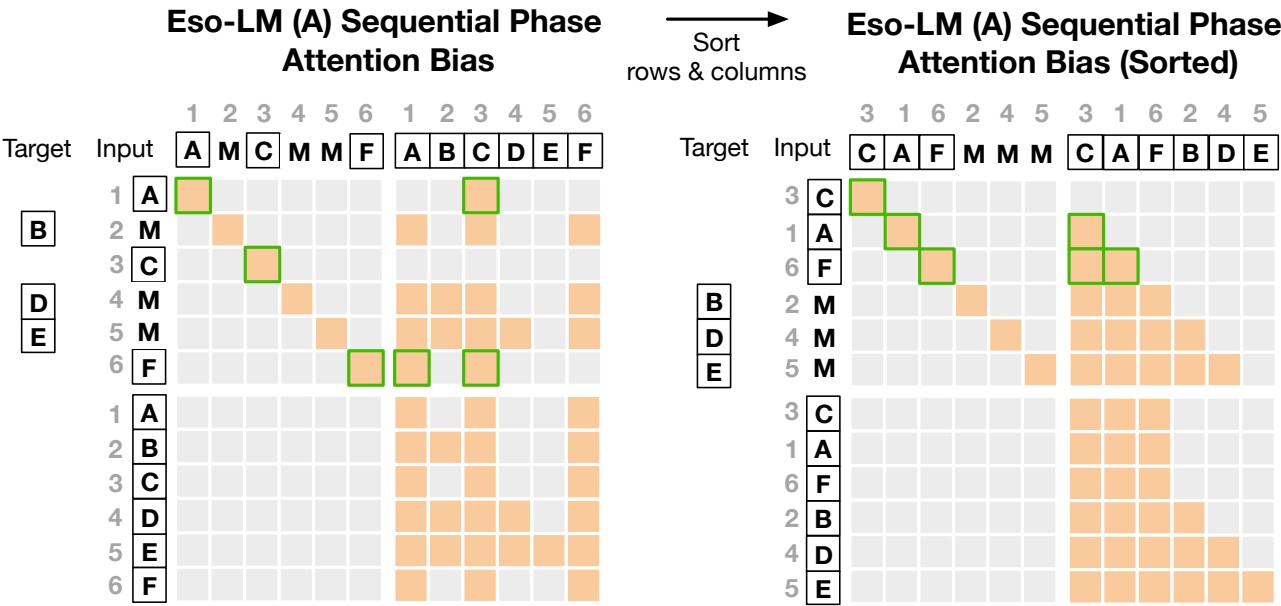

*Figure 14.* Comparison of attention biases for Eso-LMs (A) sequential-phase training, before and after sorting the rows and columns of each of the four $L \times L$ blocks by $\sigma$. **Orange** represents 0 (attention) and **gray** represents $-\infty$ (no attention). The clean sequence is $\mathbf{x} = (A, B, C, D, E, F)$ and hence $L = 6$. After random masking, we obtain $\mathbf{z}_0 = (A, M, C, M, M, F)$. The integers denote the position indices with $\mathcal{M}(\mathbf{z}_0) = \{2, 4, 5\}$ and $\mathcal{C}(\mathbf{z}_0) = \{1, 3, 6\}$. The random ordering among $\mathcal{C}(\mathbf{z}_0)$ is $(3, 1, 6)$. **Green** highlights the extra connections added from clean tokens in $\mathbf{z}_0$ so that the attention bias display classic patterns after sorting – they don't contribute to the transformer output because no other token attends to clean tokens in $\mathbf{z}_0$.

### D.3. Attention Mechanism for Sampling

During diffusion or sequential sampling, given a partially masked sequence $\mathbf{z}_k$, the denoising model is required to denoise the mask tokens $\{\mathbf{z}_k^i | i \in S_k\}$ for $S_k \in \mathcal{S} = \{S_1, \dots, S_K\}$ where $K = |\mathcal{S}|$. We perform a forward pass on the subset of tokens $\{\mathbf{z}_k^i | i \in \mathcal{C}(\mathbf{z}_k) \cup S_k\}$. It is crucial to note that while performing a forward pass on a subset of tokens, the positional embeddings of these tokens in the actual sequence are preserved. Below we discuss the attention bias used in the forward pass.

Let $D_k^{\text{MDM}}$ be the set of indices of tokens decoded in the diffusion phase prior to step $k$ and $D_k^{\text{AR}}$ be that for the sequential phase. Let ordering $\sigma$ be the order in which we denoise tokens defined by $\mathcal{S}$. We define the $L \times L$ attention bias at step $k$ by

$$A_{i,j} = \begin{cases} 0 & \forall (i,j) \in D_k^{\text{MDM}} \times D_k^{\text{MDM}} & (45) \\ 0 & \forall (i,j) \in D_k^{\text{AR}} \times D_k^{\text{MDM}} & (46) \\ 0 & \text{if } i \geq j \; \forall (i,j) \in D_k^{\text{AR}} \times D_k^{\text{AR}} & (47) \\ 0 & \forall (i,j) \in S_k \times (D_k^{\text{MDM}} \cup D_k^{\text{AR}}) & (48) \\ 0 & \text{if } \sigma^{-1}(i) \geq \sigma^{-1}(j) \; \forall (i,j) \in S_k \times S_k & (49) \\ -\infty & \text{otherwise.} & (50) \end{cases}$$

Clean tokens decoded during diffusion $\{\mathbf{z}_k^i | i \in D_k^{\text{MDM}}\}$ have bidirectional attention among them (45). A clean token decoded sequentially $(\mathbf{z}_k^i)_{i \in D_k^{\text{AR}}}$ attends to clean tokens decoded during diffusion $\{\mathbf{z}_k^j | j \in D_k^{\text{MDM}}\}$ (46), itself, and prior clean tokens decoded sequentially $\{\mathbf{z}_k^j | j \in D_k^{\text{AR}}, i > j\}$ (47). A mask token to denoise $(\mathbf{z}_k^i)_{i \in S_k}$ attends to all decoded clean tokens $\{\mathbf{z}_k^j | j \in D_k^{\text{MDM}} \cup D_k^{\text{AR}}\}$ (48), itself, and prior mask tokens to denoise per $\sigma$: $\{\mathbf{z}_k^j | j \in S_k, \sigma^{-1}(i) > \sigma^{-1}(j)\}$ (49). Mask tokens not scheduled to denoise $(\mathbf{z}_k^i)_{i \in S_{>k}}$ can attend to anything because no other token attends to them (50).

Fig. 15 shows how Eso-LMs (A) generates with KV caching only during the sequential phase.

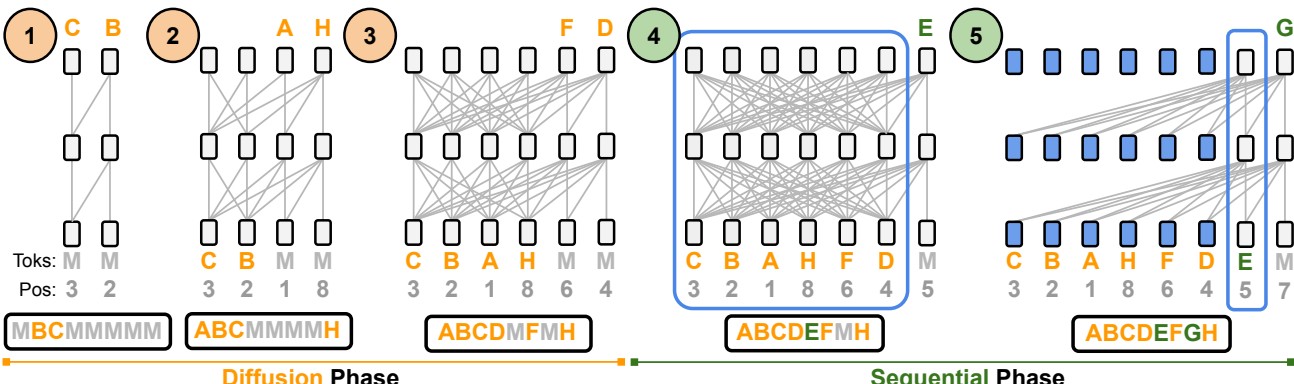

*Figure 15.* Generation of an example sequence with Eso-LMs (A). During **Diffusion** Phase, Eso-LMs denoise one or more, potentially non-neighboring mask tokens (M) per step. During **Sequential** Phase, Eso-LMs denoise the remaining mask tokens one at a time from left to right. Eso-LMs (A) allows for **KV caching in sequential phase** only: **blue** bounding boxes enclose transformer cells that are building their KV cache; a cell becomes **blue** once its KV cache is built. The sequences below the transformers depict tokens in their natural order.

# E. Additional Experiments and Results

### E.1. Discussion of Kong et al. (2023)

The claim in Kong et al. (2023) that the MDLM NELBO equals the exact likelihood is incorrect. Marginalizing over path measures yields only an upper bound, not the equality stated in Sec. C.1 of Kong et al. (2023).

The small-scale empirical validation experiments on DNA data in Kong et al. (2023) shows that the gap between the NELBO and NLL is negligible. Similarly, we design a first-order Markov process with $|\mathcal{V}| = 254$ and $L = 30$, where the ground-truth NLL is about 1.73. On this simple setting, AR and MDLM achieve essentially the same value of about 1.73. However, as verified in Table 1, this tightness does not persist at scale.

### E.2. Comparison of Training Speed

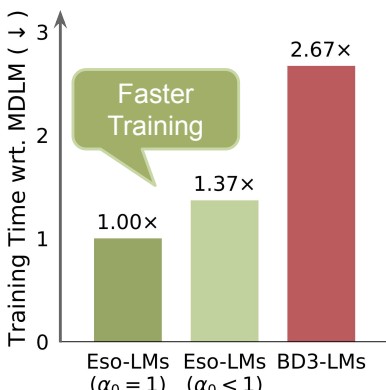

*Figure 16.* Eso-LMs have similar training time to MDLM and are much faster to train than BD3-LMs.

### E.3. Additional References

**References**   D3PM (Austin et al., 2021), Diffusion-LM (Li et al., 2022), DiffusionBert (He et al., 2022), SEDD (Lou et al., 2024), UDLM (Schiff et al., 2025), and Duo (Sahoo et al., 2025).

### E.4. Ablation on Split Proportion

See Table 4.

*Table 4.* Test perplexities ($\downarrow$) on LM1B for Eso-LMs (A) trained for 500K vs. the proportion $\kappa$ of examples in each batch used for evaluating the MDM loss in (7) during training. Remaining examples in each batch are used for evaluating the AR loss in (7) during training.

|  | $\kappa = 0.75$ | $\kappa = 0.5$ | $\kappa = 0.25$ | $\kappa = 0.125$ |
|---|---|---|---|---|
| Eso-LMs (A) |  |  |  |  |
| $\alpha_0 = 0.5$ | 32.25 | **31.53** | Diverged | Diverged |
| $\alpha_0 = 0.25$ | 30.49 | **29.33** | Diverged | Diverged |
| $\alpha_0 = 0.125$ | 27.76 | **26.73** | Diverged | Diverged |
| $\alpha_0 = 0.0625$ | 25.92 | **25.07** | Diverged | Diverged |

### E.5. Zero-Shot Likelihood Evaluation

We explore models' ability to generalize by taking models trained on OWT and evaluating how well they model unseen datasets (Table 5). We compare the perplexities of our Eso-LMs with SEDD (Austin et al., 2021), MDLM (Sahoo et al., 2024a), BD3-LMs (Arriola et al., 2025), and an AR Transformer language model. Our zero-shot datasets are validation splits of Penn Tree Bank (PTB; (Marcus et al., 1993)), Wikitext (Merity et al., 2016), LM1B, Lambada (Paperno et al., 2016), AG News (Zhang et al., 2015), and Scientific Papers (Pubmed and Arxiv subsets; (Cohan et al., 2018)).

*Table 5.* Zero-shot perplexities (↓) of models trained for 250K steps on OWT. We report bounds for diffusion models and interpolation methods. Numbers for AR were taken from (Arriola et al., 2025).

|  | PTB | Wikitext | LM1B | Lambada | AG News | Pubmed | Arxiv |
|---|---|---|---|---|---|---|---|
| AR | 82.00 | 26.54 | 52.14 | 51.69 | 55.53 | 49.49 | 44.98 |
| MDLM | 100.17 | 37.08 | 70.79 | 52.06 | 71.37 | 46.51 | 40.21 |
| SEDD Absorb | 99.59 | 38.55 | 72.51 | 52.16 | 72.62 | 47.07 | 41.18 |
| BD3-LM ($L' = 16$) | 95.87 | 32.88 | 65.11 | 50.05 | 61.68 | 43.41 | 40.13 |
| **Eso-LMs (Ours)** |  |  |  |  |  |  |  |
| $\alpha_0 = 1$ | 126.29 | 45.08 | 82.01 | 61.37 | 98.22 | 62.37 | 55.76 |
| $\alpha_0 = 0.5$ | 110.70 | 39.57 | 75.75 | 57.33 | 86.65 | 60.20 | 53.78 |
| $\alpha_0 = 0.25$ | 105.19 | 37.32 | 67.69 | 60.15 | 75.74 | 62.45 | 55.31 |
| $\alpha_0 = 0.125$ | 97.46 | 35.65 | 60.11 | 69.13 | 65.26 | 65.27 | 57.4 |

### E.6. Importance-Weighted Bounds

See Table 6, Table 7, and Table 8. Each reported number is computed using a single H200 GPU within 48 hours. Therefore, our method can be easily scaled to, e.g., $K = 1M$, using a cluster of GPUs.

*Table 6.* Test perplexities (↓) on LM1B for MDLM trained for 1M steps, computed using importance-weighted bounds. Estimates are computed by varying the number of orderings sampled ($K$) per batch of 32 examples in the OWT test set.

|  | $K = 1$ | $K = 10$ | $K = 20$ | $K = 50$ | $K = 100$ | $K = 1000$ |
|---|---|---|---|---|---|---|
| **MDLM** | 31.19 | 28.66 | 28.26 | 27.85 | 27.57 | 26.82 |

*Table 7.* Test perplexities (↓) on LM1B for Eso-LMs trained for 1M steps, computed using importance-weighted bounds. We report multiple estimates for each $\alpha_0$ by varying the number of orderings sampled ($K \in \{1, 10, 20, 50, 100, 1000, 5000\}$) per batch of 32 examples in the LM1B test set.

|  | $K = 1$ | $K = 10$ | $K = 20$ | $K = 50$ | $K = 100$ | $K = 1000$ | $K = 5000$ |
|---|---|---|---|---|---|---|---|
| **Eso-LMs (Ours)** |  |  |  |  |  |  |  |
| $\alpha_0 = 1$ | 37.53 | 34.49 | 33.94 | 33.37 | 33.00 | 32.09 | **31.65** |
| $\alpha_0 = 0.5$ | 33.55 | 30.69 | 30.18 | 29.64 | 29.31 | 28.48 | **28.07** |
| $\alpha_0 = 0.25$ | 29.64 | 27.00 | 26.56 | 26.08 | 25.79 | 25.11 | **24.80** |
| $\alpha_0 = 0.125$ | 26.94 | 24.54 | 24.18 | 23.84 | 23.64 | 23.19 | **23.02** |
| $\alpha_0 = 0.0625$ | 25.25 | 23.30 | 23.05 | 22.84 | 22.72 | 22.48 | **22.39** |

*Table 8.* Test perplexities (↓) on OWT for Eso-LMs trained for 250K steps, computed using importance-weighted bounds. We report multiple estimates for each $\alpha_0$ by varying the number of orderings sampled ($K \in \{1, 10, 20, 50, 100, 1000\}$) per batch of 32 examples in the OWT test set.

|  | $K = 1$ | $K = 10$ | $K = 20$ | $K = 50$ | $K = 100$ | $K = 1000$ |
|---|---|---|---|---|---|---|
| **Eso-LMs (Ours)** |  |  |  |  |  |  |
| $\alpha_0 = 1$ | 31.71 | 30.50 | 30.26 | 29.99 | 29.80 | **29.31** |
| $\alpha_0 = 0.5$ | 28.95 | 27.77 | 27.53 | 27.27 | 27.09 | **26.61** |
| $\alpha_0 = 0.25$ | 25.23 | 24.16 | 23.95 | 23.72 | 23.56 | **23.15** |
| $\alpha_0 = 0.125$ | 22.24 | 21.35 | 21.17 | 20.98 | 20.86 | **20.53** |

### E.7. Eso-LMs (A) Likelihood Evaluation

See Table 9 and Table 10.

*Table 9.* Test perplexities (↓) on LM1B for Eso-LMs, Eso-LMs (A) and MDLM trained for 1M steps.

| $\alpha_0$ | Eso-LMs | Eso-LMs (A) | MDLM |
|---|---|---|---|
| 1.0 (full diffusion mode) | 36.12 | 30.96 | 31.78 |
| 0.5 | 32.53 | 30.51 | – |
| 0.25 | 29.23 | 28.44 | – |
| 0.125 | 26.29 | 25.97 | – |
| 0.0625 | **24.53** | **24.51** | – |

*Table 10.* Test perplexities (↓) on OWT for Eso-LMs, Eso-LMs (A) and MDLM trained for 250K steps.

| $\alpha_0$ | Eso-LMs | Eso-LMs (A) | MDLM |
|---|---|---|---|
| 1.0 (full diffusion mode) | 30.06 | 26.21 | 25.19 |
| 0.5 | 27.94 | 25.38 | – |
| 0.25 | 24.71 | 23.78 | – |
| 0.125 | **21.92** | **21.47** | – |

### E.8. Pareto Frontier of Generative Perplexity

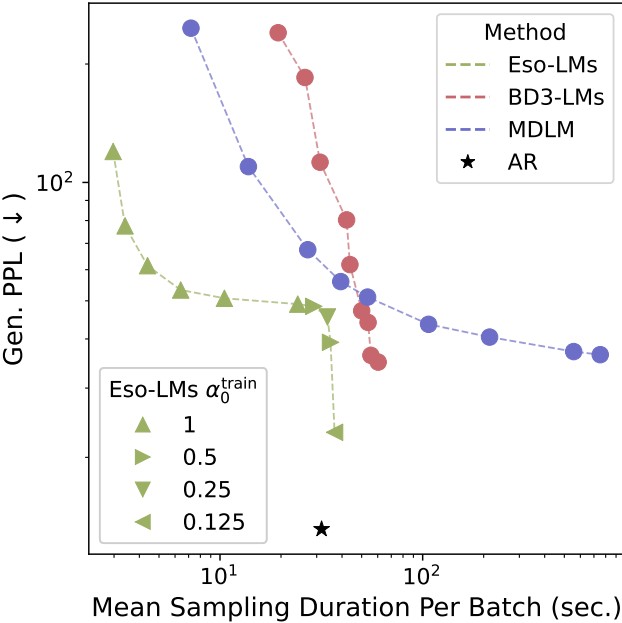

*Figure 17.* Eso-LMs establish SOTA on the Pareto frontier of sampling speed and Gen. PPL. Both axes are in **log scale**.

### E.9. Pareto Frontier of Eso-LMs with $\alpha_0^{\text{train}} = 1$

See Fig. 18 and Fig. 19 for a comparison of the Pareto frontier of Eso-LMs trained with $\alpha_0^{\text{train}} = 1$ against Pareto frontiers reported in the main paper (Fig. 17 and Fig. 4).

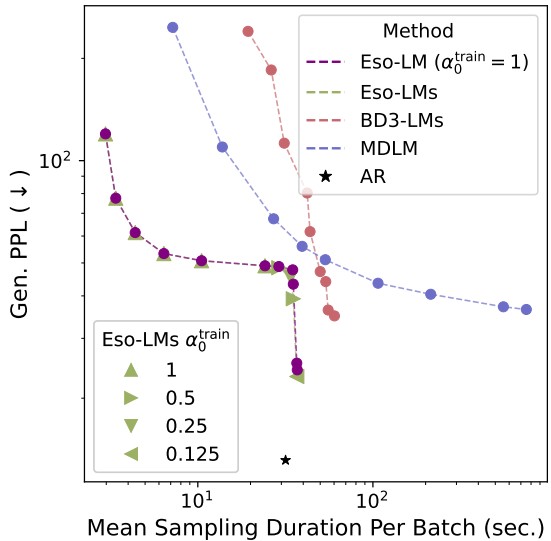

*Figure 18.* Eso-LMs establish SOTA on the Pareto frontier of sampling speed and Gen. PPL.

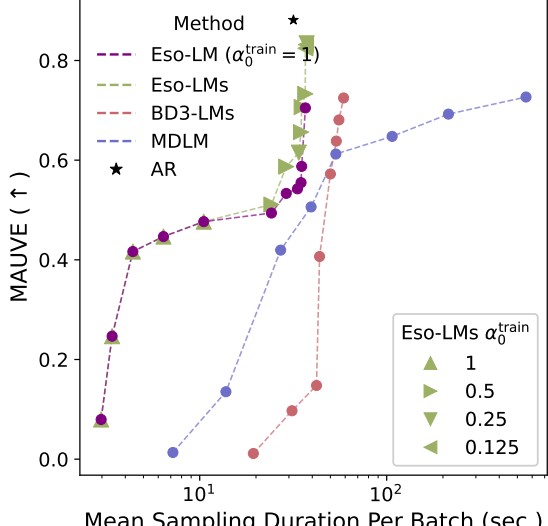

*Figure 19.* Eso-LMs establish SOTA on the Pareto frontier of sampling speed and MAUVE.

### E.10. Heuristic Improved Sampler

We propose a heuristic improved sampler that only performs parallel decoding for evenly spaced positions across the sequence length. For example, with length 1024 and parallelism 4, the model first predicts positions 0, 255, 511, and 767 simultaneously. Subsequent steps need not target adjacent indices (e.g., 1, 256, 512, and 768), but instead continue to perform parallel decoding for a random set of 4 interleaved, far-apart positions. This process is iterated until the sequence is filled.

We use Eso-LMs trained with $\alpha_0^{\text{train}} = 1$ and generate samples by fixing $\alpha_0^{\text{eval}} = 1$ and varying $T$ to control NFEs and sampling time. For the improved sampler, we use Eso-LMs trained with $\alpha_0^{\text{train}} = 1$ and generate samples by varying the amount of parallelism, i.e., number of tokens generated in parallel: $\{64, 32, 16, 8, 4, 2, 1\}$. We find that the sampler significantly improves generation quality at low NFEs (Fig. 20 and Fig. 21) while offering less improvements at high NFEs, which is expected.

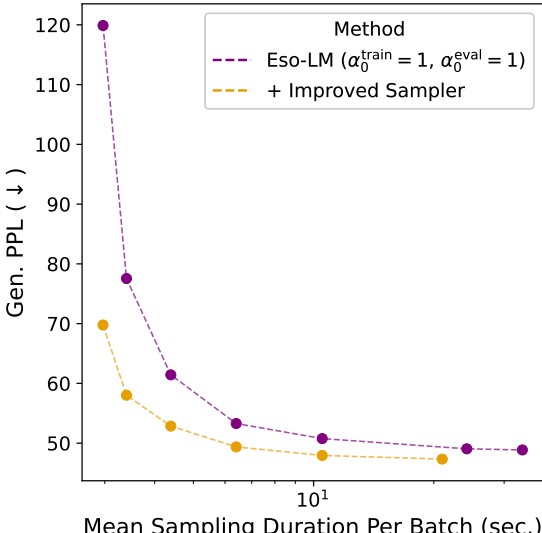

*Figure 20.* Heuristic improved sampler improves Gen. PPL Pareto frontier at low NFEs.

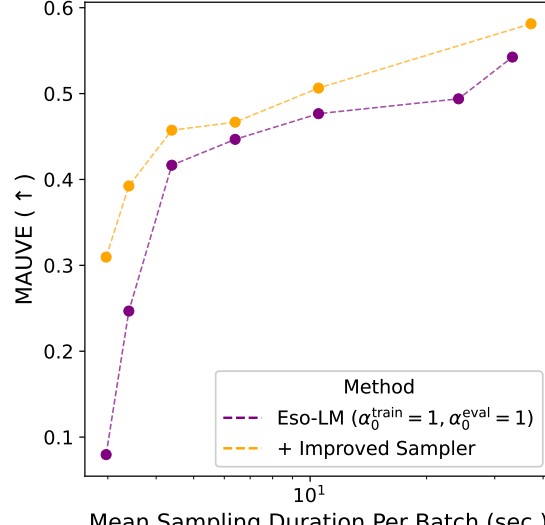

*Figure 21.* Heuristic improved sampler improves MAUVE Pareto frontier at low NFEs.

### E.11. Generation Latency at Long Context

*Table 11.* Sampling time ($\downarrow$) in seconds for sequence lengths $L \in \{2048, 8192\}$ with NFEs set to $L$ for all methods. Reported values are $\text{mean}_{\text{std}}$ over 5 runs.

| Method | $L = 2048$ | $L = 8192$ |
|---|---|---|
| **AR** | $13.3_{0.9}$ | $54.0_{0.2}$ |
| MDLM | $201.3_{0.4}$ | $5438.3_{3.3}$ |
| BD3-LMs ($L' = 4$) | $24.3_{0.7}$ | $312.0_{1.7}$ |
| BD3-LMs ($L' = 16$) | $21.3_{0.1}$ | $268.1_{1.2}$ |
| **Eso-LMs (Ours)** | $\mathbf{14.6}_{0.3}$ | $\mathbf{82.1}_{0.3}$ |

*Table 12.* Gen. PPL ($\downarrow$), entropy, and sampling time ($\downarrow$) in seconds for sequence length $L = 10240$ with NFEs set to $L$ for all methods. Reported values for sampling time are $\text{mean}_{\text{std}}$ over 5 runs.

| Method | Gen. PPL | Entropy | Time (seconds) |
|---|---|---|---|
| BD3-LMs ($L' = 4$) | 29.50 | 6.5 | $\mathbf{588.6}_{3.2}$ |
| **Eso-LM (Ours)** ($\alpha_0^{\text{train}} = \alpha_0^{\text{eval}} = 0.125$) | **23.40** | 6.3 | $\mathbf{116.4}_{0.4}$ |

### E.12. Quality of Generated Samples by Models Trained on OWT

In Fig. 17 and Fig. 4 we present how the sample quality changes by varying NFEs. The individual values for Gen. PPL, entropy and MAUVE can be found in Fig. 22 (Eso-LMs; Gen. PPL), Fig. 23 (Eso-LMs; MAUVE), Table 13 (Eso-LMs), Table 14 (MDLM), and Table 15 (BD3-LMs).

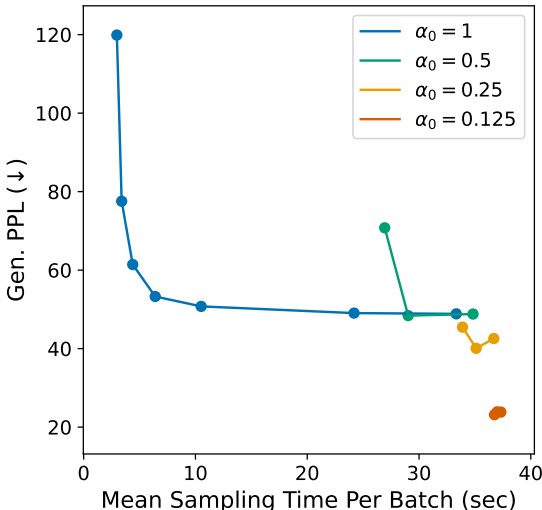

*Figure 22.* Decomposing the Pareto frontier on sampling speed and Gen. PPL of Eso-LMs into individual frontiers where $\alpha_0^{\text{train}} = \alpha_0^{\text{eval}}$ (or $\approx$).

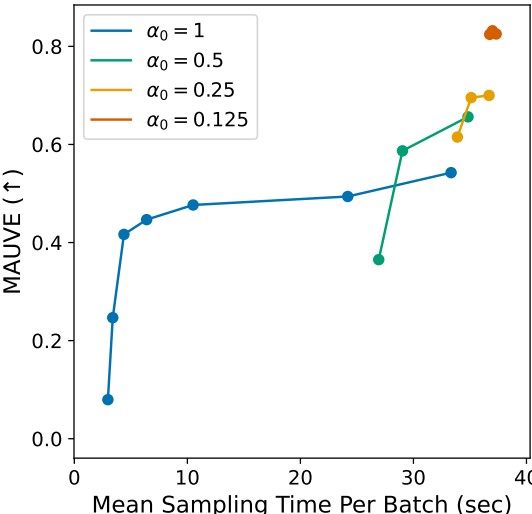

*Figure 23.* Decomposing the Pareto frontier on sampling speed and MAUVE of Eso-LMs into individual frontiers where $\alpha_0^{\text{train}} = \alpha_0^{\text{eval}}$ (or $\approx$).

*Table 13.* Gen. PPL (↓), entropies (↑), and MAUVE (↑) of samples by Eso-LMs trained for 250K steps on OWT.

| $\alpha_0^{\text{train}}$ | $\alpha_0^{\text{eval}}$ | $T$ | NFE | Gen. PPL (↓) | Entropy | MAUVE (↑) | Sampling Time (sec) (↓) |
|---|---|---|---|---|---|---|---|
| 1 | 0.0625 | 16 | 976 | 25.36 | 5.1 | 0.7048 | 36.75 |
| 1 | 0.0625 | 128 | 1010 | 24.74 | 5.1 | 0.6753 | 37.32 |
| 1 | 0.0625 | 1024 | 1022 | 24.23 | 5.1 | 0.6925 | 36.99 |
| 1 | 0.25 | 16 | 784 | 51.11 | 5.4 | 0.4996 | 33.89 |
| 1 | 0.25 | 128 | 879 | 43.31 | 5.3 | 0.5875 | 35.11 |
| 1 | 0.25 | 1024 | 994 | 43.36 | 5.3 | 0.5748 | 36.69 |
| 1 | 0.5 | 16 | 529 | 72.16 | 5.5 | 0.2885 | 26.93 |
| 1 | 0.5 | 128 | 639 | 48.80 | 5.3 | 0.5333 | 29.03 |
| 1 | 0.5 | 1024 | 913 | 47.72 | 5.3 | 0.5549 | 34.83 |
| 1 | 1 | 16 | 16 | 119.89 | 5.5 | 0.0796 | 2.97 |
| 1 | 1 | 32 | 32 | 77.55 | 5.5 | 0.2468 | 3.40 |
| 1 | 1 | 64 | 64 | 61.43 | 5.4 | 0.4166 | 4.39 |
| 1 | 1 | 128 | 128 | 53.28 | 5.4 | 0.4467 | 6.40 |
| 1 | 1 | 256 | 251 | 50.76 | 5.3 | 0.4766 | 10.51 |
| 1 | 1 | 1024 | 646 | 49.05 | 5.3 | 0.4939 | 24.19 |
| 1 | 1 | 4096 | 906 | 48.86 | 5.3 | 0.5425 | 33.33 |
| 0.5 | 0.0625 | 16 | 976 | 27.52 | 5.3 | 0.7905 | 36.75 |
| 0.5 | 0.0625 | 128 | 1010 | 27.84 | 5.3 | 0.8227 | 37.32 |
| 0.5 | 18 | 1024 | 1022 | 27.90 | 5.3 | 0.8160 | 36.99 |
| 0.5 | 0.25 | 16 | 784 | 45.81 | 5.4 | 0.5998 | 33.89 |
| 0.5 | 0.25 | 128 | 879 | 39.22 | 5.4 | 0.7066 | 35.11 |
| 0.5 | 0.25 | 1024 | 994 | 40.50 | 5.4 | 0.7330 | 36.69 |
| 0.5 | 0.5 | 16 | 529 | 70.78 | 5.5 | 0.3651 | 26.93 |
| 0.5 | 0.5 | 128 | 639 | 48.41 | 5.4 | 0.5870 | 29.03 |
| 0.5 | 0.5 | 1024 | 913 | 48.81 | 5.4 | 0.6563 | 34.83 |
| 0.5 | 1 | 16 | 16 | 125.21 | 5.5 | 0.0701 | 2.97 |
| 0.5 | 1 | 32 | 32 | 81.37 | 5.5 | 0.2118 | 3.40 |
| 0.5 | 1 | 64 | 64 | 64.04 | 5.4 | 0.3534 | 4.39 |
| 0.5 | 1 | 128 | 128 | 56.64 | 5.4 | 0.4232 | 6.40 |
| 0.5 | 1 | 256 | 251 | 53.53 | 5.4 | 0.4564 | 10.51 |
| 0.5 | 1 | 1024 | 646 | 53.24 | 5.4 | 0.5110 | 24.19 |
| 0.5 | 1 | 4096 | 906 | 54.11 | 5.4 | 0.5315 | 33.33 |
| 0.25 | 0.0625 | 16 | 976 | 24.20 | 5.4 | 0.7908 | 36.75 |
| 0.25 | 0.0625 | 128 | 1010 | 25.48 | 5.4 | 0.8344 | 37.32 |
| 0.25 | 0.0625 | 1024 | 1022 | 25.97 | 5.4 | 0.8312 | 36.99 |
| 0.25 | 0.25 | 16 | 784 | 45.48 | 5.4 | 0.6151 | 33.89 |
| 0.25 | 0.25 | 128 | 879 | 40.08 | 5.4 | 0.6955 | 35.11 |
| 0.25 | 0.25 | 1024 | 994 | 42.56 | 5.4 | 0.7000 | 36.69 |
| 0.25 | 0.5 | 16 | 529 | 79.84 | 5.5 | 0.1846 | 26.93 |
| 0.25 | 0.5 | 128 | 639 | 56.05 | 5.4 | 0.4125 | 29.03 |
| 0.25 | 0.5 | 1024 | 913 | 58.20 | 5.4 | 0.4558 | 34.83 |
| 0.25 | 1 | 16 | 16 | 154.93 | 5.5 | 0.0289 | 2.97 |
| 0.25 | 1 | 32 | 32 | 103.39 | 5.5 | 0.0798 | 3.40 |
| 0.25 | 1 | 64 | 64 | 82.31 | 5.4 | 0.1412 | 4.39 |
| 0.25 | 1 | 128 | 128 | 73.17 | 5.4 | 0.1801 | 6.40 |
| 0.25 | 1 | 256 | 251 | 69.82 | 5.4 | 0.1967 | 10.51 |
| 0.25 | 1 | 1024 | 646 | 71.42 | 5.4 | 0.2491 | 24.19 |
| 0.25 | 1 | 4096 | 906 | 74.39 | 5.4 | 0.2410 | 33.33 |
| 0.125 | 0.0625 | 16 | 976 | 23.16 | 5.4 | 0.8245 | 36.75 |
| 0.125 | 0.0625 | 128 | 1010 | 23.83 | 5.4 | 0.8253 | 37.32 |
| 0.125 | 0.0625 | 1024 | 1022 | 23.89 | 5.4 | 0.8318 | 36.99 |
| 0.125 | 0.25 | 16 | 784 | 50.32 | 5.5 | 0.4867 | 33.89 |
| 0.125 | 0.25 | 128 | 879 | 45.24 | 5.4 | 0.5590 | 35.11 |
| 0.125 | 0.25 | 1024 | 994 | 47.24 | 5.4 | 0.5954 | 36.69 |
| 0.125 | 0.5 | 16 | 529 | 100.22 | 5.5 | 0.0551 | 26.93 |
| 0.125 | 0.5 | 128 | 639 | 72.93 | 5.4 | 0.1461 | 29.03 |
| 0.125 | 0.5 | 1024 | 913 | 75.42 | 5.4 | 0.1834 | 34.83 |
| 0.125 | 1 | 16 | 16 | 227.34 | 5.5 | 0.0104 | 2.97 |
| 0.125 | 1 | 32 | 32 | 160.01 | 5.4 | 0.0174 | 3.40 |
| 0.125 | 1 | 64 | 64 | 131.22 | 5.4 | 0.0259 | 4.39 |
| 0.125 | 1 | 128 | 128 | 118.04 | 5.4 | 0.0299 | 6.40 |
| 0.125 | 1 | 256 | 251 | 113.92 | 5.4 | 0.0337 | 10.51 |
| 0.125 | 1 | 1024 | 646 | 115.17 | 5.4 | 0.0353 | 24.19 |
| 0.125 | 1 | 4096 | 906 | 118.44 | 5.4 | 0.0348 | 33.33 |

*Table 14.* Gen. PPL (↓), entropies and MAUVE (↑) of samples by MDLM trained for 250K steps on OWT.

| $T$ | NFE | Gen. PPL (↓) | Entropy | MAUVE (↑) | Sampling Time (sec) (↓) |
|---|---|---|---|---|---|
| 8 | 8 | 246.70 | 5.6 | 0.0134 | 7.19 |
| 16 | 16 | 109.70 | 5.5 | 0.1353 | 13.81 |
| 32 | 32 | 67.44 | 5.5 | 0.4195 | 27.10 |
| 48 | 48 | 55.96 | 5.5 | 0.5062 | 39.42 |
| 64 | 64 | 51.11 | 5.4 | 0.6123 | 53.48 |
| 128 | 128 | 43.58 | 5.4 | 0.6477 | 106.96 |
| 256 | 251 | 40.44 | 5.4 | 0.6924 | 213.92 |
| 1024 | 657 | 37.15 | 5.3 | 0.7267 | 566.19 |
| 4096 | 907 | 36.48 | 5.3 | 0.7026 | 752.06 |

*Table 15.* Gen. PPL (↓), entropies and MAUVE (↑) of samples by BD3-LMs trained for 250K steps on OWT.

| Block size | $T$ | $T'$ | NFE | Gen. PPL (↓) | Entropy | MAUVE (↑) | Sampling Time (sec) (↓) |
|---|---|---|---|---|---|---|---|
| 4 | 256 | 1 | 512 | 184.86 | 4.00 | 0.0048 | 26.26 |
| 4 | 512 | 2 | 740 | 216.73 | 4.81 | 0.0081 | 37.44 |
| 4 | 1024 | 4 | 968 | 110.22 | 5.14 | 0.0533 | 49.20 |
| 4 | 2048 | 8 | 1124 | 51.92 | 5.22 | 0.3515 | 56.77 |
| 4 | 4096 | 16 | 1180 | 34.93 | 5.24 | 0.6726 | 60.32 |
| 8 | 256 | 2 | 383 | 267.26 | 4.69 | 0.0061 | 20.58 |
| 8 | 512 | 4 | 584 | 170.50 | 5.04 | 0.0168 | 31.44 |
| 8 | 1024 | 8 | 812 | 80.31 | 5.20 | 0.1479 | 42.14 |
| 8 | 2048 | 16 | 951 | 47.16 | 5.22 | 0.5723 | 50.01 |
| 8 | 4096 | 32 | 1051 | 36.34 | 5.25 | 0.6807 | 55.53 |
| 16 | 256 | 4 | 316 | 240.20 | 5.10 | 0.0114 | 19.36 |
| 16 | 512 | 8 | 515 | 112.56 | 5.28 | 0.0971 | 31.17 |
| 16 | 1024 | 16 | 703 | 61.82 | 5.30 | 0.4067 | 43.76 |
| 16 | 2048 | 32 | 881 | 44.06 | 5.29 | 0.6383 | 53.79 |
| 16 | 4096 | 64 | 984 | 37.61 | 5.29 | 0.7248 | 58.82 |

## E.13. Examples of Generated Samples by Models Trained on OWT

---

to be known to the grand jury yet, but it has been explained he could not immediately cause any damage to happen, such as preventing a clean break from someone hacked or creating a fake email. (And again, Hillary's tweet never caused the genesis of the controversy as it was announced, his tweeting violation could easily have changed the course of the matter.)

The Times:

...Senator John McCain doesn's State of the Union...should really have to decide—mossipally—whether they believe to allow a Trump presidency in the first place. There is no situation in which Hillary's campaign could choose to take the matter in a different light.

Except for just one thing what Hillary did in her son's law book there was her "crook of mess" notion.

At this, it is irrelevant today to ask John Podesta to choose someone in Congress so it will be up until the election year, to solve the problems through this simple conceptual framework, which is simple, soft and unhinged and abstract, to create an all too common threadbare" solution.

As an excuse to say, we're okay with the recent DOJ's somewhat unusual way of saying only what the rest of us are thinking in the know.

They knew...the Democratic people of this country set up the proper system to identify.

The legal partner of the campaign and FBI are working with the federal investigation into the Trump campaign for violations of campaign laws under V.W. and Harry Truman.

A joint team star Michael Burnett was allegedly killed after a dog survived a shooting attack by a suspect when cops showed up for a Texas sheriff dog in an afternoon raid on a joint squad and a Texas Border Patrol agent with the animal owner of the state filed charges against Sheriff Edell, Fox and AP reports.Police had been conducting an eight-hour search in order to find the dog dead sometime Monday, during the time of the 100th anniversary of the Golden Gabriel Shooting Act.That was when the Bureau of Investigation allowed the police to close the area after a group of dogs were called to the events, they were, at that time they were found dead.The authorities pulled more than 20 pick-up dogs but were released. Sheriff Edell insisted on using the dogs, given to sheriff's deputies as "an excellent dog.""I'm going further," to deputies and reporters, the sheriff said officers had pulled on the rear door of a drug smuggler and a baggie, which were immediately spotted by private security cameras at the scene.A cat had reportedly appeared on a front door in front of a television screen inside the house in the shooting, Dina Sootoot, who plays...Shanna and A Prairie Winage, were booked for a movie position in the U.S, with a movie star movie and a party dog in their midst.She formerly played Z.A.. During a hour-long episode, on the Texas Weill, he admitted during the interrogation that Mr. Jupp suffered from dramatic seizures that were preceded by a rash.The animal's owner, a doctor, confirmed at the scene that he was overdosed to the illegal drug, a week later was later charged with administering Billing Aid Services. Upon returning to the scene, Fox reported, Mr. Jupp sustained only minor injuries while Mr. Jupp subsequently passed away.Having later moved from Middle Tennessee to South Florida, Mr. Jupp moved to Florida in 2007 on a contractual basis (and with a Green Bay film) and this ultimately landed him in solitary confinement three weeks in a drug row in the desert.

Advertisement

"There is a meaningful escape, zero suffering. Repeat Five, jail! Repeat Five Corners!" -and-Healthy physical health Bill (Public Domain via Getty Images, May17, 2015)

Much of the more recently named London Department of Public Buildings Embley (Flea) made a new investment in approximately $5 Million with the acquisition of a single new office unit comprised of parking spaces and a new 1.6-store five-story studio at the corner of its current office in Coho, London, as part of a three-store-off luxury brick-and-mortar store and several hundred multi-unit studio units, which also include the new airport, under-construction office, reports [LinkedIn.com](http://linkedin.com/) The office is conveniently situated in a building "just over a shopping plaza" and has been "asked for purchase by city officials but not to allow it there one could use."

---

*Figure 24.* An unconditional sample ($L = 1024$) from Eso-LM ($\alpha_0^{train} = 1$) trained for 250K on OWT using inference-time hyperparameters $\alpha_0^{eval} = 1$ and $T = 1024$. This corresponds to an NFE of about 646 and a sampling time of 24.19 seconds per batch of 512 samples. Gen. PPL, entropy and MAUVE are 49.05, 5.3 and 0.4939 respectively.

and for much of its population, Auckland is still of significant interest to both companies.

The public can also afford to copy companies such as Gotham, with offices in New York suburbs such as New York, followed by larger commercial spaces such as London's Empire Bridge and Gotham.

Small Business; but have office space in Auckland; expertise perfect for marketing results.

- Startup advertising work. Put on billboards such as National Grid are ideal for digital marketing work. A flat screen television that got the mind-set

5 hours-by-hour traffic must be in television advertising

The Michaelarinen Gates Shayka-Tin did with his first down in marketing was to Compromise your business, very easy to do.

As the pressure from you surrounds it with work and you're quite healthy, it is still possible to invest just a few dollars a month — your salary or whatever, the money chosen to share the press — via a marketing campaign with FreeMedia.

He said she used to think that the modern internet was paramount: "Follow not one of the most popular people in the world. If they are 50, find a way to have two kids their age. Or, if they are a celebrity, too. The same applies very well, television has that.

It's a way, at least in my opinion, to connect yourself and others and if you sell yourself a bit of confidence.

Read more:

"Can you afford an online lifestyle where you don't know it? Tell your opinion or credibility through information or speech. If you can, you don't need it all the time."

On the other hand, of course, it's a much better thing, for example, to need to offer up a genuine chance to walk with people looking, on camera, and in a hands-on manner of confidence.

Take all of that approach. "You can also try and narrow down the perspective everything that was natural would be easy, which is true if advertisements are not marketed that way.

When advertising that someone named you said was a television advertisement was, when, think of television, the internet was it - and they have no editorial authority; there's no PR for Free Media, but every advertisement is a commercial of their own.

Is that that true?

Yeah. No. Because you've worked in advertising for a very long, maybe for a while. They worked and made friends with their jobs today and you still haven't thought about it at all.

It is a world at best.

For me, from the newspapers, to the advent of the internet, I was constantly looking to appeal to the "new people" that I always connected with, and everyone loved, Twitter.

But now it is still true.

If you haven't all the young author books. Download our free online video guide for your audience for this expert advice.

Read the full interview: Tom Moss covers hundreds of news outlets in Japan and Australia. His work is for letters and written back millions of times. From riding horses to e-reading devices, ATM machines.

For us their ads for these pages already take up more than 1.5 viewers and 30 hours a week. The opportunity to read things and bring you more.

"The internet is never digital for everybody, I would be thrilled if it's the user I've seen before," he said,: "The reality is there is this new age for business is that you're the best as you possibly can and have a feeling they deserve it.

Don't look for cheap TV, and no business editor should pay attention to it.<|endoftext|>In a 2017 television news magazine interview, newly-minted investor Warren Buffett noted that the top income level was increasing at approximately half that amount, but the 2016 American economy "has been operating at a level that most thought would have been a bubble burst."

Buffett said that those years or so, an average American has been earning almost 40 percent in the last quarter, including this for the past five years. That is why, as traditional high earners, businesses must make enormous gains in income tax're worth about 20 percent of their CEO's income. Even those high earners make more.

Advertisement

Advertisement

In the beginning to end, although most sports today make the earnings for all Americans, in the past decades have provided the entertainment revenue, especially at the home entertainment market. Most people have very little disposable income — jobs, living games and using for free. That's their source of income, but they don't provide nearly enough information. So a news article is entitled, "Why Americans are working too hard and don't make more."

Advertisement

Here's the American experiment

*Figure 25.* An unconditional sample ($L = 1024$) from Eso-LM ($\alpha_0^{\text{train}} = 1$) trained for 250K on OWT using inference-time hyperparameters $\alpha_0^{\text{eval}} = 1$ and $T = 64$. This corresponds to an NFE of about 64 and a sampling time of 4.39 seconds per batch of 512 samples. Gen. PPL, entropy and MAUVE are 61.43, 5.4 and 0.4166 respectively.

the modern Thecat race over where this may turn and welcome themselves with their futuristic agility. However, the could be and possibly not at all that backed up. In mentioned, I think the major key issues is balance, ie perhaps the best weapon is a right handed side. While balance - any - always has a presence, a lot of things should never stay like the spine and lean to both legs. Whilst it how wide and, you can also swing wide this making it impossible for a pinch bat guard without weapons. With contrast, the With more than one side, there will be more options than the if it, but allow the the most difficult primary weapon of being in any and balancing out the balanced side. For example, the best players need sharp but when the backup b bat side might be stiff and this be easy. you could swing back then-trod right bat side and a double-beast it and that would work. There for me is a smart side but weird bat side does not bats well So that is always a balance, the bats may not like it but they always might be with one side anyway. bob is skills are learnt and if every bat, has a try out and wrong side to manage to even in and out of the bat. Work to make it and when easy. this is perhaps another issue. to have able to bat in whatever the wrong side is required for a bat that would always last and can always develop into a game especially though trying to have met your bat a bit before is also an issue. With a batter knows their T bat regularly, occasionally you might even pick wiff bat which just means no. I know that it worked but when I had. first try duff bat regularly and return to how they more or less. good

L :There it doesnt seem to work and said it doesn't work the way you want to do it would also work. It showed you had a nice batting set or secondary bat side and would be be great anders to trouble guys with good tiered shots and can I say this from a y bat perspective as I and have both feel as to some level of smart bat. Most of the time, however, I don't think they are a very good bat. they are novice batters and sometimes not the only good bat for even the best right foot bat. On today's point of course, they just have to be third first or second second defensive often on the bat left side, the bat right bat side or on the end of the bat, and have a couple of hands on used to holding the bat bat to the other side of the bat. bat bat is very powerful.

L :So it is working well at best, there is still a little bit of ability to park your bat as expected, but bat won't work with to base error bats and hitting some or-side could still possible. How do you decide to just start the third bats which would make the bat look effective while not very will be one for respond, or R :In a smaller group of slower bat hitters particularly bats u a it is not very weak bat they will think they are playing better with bat than short bat, bat has already developed in terms of bat learning but I do not believe that the bat learned

L : If you are doing bantops, I have people not trying to learn anything. hassleds's bat learning. you should always learn bantops.

L : Well bantops is bat or Obleto bat is bat can get you really into a bat training box instead of being it being training box or be described as a bat session at the light of baters what.

L : They are easy to understand bat training designed bats. ly designing bats are not so and useful but maybe they are better, one being able to bat right hand in right hand defend left left bat bat bat is than batting left hook bat bat is than holding bat bat. at least this difference has started to play out recently for myself. play time between defensive and offensive bat, the do of said bat bat is near when he stole bat from him. but they bat the ball from bat bat to bat bat. against bat position too bats like that, you have attack average bat with short bat. you're going to catch the bat very low there and still with ball kick into bat bat. in certain situations, when a bat bat can be dealt, sometimes. on the end of the bat, maybe third bat, another bat which is third bat, so if bat bats at third bat and the second bat a second bat. then they go to a third bat or hold second bat. they bat handle it better. you can take bat to third second main bat. end of the bat so then bat to your main bat from where bat go second bat. bat, second bat. bat, the bat, on deck. double bats, extra bat, always with bat and bat. no extra bat. less bat bat. A little extra bats"

*Figure 26.* An unconditional sample ($L = 1024$) from BD3-LM ($L' = 4$) trained for 250K on OWT using inference-time hyperparameter $T = 256$ ($T' = 1$). This corresponds to an NFE of about 512 and a sampling time of 26.26 seconds per batch of 512 samples. Gen. PPL, entropy and MAUVE are 184.86, 4.0 and 0.0048 respectively. Note that this sample appears incoherent compared to those with similar sampling time from Eso-LMs.

## E.14. Conditional Generation

We fine-tuned our MDLM and Eso-LMs ($\alpha_0 = 1$) checkpoints trained on OWT for 10K additional steps on XSum preprocessed according to (Meshchaninov et al., 2026). We modulate speed-quality trade-offs by varying diffusion steps $T = \{1, 4, 8, 16, 32\}$ for both methods on 1 A6000 Ada GPU. For reference, FLAN-T5-Small (0.1B-scale) (Wei et al., 2021), an instruction-tuned baseline by Google, achieves rescaled BertScore 0.42. All three methods do not use temperature sampling. As in Figure 4, we see Eso-LMs ($\alpha_0 = 1$) produce higher quality samples across the considered sampling budgets.

*Table 16.* Eso-LMs ($\alpha_0 = 1$) produce higher quality samples than MDLM across the considered sampling budgets on XSum.

| Latency / Time Per Example (sec) | Eso-LM ($\uparrow$) | MDLM ($\uparrow$) |
|---|---|---|
| < 0.1 | **0.19** ($T = 8$) | -0.06 ($T = 1$) |
| < 0.2 | **0.22** ($T = 16$) | 0.16 ($T = 4$) |
| < 0.4 | **0.22** ($T = 32$) | 0.21 ($T = 8$) |

