# OpenReview forum: "Esoteric Language Models: A Family of Any-Order Diffusion LLMs"
_ICML.cc/2026/Conference — ICML 2026 regular_

### Official Review · Reviewer_C8yr · 2026-03-07

**Soundness:** 4
**Presentation:** 2
**Significance:** 3
**Originality:** 4
**Overall Recommendation:** 5
**Confidence:** 3

**Summary:**

This paper introduces Eso-LMs, a new family of language models combining AR and MDM paradigms, which enables exact KV caching for MDMs and one-shot exact likelihood computation along a specific generation order. Eso-LMs build upon an ELBO training objective combining both MDM and any-order AR losses, and require an intricate design of attention masks to guarantee the scope of attention. At inference time, Eso-LMs allow flexible generation schedules and exact KV caching. Eso-LMs interpolate between AR and MDM perplexities, and achieve a nice speed-quality tradeoff.

**Compliance With Llm Reviewing Policy:**

Affirmed.

**Final Justification:**

The authors have fully addressed my question during the rebuttal and I remain my original judgement that this is a good paper worthy of publication.

**Key Questions For Authors:**

1. I don't fully get the point why in the sequential training phase we need double-length inputs. Is it possible to just use one sequence but redesign the attention masks to achieve the same effect (e.g., non-causal attention mask even after shuffling by $\sigma$)? Also, I don't fully understand the same requirement for the evaluation of likelihood.
2. Since at sampling time there's no difference between the MDM phase and the AR phase (i.e., the AR phase is just a special case with $S _ k=1$), I'm wondering what's the main difference between the two training phases. I feel the algorithm can be summarized as **any-order blockwise AR**, i.e., given a generation order $\sigma$ and partition of the sequence into blocks, we train the model to predict all positions in block $k$ simultaneously given the previous blocks $1,...,k-1$ in the order $\sigma$, and always use causal attention masks. KV caching is then a natural consequence of the causal attention masks but with one block delay. Please let me know if I misunderstood something here. Also, can the authors explain the possible reason for the failure of the training when the percentage of MDM loss is less than 50%?
3. In (6), the generation process is first MDM and then AR. Have the authors considered first AR and then MDM? Is this achievable with the same architecture and training procedure? If not, what's the main obstacle?
4. What do the two rows in each loss in Fig. 3 represent? Are all orders (except in the right-bottom) the physical (absolute) order before applying the permutation $\sigma$?

**Limitations:**

Yes.

**Strengths And Weaknesses:**

**Strengths:**

- Improving the generation quality and efficiency of MDMs is an important problem and timely topic, which has been attracting increasing attention in the community.
- The paper is technically sound with detailed theoretical derivations and explanations of the proposed method and architecture. I have checked most of the mathematical derivations and I don't see any major issues. Many of the illustrations are helpful for understanding the proposed method in addition to the mathematical formulas.
- Experiments are well-designed and comprehensive. I highly appreciate the Pareto frontier part between generation quality and speed, which is a very important aspect for evaluating the practical utility of MDMs.

**Weaknesses:**

- The paper is quite dense and technical, requiring a lot of effort to fully understand. I acknowledge that the authors have made efforts to make the presentation more accessible, but I still find some parts hard to follow. See the questions below.
- Eso-LMs interpolate between AR and MDM perplexities, and with smaller $\alpha _ 0$ the performance gets better. However, Eso-LMs still underperform AR models in terms of perplexity. This could be due to the fact that human language is inherently sequential, and it remains an open question whether Eso-LMs can outperform both AR and MDM models in other domains where the sequential nature of the data is less pronounced.

---

> ### Author Rebuttal · Authors · 2026-03-30
>
> We thank the reviewer for the thoughtful assessment that the paper is technically solid and potentially impactful. We are glad the main contributions of Eso-LMs came across clearly: combining AR and MDM, (1) enabling exact KV caching for MDMs, (2) allowing one-shot NELBO estimation, and (3) achieving a strong speed–quality tradeoff.
>
> # Clarification 1: Doubled sequence length for the AR phase
> > Since at sampling time there's no difference between the MDM phase and the AR phase (i.e., the AR phase is just a special case with $S_k=1$), I'm wondering what's the main difference between the two training phases.
> >
>
> In Tab. 11, we observe that Eso-LMs yield the best quality samples when $\alpha_0^{\text{eval}}$ matches  $\alpha_0^{\text{train}}$. But, as noted in L[369-379], Eso-LMs ($\alpha_0^{\text{train}}=1$) trained only with MDM phase can flexibly use different $\alpha_0^{\text{eval}}$ during inference to achieve a competitive Pareto Frontier.
>
> > why in the sequential training phase we need double-length inputs
> >
>
> Training on arbitrary orders requires providing target positions in addition to input positions. Here, each token carries its own position, and each mask token is paired with its target position (e.g., Fig. 2). Not using mask tokens leads to architectural changes to the transformer in prior work (L[418-430]).
>
> Example showing why double-length inputs are needed in AR phase training:
>
> Given:
>
> - Clean sequence ($\mathbf{x}$): `ABCDEFGH`,
> - Randomly masked sequence ($\mathbf{z}_0$): `AB[M1]DE[M2]GH` (`[M1]` and `[M2]` are mask tokens)
>
> During the sequential training phase, the denoising model needs to predict:
>
> 1. `C` for `[M1]`
> 2. `F` for `[M2]`, assuming `C` exists at `[M1]`.
>
> Achieving (1) and (2) with a single forward pass of $z_0$ is very challenging due to the assumption in (2) which requires `[M1]` to be replaced with C. To resolve this, we perform a forward pass of the concatenated sequence `ABCDEFGH AB[M1]DE[M2]GH` , so each later mask `[M_i]` can attend to the clean targets of earlier masks  `[M_{j < i}]`.
>
> This is resolved during sampling with no distribution shift.
>
> ### Note on training speed:
>
> Since only half of each batch is used for sequential training, doubling the sequence length has only a modest impact on training speed (Fig. 16), making it 1.37× slower than MDLM. Eso-LMs remain significantly faster to train than BD3-LMs, which are 2.7× slower than both MDLM and Eso-LMs ($\alpha^\text{train}_0 = 1$).
>
> Note that Figs [18 and 19], we demonstrate that **Eso-LMs trained in the full diffusion mode (which doesn’t experience slowdown) outperforms all prior methods**: MDLM and BD3-LMs on the speed-quality Pareto frontier.
>
> # Comments:
>
> > Eso-LMs still underperform AR models in terms of perplexity.
> >
>
> While Masked Diffusion Models / Eso-LMs possess a worse perplexity than AR models, Eso-LMs are desirable because:
>
> 1. **Eso-LMs achieve an inference-time speed–quality tradeoff unavailable to AR models**. As shown in `Figs. 4 & 17`, they achieve comparable sample quality to AR under similar sampling-time budgets while offering the possibility of faster generation.
> 2. **At 8B scale, masked diffusion models are competitive with or better than AR models on math/science reasoning and coding, and they avoid the reversal curse [1]**, even though they still lag behind AR models in perplexity.
> 3. **Masked diffusion models are better suited for controlled generation tasks** [2].
>
> Because Eso-LMs inherit these advantages from masked diffusion models while significantly reducing sampling time, they are a compelling alternative to AR models.
>
> [1] Nie et al., LLaDa, NeurIPS 2025.
>
> [2] Schiff et al., “Simple Guidance Mechanisms For Discrete Diffusion Models”, ICLR 2025.
>
> > I feel the algorithm can be summarized as **any-order blockwise AR**
> >
>
> That’s pretty accurate!
>
> > can the authors explain the possible reason for the failure of the training when the percentage of MDM loss is less than 50%?
> >
>
> It’s because the datapoints within a training batch that are trained with the MDLM training loss incur higher variance as opposed to the datapoints being trained with the AR loss.
>
> > Have the authors considered first AR and then MDM? Is this achievable with the same architecture and training procedure?
> >
>
> Yes, this is definitely achievable with our current set-up. As the reviewer pointed out earlier, Eso-LMs support arbitrary generation order during sampling.
>
> ## Clarification
>
> > What do the two rows in each loss in Fig. 3 represent?
> >
>
> The two separate rows represent the training data points within a training batch trained with the AR loss (top) and the diffusion loss (bottom) as described in L[266 - 271]. The top most row also features the concatenated input sequence that is fed into the transformer with the dotted arrows depicting the attention mechanism for a given mask token.
>
> > Are all orders (except in the right-bottom) the physical (absolute) order before applying the permutation?
> >
>
> Yes

---

> > ### Author Rebuttal · Reviewer_C8yr · 2026-04-02
> >
> > I thank the authors for the response, which has fully addressed my questions. For the argument concerning double-length inputs, I later realized that this follows from the block diffusion paper, which leveraged similar strategies during training. This is a quite clever design without architectural changes, as the authors have motioned. Also, I generally agree with your point on why Eso-LMs are desirable and hope to see its further applications in the future. I maintain my positive rating on this paper.

---

> > > ### Author Response · Authors · 2026-04-06
> > >
> > > We thank the reviewer for recognizing the engineering efforts in our paper and for their thorough engagement during the rebuttal process. We look forward to seeing how the community builds on this work.

---

### Official Review · Reviewer_3aNs · 2026-03-09

**Soundness:** 2
**Presentation:** 2
**Significance:** 3
**Originality:** 3
**Overall Recommendation:** 4
**Confidence:** 3

**Summary:**

The paper presents a novel LLM model called Eso-LM that mixes masked diffusion and auto-regressive architectures. The authors claim that this interpolation helps to overcome some of the biggest limits of masked diffusion models: the intractable likelihood computation and the possibility to use KV caching.

**Compliance With Llm Reviewing Policy:**

Affirmed.

**Final Justification:**

The paper's idea and methodology are very sound. While the paper lacks any proof that their architecture beats previous models in maximal generation quality, the fact that it beats other diffusion-AR hybrids at equal NFEs is valuable.

**Key Questions For Authors:**

I would like the authors to answer to my doubts in the **Weaknesses** section above, i.e.,
- How do the authors handle the any-order causal prediction without giving the model additional information on the next-token position?
- Why do you need to double the sequence length for the AR batches during training? And does this training-inference discrepancy limits the performance of the model?
- Why should one use Eso-LMs over standard AR models, since they exhibit lower perplexity and longer training times?

I might have misunderstood parts of the architecture, so I will be happy to increase my score if my concerns are addressed.

**Limitations:**

Yes.

**Strengths And Weaknesses:**

**Strengths**

The paper's topic is very timely. The idea of mixing diffusion and auto-regressive architectures is very relevant, as many recent works in this area try to overcome limits of pure MDM architectures by using them in conjunction with AR architectures. This paper takes a different, novel route then other models by splitting the generation process into a diffusion phase, which generates/unmasks some of the tokens first, followed by an AR phase, which completes the generation of the remaining tokens. The possibility to use KV caching is especially welcome, as the main advantage of diffusion models would be faster inference enabled by parallel generation, but this advantage does not materialize in practice due to the lack of KV caching capability. Furthermore, I like the Importance-Weighted bounds on the exact likelihood, which I think are an important contribution that risks to be overlooked due to the density of the paper's presentation. I also appreciate the inclusion of the (very clear) code for the important new components of the architecture.

**Weaknesses**

- I think that the paper's presentation can be vastly improved. The paper is very dense, and the topics are presented in a shuffled manner. For example, technical details of the Eso-LM model architecture are randomly distributed in Sec. 3 (the parametrized distribution (6), the loss function (7), part of the sampling strategy and a hint of the KV caching implementation in Sec. 3.2), Sec. 4 (details on the attention mechanism and more details on sampling), and several Appendices. Both Sec. 3 and 4 include subsections "Training" and "Sampling"; it is not clear why they cannot be merged together.
- Probably because of the density and the confusion of the presentation, I find it very hard to actually parse and understand all the architectural choices of the authors. More specifically, I have the following doubts:
  - It seems to me that, deep down, the main technical change introduced by Eso-LM is the use of an any-order type of architecture, that predicts tokens in a causal manner but in a shuffled order. If that is the case, then I don't understand how the authors do not incur in the same issues of any-order models like XLNet and $\sigma$-GPT. The issue is simple: if the model has to predict tokens in a randomly-shuffled order, it is fundamental, at every instant, to provide the model with the additional information of the _next position_ to predict. Without this information, the model's prediction for the next token would be the same no matter what is the next token's position. XLNet addresses this problem by implementing a two-stream attention module. $\sigma$-GPT modifies each token's positional embedding to include both the current and the next positions. I don't understand what do the authors do to address this issue with Eso-LMs.
  - I also don't understand why there is a need to double the sequence during training for part of the training batches devoted to AR. This seems to be a very important point since it affects training speed and it causes a discrepancy between training and inference, whose effect in the final performance is unclear.
- Another important point that the paper does not address is: why to use Eso-LM, which seems to essentially be an Any-Order AR model, over simple AR models? All the results in the paper seem to imply that Eso-LMs are always worse than AR models, matching them only in full-AR mode ($\alpha_0 = 1$) but with slower training speed due to the sequence doubling.

---

> ### Author Rebuttal · Authors · 2026-03-30
>
> We appreciate the reviewer's recognition that our contributions are timely and novel: bridging AR-MDM, exact KV caching, and exact likelihood via importance weighted bounds. We now address concerns.
>
> # Clarification 1: positional information for target tokens
>
> The reviewer correctly highlights issues with prior any-order methods, such as XLNet and SigmaGPT, which require significant architectural modifications to the transformer.
>
> Unlike XL-Net and sigma-GPT, our method does not modify the Transformer architecture. Each token carries its own position ID, and each mask token is paired with the target position ID. In full diffusion training, we simply: (1) shuffle tokens together with their position IDs, (2) place clean tokens on the left and masked tokens on the right, and (3) apply standard causal attention. This allows the model to predict tokens at arbitrary positions without architectural changes. **For example, in Fig. 2, the two mask tokens carry the position IDs 1 and 8 and are denoised to A and H.**
>
> Our approach is much more efficient:
>
> 1. Unlike sigma-GPT, it is compatible with RoPE and does not require extra parameters for target-position embeddings
> 2. Unlike XL-Net, it avoids two-stream attention, which is roughly twice as expensive as causal attention. Note that in Figs [18 and 19], we demonstrate that **Eso-LMs trained in the full diffusion mode outperforms all prior methods**: MDLM and BD3-LMs on the speed quality Pareto frontier. Thus, XL-Net is slower to train than our method.
>
> # Clarification 2: Doubled sequence length for the AR phase
>
> Below we provide a concrete example to illustrate **why this technique is necessary for the sequential-phase training** of Eso-LMs and why it **does not introduce any distribution shift** between training and sampling (no concatenation happens during sampling).
>
> Given:
>
> - Clean sequence ($\mathbf{x}$): `ABCDEFGH`,
> - Randomly masked sequence ($\mathbf{z}_0$): `AB[M1]DE[M2]GH` (`[M1]` and `[M2]` represent mask tokens)
>
> During the sequential training phase, the denoising model needs to predict:
>
> 1. `C` for `[M1]`
> 2. `F` for `[M2]`, assuming `C` exists at `[M1]`.
>
> Achieving (1) and (2) with a single forward pass of $z_0$ is very challenging due to the assumption in (2) which requires `[M1]` to be replaced with C. To resolve this, we perform a forward pass of the concatenated sequence `ABCDEFGH AB[M1]DE[M2]GH` , so each later mask `[M_i]` can attend to the clean targets of earlier masks  `[M_{j < i}]`.
>
> > This seems to be a very important point since **it affects training speed** and **it causes a discrepancy between training and inference**, whose effect in the final performance is unclear.
> >
>
> ### Note on training-inference discrepancy:
>
> **During sampling, in the AR generation phase, the above conflict is naturally resolved because all the mask tokens are denoised auto-regressively from left to right.** A visual depiction is provided in `Fig. 2`.
>
> ### Note on training speed:
>
> Since only half of each batch is used for sequential training, doubling the sequence length has only a modest impact on training speed (Fig. 16), making it 1.37× slower than MDLM. Eso-LMs remain significantly faster to train than BD3-LMs, which are 2.7× slower than both MDLM and Eso-LMs ($\alpha^\text{train}_0 = 1$).
>
> Note that in Figs [18 and 19], we demonstrate that **Eso-LMs trained in the full diffusion mode (which doesn’t experience slowdown) outperforms all prior methods**: MDLM and BD3-LMs on the speed-quality Pareto frontier. See Remark 2 `L[376-380]` for details.
>
> # Clarification 3: Why Eso-LMs over Autoregressive Models
>
> This a great question! Here are the reasons as to why Eso-LMs are desirable:
>
> 1. **In particular, Eso-LMs achieve an inference-time speed–quality tradeoff unavailable to AR models**. As shown in `Figs. 4 & 17`, they achieve comparable sample quality to AR under similar sampling-time budgets while offering the possibility of much faster generation.
> 2. **At 8B scale [1], masked diffusion models are competitive with or better than AR models on math/science reasoning and coding, and they avoid the reversal curse**, even though they still lag behind AR models in perplexity.
> 3. **Masked diffusion models are better suited for controlled generation tasks** [2].
>
> Because Eso-LMs inherit these advantages from masked diffusion models while significantly reducing sampling time, they are a compelling alternative to AR models.
>
> [1] Nie et al., “Large Language Diffusion Models”, NeurIPS 2025.
>
> [2] Shiff et al., “Simple Guidance Mechanisms For Discrete Diffusion Models”, ICLR 2025.
>
>
> # Comments on writing
>
> We appreciate the reviewers’ feedback on the writing.
>
> Section 3 is intended to describe the algorithm independently of its specific transformer-based instantiation. Accordingly, the discussion of KV caching currently in Section 3 would be better placed in Section 4, which focuses on transformer-specific details.
>
> We will make this change in the next revision of the paper.

---

> > ### Author Rebuttal · Reviewer_3aNs · 2026-04-03
> >
> > I thank the authors for their thorough replies, now I have a much clearer understanding of the general mechanism of the model. I believe that this is an interesting work, therefore I increased my score to 4.
> >
> > On a side note, I would advise the authors to mention the AR generation phase in Sec. 4.2 on sampling. As it is right now, it seems that the model is only in diffusion phase at inference.

---

> > > ### Author Response · Authors · 2026-04-06
> > >
> > > We thank the reviewer for acknowledging that our work is interesting and proactively engaging with us throughout the rebuttal. Yes, adding AR generation to sec 4.2, makes sense and we'll update it in the next revision.

---

### Official Review · Reviewer_ZDgE · 2026-03-12

**Soundness:** 3
**Presentation:** 3
**Significance:** 2
**Originality:** 2
**Overall Recommendation:** 4
**Confidence:** 5

**Summary:**

This paper introduces esoteric language models (Eso-LMs), which interpolate between masked diffusion models (MDMs) and autoregressive models (ARMs). Eso-LMs use two generation phases: first, a diffusion phase that generates a certain fraction of the sequence in an any-order manner, and second, a sequential phase that fills in the remaining tokens left-to-right. The authors argue that Eso-LMs combine the advantages of both MDMs and ARMs: (1) parallel generation, (2) exact KV caching, (3) single-pass likelihood, and (4) exact likelihood.

**Compliance With Llm Reviewing Policy:**

Affirmed.

**Final Justification:**

The contribution is good, and the additional experiments presented during the rebuttal further strengthen the paper.

**Key Questions For Authors:**

Overall, I would appreciate the authors’ response to the weaknesses above, especially **W1**. In particular:

1. In Fig. 4, for the case $\alpha_0^{train}=1$, the lower-left data point seems to correspond to a relatively large $\alpha_0^{eval}$, i.e., a setting very close to MDLM. If so, why does it perform better than MDLM? Intuitively, causal attention seems to have access to less information, so I would expect generation quality to be worse, not better. This part is hard to reconcile intuitively.
2. If the explanation is that MDLM can perform multiple sampling events within one time step, what would happen if we forced it to perform only one sampling event per time step? For example, the first-hitting sampler [3], which samples one random position at a time, was shown to induce theoretically the same distribution. With such a sampler, would the performance of Eso-LM and MDLM become similar in this setting?

[3] Zheng et al., *Masked Diffusion Models are Secretly Time-Agnostic Masked Models and Exploit Inaccurate Categorical Sampling*, ICLR 2025

**Limitations:**

Yes

**Strengths And Weaknesses:**

## Strengths

1. Although the KV caching mechanism of Eso-LM is fairly complex, its design is at least well aligned with the goal of building a single unified KV cache for both MDM-style and AR-style generation.
2. The method achieves a strong speed-quality Pareto frontier (Fig. 4, Fig. 17). In low-NFE settings, it also appears to generate more diverse and higher-quality samples than BD3LM, which similarly interpolates between MDMs and ARMs.
3. While somewhat over-emphasized at times, the findings and remarks are presented clearly and make the paper relatively easy to follow.

## Weaknesses

1. The claims around **single-pass likelihood** and **exact (asymptotic) likelihood** feel somewhat overstated, and their practical value remains unclear.
- The quantity $\mathcal{L_{AO}^K}$ (exact asymptotic likelihood) still requires Monte Carlo sampling and is still a bound, much like the MDM NELBO. Even if it is tighter than the MDM NELBO, I am not sure what practical advantage this provides. Although $\mathcal{L}_{AO}^K \to p(x)$ as $K \to \infty$, there is already prior work [1] showing, both theoretically and empirically, that MDMs can also estimate likelihood via Monte Carlo sampling.
- More importantly, beyond the theory, it is unclear what practical benefit $\mathcal{L}_{AO}^K$ provides. For a given sequence $x \in \mathcal{V}^L$, an ARM can compute $p(x)$ in a single forward pass. I believe this property is valuable in downstream settings such as RL. By contrast, this paper mainly states that exact likelihood estimation is possible, but it also requires MC sampling, and the authors do not clearly explain what concrete practical advantage this offers.
- To really support the usefulness of exact likelihood, I think the paper would need results on a downstream task involving conditional likelihood estimation, such as HellaSwag, showing that under matched train/inference FLOPs it outperforms the MDM NELBO. However, the paper explicitly states in line 310 that “downstream tasks are left for future work.”
- The variance of the $\mathcal{L}_{AO}^K$ estimator is also not reported. For the same number of Monte Carlo samples $K$, is its variance actually lower than that of the MDM NELBO estimator?
- Relatedly, even if Theorem 3.1 is theoretically correct, without this kind of empirical validation, the claim in lines 349–350 that “this is the first work to report exact likelihoods for MDMs” feels somewhat too strong.
- Finally, I am also unsure about the utility of **single-pass likelihood**. It only gives the single-pass likelihood for one particular permutation $\sigma$, and I am not sure where this is especially useful. Section 3.3 mentions GRPO as an example, but in GRPO the model fundamentally generates a sequence through $L$ forward steps and optimizes likelihood over that process. Since the likelihoods of all forward steps are already available, I do not clearly see what extra advantage the single-pass likelihood of a particular pair $(x,\sigma)$ offers over standard MDMs in that setting.

2. The results across different $\alpha$ values are combined in a way that makes the method look broadly SOTA, but the benefit of each individual $\alpha$ setting is less clear.
- When $\alpha <= 0.25$, the model is already very close to an ARM, yet it is actually worse than ARM in performance, so it is unclear what advantage it provides. In general, unless the model is exactly ARM ($\alpha=0$), its test PPL is worse than ARM. Even in Figure 4 (Mauve per duration), when $\alpha \le 0.25$, the quality does not surpass ARM while the sampling speed is nearly the same. In this regime, the method seems to inherit neither the benefits of MDMs nor those of ARMs.
- When $\alpha=0.5$ or $\alpha=1.0$, the test PPL is even worse than MDLM. Table 7 reports zero-shot PPL, and on all seven benchmarks the results are consistently worse than MDLM. I agree that PPL is not a perfect measure when sampling time is finite, but I still think it is difficult to support a broad SOTA claim based mainly on Mauve score and generative PPL alone. Also, the ablation called Eso-LM (A) is not actually the main technique emphasized by the paper, so whether it matches MDLM in PPL seems of limited significance.
- In Remark 2, the authors state that setting $\alpha_0^{train}=1$ is sufficient. But if so, then the method seems to differ from the concurrent work Any-order GPT [2] mainly only in its inference-time technique.
- Taken together, these points make it unclear how generalizable and scalable the method really is. The mechanism itself and its attention pattern are already quite complex, and performance appears to depend heavily on the hyperparameter $\alpha$.

[1] Jeon et al., *Information-Theoretic Discrete Diffusion*, NeurIPS 2025
[2] Xue et al., *Any-order GPT as Masked Diffusion Model: Decoupling Formulation and Architecture*, ICML Workshop 2025

---

> ### Author Rebuttal · Authors · 2026-03-30
>
> We thank the reviewer for the detailed review.
>
> # Concern 1: Practical utility of the single-pass likelihood estimator
>
> Anonymous work [1] demonstrates the practical utility of the single-pass estimator at 8B scale. In [1], d2 converts Llada (8B) [3] to an Eso-LM and applies RL post-training. Relative to d1 [2], the single-pass estimator yields a much larger gain.
> | Method  | Pre-RL| Post-RL |
> | --- | --- | --- |
> | d1 [2] | 59% | 60% |
> | **d2 w/ single-pass estimator** | 59% | **64%** |
>
> Current methods [2] do not compute the likelihood of a trajectory because they ignore generation order, unlike our method. This explains the improved RL performance of our single-pass NELBO estimator.
>
> [1] Anonymous et al.,  “d2: Improved Techniques for Training Reasoning Diffusion Language Models” [Caution: the arxiv version of this work cites the de-anonymized version of our work]
>
> [2] d1: Scaling Reasoning in Diffusion Large Language Models via Reinforcement Learning
>
> [3] Nie et al, 2025  Large Language Diffusion Models
>
>
> # Concern 2: Utility of Exact Likelihood Estimation
>
> Exact likelihood is central to density modeling. For MDMs, perplexity is typically only a bound on likelihood, whereas for AR models it is exact. Exact likelihood therefore enables fairer comparison and supports applications such as compression [1] and adversarial example detection [2], and semi-supervised learning.
>
> [1] Cover and Thomas. Data compression, 2005
>
> [2] Song et al., 2017 “Pixel-defend”
>
> > Evaluation on HellaSwag
>
> **[New Exp]** We report HellaSwag validation accuracy for Eso-LMs ($\alpha_0=1$) trained on OpenWebText at 110M with different likelihood estimators. Our estimators consistently outperform the MDLM NELBO formulation.
>
> | Model | Acc (%) $\pm$ CI |
> | --- | --- |
> | MDLM NELBO  | 26.0 $\pm 0$ |
> | $L_{AO}^{K=1}$ (OURS) | **27.4** $\pm 0$ |
> | $L_{AO}^{K=10}$ (OURS) | **28.1** $\pm 0$ |
>
> CI: 95% confidence interval. $\mathcal{L}_{AO}^{K=1}$ has the same expectation as the MDM NELBO but much lower variance `L[211-219]`.
>
> # Concern 3: Eso-LMs ($\alpha_0=1$) worse likelihood and zero-shot PPL than MDLM
>
> As noted in L[358-360], this gap comes from architectural changes needed to enable KV caching in Eso-LMs. These changes make each denoising step much faster, but at the cost of worse likelihood. **Eso-LMs are preferable over MDLM because they generate higher quality samples for a given time budget**, as shown in Figs. 4 & 17.
>
> > why does it perform better than MDLM?
>
> See above. Also see Tab 11 & 12 (Eso-LMs point use a higher T than MDLM point; 16 vs 8).
>
> # Clarifications
>
> > Variance of single-pass likelihood
>
> As discussed in L[211-219], it should have lower variance because a single order $\sigma$ captures all $L$ latents along one diffusion trajectory. We verify this empirically on one OWT validation datapoint over 100 trials.
>
> | Estimator | MC samples | NELBO - Mean | Std Dev |
> | --- | --- | --- | --- |
> | MDLM NELBO | 10 | 3.25 | 0.56 |
> |  $L^{K=1}_{AO}$ (OURS) | 1 | 3.28 | **0.03** |
>
> > Connection to Any-order GPT
>
> AO-GPT is a special case of Eso-LMs when $\alpha_0=1$. Unlike AO-GPT, Eso-LMs can interpolate between AR and diffusion regimes and doesn't require tweaking the transformer architecture as discussed in Related works.
>
> > Eso-LMs is complex to implement
>
> The modifications relative to MDLMs are minimal and can be implemented in about a dozen lines of code [Fig 12, L1240-1253].
>
> > Comparison to [1] Information-Theoretic Diffusion Model
>
> **The claim in [1] that the MDLM NELBO equals the exact likelihood is incorrect**. Marginalizing over path measures yields only an upper bound, not the equality stated in Sec. C.1 of [1].
>
> **[New Exp]** The small-scale empirical validation experiments on DNA data in [1] shows that the gap between the NELBO and NLL is negligible. Similarly, we design a first-order Markov process with |V|=254 and L=30, where the ground-truth NLL is 1.73. We train an AR transformer, MDLM, and Eso-LMs -- all achieve essentially the same value.
>
> |  | Val. BPD |
> | --- | --- |
> | Ground truth | 1.73 |
> | AR  | 1.73 |
> | MDLM NELBO / [1] | 1.73 |
> | $L_{AO}^{K=100}$ | 1.73 |
>
> However, as verified in `Tab 1`, **this tightness does not persist at scale**.
>
> > Advantage of Eso-LMs over AR? ... The results across different values are combined in a way that **makes the method look broadly SOTA**
>
> Eso-LMs provide a speed-quality tradeoff unavailable in AR models. To study this tradeoff, we vary $\alpha_0$ during inference; see L[200-205].
>
> **Comment on metrics:** GenPPL and Mauve are the only known metrics in the literature that capture sample quality and generation at 110M scale.
>
> > one sampling event per time step
>
> **[New Exp]** We include results for MDLM with a first-hitting sampler and compare it with EsoLM ($\alpha_0^{train}=1$). **Our method produces similar-quality samples to MDLM while being much faster.**
>
> | Method | MAUVE | Sampling time |
> | --- | --- | --- |
> | MDLM | 0.70 | 752.06 |
> | EsoLM (OURS) | 0.70 | **36.75** |

---

> > ### Author Rebuttal · Reviewer_ZDgE · 2026-04-03
> >
> > Thank you for the detailed rebuttal. My major concern was whether the likelihood estimation of Eso-LM, and, more broadly, Any-order GPT-style methods, is actually more reliable than that of MDM. The authors did not provide empirical evidence for this in the initial submission, but the extensive additional experiments in the rebuttal convincingly demonstrated its usefulness. I would appreciate it if these results could be incorporated into the main paper in the event of acceptance. I will raise my score to 4.
> >
> > I have one final question after reading the response. If MDM were evaluated using $L_{AO}^K$, would its likelihood accuracy become comparable to that of Any-order GPT-style methods? In fact, the MDM NELBO may inherently suffer from high variance because its scaling term becomes excessively large when $t$ is close to 0. However, considering E.1 in Kim et al. [1], Prop 3.2 in Zheng et al. [2], and the widely chosen time-agnostic network for MDLM, it seems that MDM should also be theoretically capable of using $L_{AO}^K$ as a likelihood estimator, by just treating learned $x_\theta^{(i)}(x_{i-1},t)$ as $p_\theta(x_i|x_{i-1})$. The identical Mauve scores under the first-hitting sampler added in the rebuttal indirectly might suggest that the two may indeed be similar in this regard. From an engineering perspective, I understand that this may not be practically meaningful, since Any-order GPT-style methods permit KV caching, whereas applying $L_{AO}^K$ to MDM would require an additional factor of $O(L)$ in compute. Still, in my view, the main difference between Any-order GPT-style methods and MDM-style methods lies in whether they use a full-attention map, and it would be very helpful for future researchers if the authors could explain in more detail whether this architectural difference would lead to a gap in likelihood performance, and if so, why. I fully understand the authors’ contribution and the role of $L_{AO}^K$ in Eso-LM, so regardless of the outcome, I will not lower my score.
> >
> > [1] Kim et al., "Train for the Worst, Plan for the Best: Understanding Token Ordering in Masked Diffusions."
> >
> > [2] Zheng et al., "Masked Diffusion Models are Secretly Time-Agnostic Masked Models and Exploit Inaccurate Categorical Sampling."

---

> > > ### Author Response · Authors · 2026-04-06
> > >
> > > We thank the reviewer for the genuine interest in our work and proactively engaging with us throughout the rebuttal.
> > >
> > > # **New Experiment**
> > >
> > > > If MDM were evaluated using L^K_AO, would its likelihood accuracy become comparable to that of any-order GPT style methods?
> > > >
> > >
> > >
> > > As the reviewer correctly points out, our proposed Importance weighted likelihood bound $\mathcal{L}^{K}_{AO}$ (`Eq. 8`) in principle can be applied to MDLM, and indeed the computation cost would require an additional factor O(L) making it intractable for long sequence length. We demonstrate this on LM1B with a sequence length $L=128$:
> > >
> > > |  | MDLM |  | Eso-LM ($\alpha_0=1$)  |  |
> > > | --- | --- | --- | --- | --- |
> > > | K in $\mathcal{L}^{K}_{AO}$ | IW Likelihood Bound $(\downarrow)$ | Eval Time per batch (s; $\downarrow$) | IW Likelihood Bound $(\downarrow)$ | Eval Time per batch (s; $\downarrow$) |
> > > | 1 | 31.19 | 4.80 | 37.53 | 0.04 |
> > > | 10 | 28.66 | 48.01 | 34.49 | 0.38 |
> > > | 20 | 28.26 | 96.03 | 33.94 | 0.75 |
> > > | 50 | 27.85 | 241.15 | 33.37 | 1.88 |
> > > | 100 | 27.57 | 481.25 | 33.00 | 3.75 |
> > > | 1000 | 26.82 | 4795.23 | 32.09 | 37.50 |
> > >
> > > Both methods are evaluated on the same hardware with a batch size 32.
> > >
> > > ### Key Takeaways:
> > > 1.	**These results demonstrate the effectiveness of our proposed importance-weighted (IW) likelihood bound**, which consistently improves the likelihood estimates for both MDLM and Eso-LMs.
> > >
> > > 2.	MDLM achieves an importance-weighted likelihood of `26.82`, compared with `31.65` for Eso-LMs with $\alpha_0 = 1$. **As discussed in the paper, this trend is expected** because we achieve faster inference with Eso-LMs than MDLM by replacing MDLM’s bidirectional attention with less expressive causal attention. **Although this worsens perplexity, Eso-LMs consistently generate higher-quality samples within a fixed sampling-time budget** (`Fig. 4`).
> > >
> > > 3.	**Computing the importance-weighted likelihood is tractable only for Eso-LMs**. Evaluating `Eq. 8` requires only $K$ forward passes for Eso-LMs (`Fig. 10`), whereas MDLM requires $L \times K$ forward passes.

---

### Official Review · Reviewer_A1A2 · 2026-03-12

**Soundness:** 3
**Presentation:** 3
**Significance:** 3
**Originality:** 3
**Overall Recommendation:** 4
**Confidence:** 3

**Summary:**

Eso-LMs proposes a hybrid AR-MDM language modeling framework that exploits the equivalence between masked diffusion models and any-order autoregressive models to replace bidirectional attention with causal attention over shuffled token sequences. This enables, for the first time, exact KV caching during the diffusion phase while preserving parallel generation. The framework also yields an asymptotically exact likelihood estimator for MDMs via an importance-weighted bound. Empirically, Eso-LMs achieves a new Pareto frontier on speed vs. generation quality and offers substantial latency improvements at long contexts.

**Compliance With Llm Reviewing Policy:**

Affirmed.

**Key Questions For Authors:**

See weakness

**Limitations:**

yes

**Strengths And Weaknesses:**

Strengths:

The causal attention + random permutation design is technically clean and well-grounded in the AO-ARM equivalence; the NELBO derivation in Appendix B supports it rigorously.

The IW likelihood bound (Theorem 3.1) is a meaningful theoretical contribution with direct practical value for RL finetuning of diffusion LMs.

Ablations over α₀, κ, T, and the Eso-LMs (A) variant are thorough, and long-context latency numbers are clearly presented with variance reported.


Weaknesses

1. Table 4 shows Eso-LMs performing worse than MDLM, BD3-LMs, and AR on every tested dataset (PTB, Wikitext, LM1B, Lambada, AG News, Pubmed, Arxiv) across all α₀ values. The paper dismisses the in-distribution PPL gap by appealing to efficiency, but Table 4 is a pure quality comparison with no speed tradeoff involved. Does the causal attention design structurally hurt generalization, and how does the paper expect this to resolve at scale?

2. All experiments use unconditioned OWT samples scored by Gen. PPL and MAUVE. The paper motivates Eso-LMs for chat systems and controllable generation, yet defers all downstream tasks to future work. Even one conditional task , summarization or code completion , would make the practical claims more credible.

3. Four concurrent methods targeting the same KV caching problem are cited (Hu et al., Wu et al., Ma et al., Xue et al.), and the paper asserts Xue et al. is a special case of Eso-LMs — but this is never shown empirically. Direct comparisons under the same hardware and sequence length condiions are needed to substantiate the SOTA claim.

---

> ### Author Rebuttal · Authors · 2026-03-31
>
> We thank the reviewer for the thoughtful assessment. We address the concerns below.
>
> ## Concern 1: Eso-LMs always perform worse in all zero-shot likelihood tasks
>
> This is **not quite accurate.**  As shown in Table 4 (included below), on **4 out of 7** (PTB, Wikitext, LM1B, AG News) unseen datasets, **the** **ordering of AR < Eso-LMs (0.125) < MDLM perplexities** (lower the better) is preserved, **consistent with the OWT validation results**. (All models were trained on OpenWebText (OWT).)
>
> | Model | OWT Val. | PTB* | Wikitext* | LM1B* | Lambada | AG News* | Pubmed | Arxiv |
> | --- | --- | --- | --- | --- | --- | --- | --- | --- |
> | AR | 17.90 | 82.00 | 26.54 | 52.14 | 51.69 | 55.53 | 49.49 | 44.98 |
> | Eso-LMs (α₀ = 0.125) | 21.87 | 97.46 | 35.65 | 60.11 | 69.13 | 65.26 | 65.27 | 57.4 |
> | MDLM | 25.19 | 100.17 | 37.08 | 70.79 | 52.06 | 71.37 | 46.51 | 40.21 |
> | BD3-LMs (L′ = 16) | 23.57 | 95.87 | 32.88 | 65.11 | 50.05 | 61.68 | 43.41 | 40.13 |
>
> NOTE: Zero-shot results are included for completeness. As discussed in L[337–346], **perplexity-based evaluations are insufficient because they do not capture sampling time, which is the primary focus of this paper**. Nevertheless, **Eso-LMs remain preferable to MDLM and BD3-LMs**, **as they achieve higher sample quality under a fixed time budget** (see Figs. 4 and 17).
>
> > Does the causal attention design structurally hurt generalization, and how does the paper expect this to resolve at scale?
> >
>
> Anonymous work [1] scales Eso-LMs to 1.7B parameters, showing that they achieve competitive accuracy with MDLM on zero-shot likelihood tasks while **dominating the speed–quality Pareto frontier for unconditional generation at up to billion-parameter scale.**
>
> | Method | ARC-e | BoolQ | OBQA | PIQA | RACE | SIQA |
> | --- | --- | --- | --- | --- | --- | --- |
> | Chance | 24.7 | 50.4 | 26.6 | 51.6 | 24.2 | 32.2 |
> | SMDM-1B [2] | 37.4 | 61.5 | 27.0 | 60.3 | 29.3 | 37.9 |
> | LLaDa-8B-Base [3] | - | - | - | 74.4 | - | - |
> | Scaling Beyond Mask Diffusion Models: |  |  |  |  |  |  |
> | AR-1.7B (Autoregressive) | 72.7 | 71.9 | 40.4 | 78.1 | 36.2 | 41.9 |
> | MDLM-1.7B (Masked Diffusion) [4] | 50.5 | 62.8 | 32.0 | 62.2 | 34.7 | 39.2 |
> | **Eso-LM-1.7B (Interpolating Diffusion)** | 46.0 | 53.4 | 29.6 | 55.6 | 26.1 | 36.1 |
>
> [1] Anonymous et al., Scaling Beyond Masked Diffusion Language Models [caution: the arxiv version of this work cites the de-anonymized version of our work]
>
> [2] Nie et al., Scaling up Masked Diffusion Models on Text
>
> [3] Nie et al., Large Language Diffusion Models
>
> [4] Sahoo et al., Simple and Effective Masked Diffusion models
>
> ## Concern 2: Lack of conditional generation tasks
>
> Thanks for mentioning this. We will include the following experiment in the revised version.
>
> **[New Exp]** **Eso-LMs** ($\alpha_0=1$) **compared to MDLM on summarization task XSum, following data preparation in [1].**
>
> We fine-tuned our MDLM and Eso-LMs ($\alpha_0=1$) 100M-scale checkpoints trained on OWT for 10K additional steps on XSum. We modulate speed-quality trade-offs by varying diffusion steps T={1, 4, 8, 16, 32} for both methods on 1 A6000 Ada GPU. For reference, FLAN-T5-Small (100M-scale) [2], an instruction-tuned baseline by Google, achieves rescaled BertScore 0.42.
>
> Similar to Figure 4, we see **Eso-LMs ($\alpha_0=1$) produce higher quality samples across every sampling budget.**
>
> | Latency / Time Per Example (s) | Eso-LM ($\uparrow$) | MDLM ($\uparrow$) |
> | --- | --- | --- |
> | <0.1 | **0.19 (T=8)** | -0.06 (T=1) |
> | <0.2 | **0.22 (T=16)** | 0.16 (T=4) |
> | <0.4 | **0.22 (T=32)** | 0.21 (T=8) |
>
> [1] Meshchaninov et al., Cosmos, 2025
>
> [2] Chung et al., Scaling Instruction-Finetuned Language Models, 2022.
>
> ## Clarification 1: Lack of empirical comparison with prior work
>
> Methods [1, 2] rely on block-wise decoding in MDLM, which introduces two key limitations:
>
> 1. Each iteration requires a forward pass over the full context, making it significantly slower than BD3-LMs and Eso-LMs.
> 2. Block-wise decoding inherits the drawbacks of BD3-LMs, where sample quality drops sharply as the number of function evaluations (NFEs) decreases (Fig. 4).
>
> Therefore, we do not expect [1, 2] to outperform BD3-LMs in terms of sampling speed or accuracy especially at long context ($L=10240$; Table 10) where BD3-LMs are 5x slower and produce worse quality samples.
>
>
>
> The dKV-Cache-Decode method introduced in [3] is another approximate KV caching approach for MDLM that supports random decoding orders. However, it refreshes the KV cache for all tokens in the context every N steps (typically N = 2 or 4). This design becomes highly restrictive at long context lengths.
>
> [1] Hu et al., FlashDLM: Accelerating diffusion language model inference via efficient kv caching and guided diffusion
>
> [2] Wu et al., Fast-dLLM: Training-free Acceleration of Diffusion LLM by Enabling KV Cache and Parallel Decoding
>
> [3] Ma et al., dkv-cache: The cache for diffusion language models

---

### Decision · Program_Chairs · 2026-04-30

**Decision:**

Accept (regular)

**Comment:**

The reviews were overall positive and support acceptance. Reviewers found the core idea novel and technically meaningful, especially the causal-attention hybrid that enables exact KV caching and a strong speed–quality tradeoff. The rebuttal also addressed most major concerns with useful clarifications and additional experiments. The remaining issues mainly concern presentation and the fact that Eso-LMs still do not surpass AR models in perplexity, but these did not substantially weaken the overall case for the paper.